# AN EQUIVALENCE BETWEEN DATA POISONING AND BYZANTINE GRADIENT ATTACKS

## ABSTRACT

To study the resilience of distributed learning, the "Byzantine" literature considers a strong threat model where workers can report arbitrary gradients to the parameter server. While this model helped generate several fundamental results, it has however sometimes been considered unrealistic, when the workers are mostly trustworthy machines. In this paper, we show a surprising equivalence between this model and data poisoning, a threat considered much more realistic. More specifically, we prove that any gradient attack can be reduced to data poisoning in a personalized federated learning system that provides PAC guarantees (which we show are both desirable and realistic in various personalized federated learning contexts such as linear regression and classification). Maybe most importantly, we derive a simple and practical attack that may be constructed against classical personalized federated learning models, and we show both theoretically and empirically the effectiveness of this attack.

## 1 INTRODUCTION

Learning algorithms now typically leverage data generated by a large number of users (Smith et al., 2013; Wang et al., 2019a;b) to often learn a common model that fits a large population (Konecný et al., 2015), but also sometimes to construct a *personalized* model for each individual (Ricci et al., 2011). Autocompletion (Lehmann & Buschek, 2021), conversational (Shum et al., 2018) and recommendation (Ie et al., 2019) algorithms are examples of such algorithms deployed at scale. To be effective, besides huge amounts of (distributed) data (Brown et al., 2020; Fedus et al., 2021), these algorithms require a high level of customization. This has motivated research into *personalized federated learning* (Fallah et al., 2020; Hanzely et al., 2020; Dinh et al., 2020).

However, in applications such as content recommendation, activists, companies, and politicians have strong incentives to promote certain views, products or ideologies (Hoang, 2020; Hoang et al., 2021). Remember for instance that, on YouTube, two views out of three result from algorithmic recommendations (Solsman, 2018). Perhaps unsurprisingly, this has led to vast amounts of fabricated activities to bias algorithms (Bradshaw & Howard, 2019; Neudert et al., 2019), like "fake reviews" (Wu et al., 2020). The scale of this phenomenon is well illustrated by the case of Facebook which, in 2019 alone, reported the removal of around 6 billion fake accounts from its platform (Fung & Garcia, 2019). This is particularly concerning in the era of "stochastic parrots" (Bender et al., 2021): climate denialists are incentivized to pollute textual datasets with claims like "climate change is a hoax", as autocompletion, conversational and recommendation algorithms trained on such data will more likely spread these views (McGuffie & Newhouse, 2020). This raises serious concerns about the vulnerability of personalized federated learning to such misleading data. Data poisoning attacks clearly constitute now a major machine learning security issue (Kumar et al., 2020).

Overall, in highly adversarial environments like social media, given the advent of *deep fakes* (Johnson & Diakopoulos, 2021), we should expect that *most data are strategically crafted and labeled*. In this context, the authentication of the data provider seems critical. In particular, the safety of learning algorithms arguably demands that they be trained solely on *cryptographically signed* data, that is, data that provably come from a known source. But even signed data cannot be wholeheartedly trusted since users usually have preferences over what ought to be recommended to others. They thus have incentives to behave strategically in order to promote certain views or products.

To address data poisoning, the Byzantine learning literature usually considers that each federated learning worker may behave arbitrarily (Blanchard et al., 2017; Mhamdi et al., 2018; El-Mhamdi et al., 2021). Recall that at each iteration of the federated learning stochastic gradient descent, each worker is given the updated model, and is asked to compute the gradient of the loss function with respect to (a batch of) its local data. Byzantine learning usually assumes that a malicious (Byzantine) worker may report *any* gradient; without having to justify whether such a gradient could have been generated through data poisoning. In fact, the gradient attack threat model has sometimes been claimed to be unrealistic in practical federated learning (Shejwalkar et al., 2021), especially when the workers are machines owned by trusted entities (cross-silo FL (Kairouz et al., 2021)).

We prove in this paper an equivalence between gradients attacks and fabricated data injection, in a general and desirable collaborative learning framework. Thereby, our paper provides the first practically compelling argument for the necessity to protect federated learning against gradient attacks.

**Contributions.** As a preamble of our main result, we formalize local PAC* learning[1] (Valiant, 1984) for personalized learning, and prove that a simple and general solution to personalized federated linear regression and classification is indeed locally PAC* learning. Our proof leverages a new concept called *gradient-PAC* learning*, which is of independent interest. We prove that it is sufficient to guarantee local PAC* learning, and that it is verified by basic learning algorithms, like linear and logistic regression. This is an important and highly nontrivial contribution of this paper.

Our main contribution is then to prove that local PAC* learning in personalized federated learning essentially implies an equivalence between data poisoning and gradient attacks. More precisely, we show how any (converging) gradient attack can be turned into a data poisoning attack, with the same resulting harm. Given how easy it generally is to create fake accounts on web platforms and to inject data poisoning by generating fake activities, this result should arguably greatly increase the concerns over the vulnerabilities of federated learning with user-generated data.

Finally, we propose a simple but very general *strategic* gradient attack, called the *counter-gradient attack* (CGA), which any participant to federated learning can deploy to bias the global model towards any target model that better suits their interest. We prove the effectiveness of this attack under fairly general assumptions, which apply to many proposed personalized learning frameworks including Hanzely et al. (2020); Dinh et al. (2020). We then show empirically how this gradient attack can be turned into a devastating data poisoning attack, with remarkably few data.

**Related work.** Byzantine learning has provided both negative and positive results in Byzantine resilience (Blanchard et al., 2017; Mhamdi et al., 2018; Baruch et al., 2019; Xie et al., 2019), some of which apply almost straightforwardly to personalized federated learning (El-Mhamdi et al., 2020; 2021). Such results study the resilience against a minority of *adversarial* users. Our paper however focuses on a different kind of malicious users. Namely, like Suya et al. (2021), we study the resilience against *strategic* users, who aim to bias the learned models towards a specific target model.

The study of the resilience against strategic users is part of the research on *strategyproof* learning. Many special cases of strategyproofness have been tackled, including regression (Chen et al., 2018b; Dekel et al., 2010; Perote & Perote-Peña, 2004; Ben-Porat & Tennenholtz, 2017), classification (Meir et al., 2012; Chen et al., 2020; Meir et al., 2011; Hardt et al., 2016), statistical estimation (Cai et al., 2015), and clustering (Perote & Sevilla, 2003). However, none of these papers tackles a general *personalized federated learning* scheme. Typically, for linear regression, Chen et al. (2018b) and Perote & Perote-Peña (2004) assume that each user can only provide a single data point. This greatly restricts the users' ability to contribute to the learning model. And while Dekel et al. (2010) allows multiple contributions, they either require payments, which might not be possible (e.g., due to ethical reasons), or they restrict the model to one dimension or a constant function in $\mathbb{R}^d$. Conversely, Suya et al. (2021) show how to arbitrarily manipulate convex learning models through multiple data injections, in the case where a single model is learned from all data at once.

A large literature has focused on *data poisoning*, with either a focus on *backdoor* (Dai et al., 2019; Zhao et al., 2020; Severi et al., 2021; Truong et al., 2020; Schwarzschild et al., 2021) or *triggerless* attacks (Biggio et al., 2012; Muñoz-González et al., 2017; Shafahi et al., 2018; Zhu et al., 2019;

---

[1]We omit complexity considerations for the sake of generality. We define PAC* to be PAC without such considerations.

Huang et al., 2020; Barreno et al., 2006; Aghakhani et al., 2021; Geiping et al., 2021). However, most of this research analyzed data poisoning without *signed* data. One noteworthy exception is Mahloujifar et al. (2019), whose universal attack amplifies the probability of a (bad) property.

Collaborative PAC learning was previously introduced by Blum et al. (2017), and then extensively studied (Chen et al., 2018a; Nguyen & Zakynthinou, 2018), sometimes with the presence of Byzantine collaborating users (Qiao, 2018; Jain & Orlitsky, 2020; Konstantinov et al., 2020). We stress however that this line of work assumes that all honest users have the same labeling function. In other words, given any query, they agree on how the query should be answered. This is a very unrealistic assumption in many critical applications, like content moderation or language processing. In fact, in such applications, removing outliers can be argued to amount to ignoring minorities' views, which would be highly unethical. The very definition of PAC learning must then be adapted, which is what we do in this paper (also, we adapt it to parameterized models).

**Structure of the paper.** The rest of the paper is organized as follows. Section 2 presents a general model of personalized learning, formalizes local PAC* learning and defines a general federated gradient descent algorithm. Section 3 proves the equivalence between data poisoning and gradient attacks, under local PAC* learning. Section 4 proves the local PAC* learning properties for federated linear regression and classification. Section 5 then describes a simple and general data poisoning attack, whose effectiveness against $\ell_2^2$ is proved theoretically and empirically. Section 6 concludes.

## 2 A GENERAL PERSONALIZED FEDERATED LEARNING FRAMEWORK

We consider a set $[N] = \{1, \ldots, N\}$ of users. Each user $n \in [N]$ has a local *signed* dataset $\mathcal{D}_n$, and aims to learn a local model $\theta_n \in \mathbb{R}^d$. Users may collaborate to improve their models. Personalized learning must then input a tuple of users' local datasets $\vec{\mathcal{D}} \triangleq (\mathcal{D}_1, \ldots, \mathcal{D}_N)$, and output a tuple of local models $\vec{\theta}^* \triangleq (\theta_1^*, \ldots, \theta_N^*)$. Like many others, we assume that the users perform federated learning to do so, by leveraging the computation of a common global model $\rho \in \mathbb{R}^d$. Intuitively, the global model is an aggregate of all users' local models, which users can leverage to improve their local models. The common global model will typically allow users with too few data to obtain an effective local model, while it may be mostly discarded by users whose local datasets are large.

More formally, we consider a very general personalized learning framework which generalizes the models proposed by Dinh et al. (2020) and Hanzely et al. (2020). Namely, we consider that the personalized learning algorithm outputs a global minimum $(\rho^*, \vec{\theta}^*)$ of a global loss given by

$$\text{Loss}(\rho, \vec{\theta}, \vec{\mathcal{D}}) \triangleq \sum_{n \in [N]} \mathcal{L}_n(\theta_n, \mathcal{D}_n) + \sum_{n \in [N]} \mathcal{R}(\rho, \theta_n), \tag{1}$$

where $\mathcal{R}$ is a regularization, typically with a minimum at $\theta_n = \rho$. For instance, Hanzely et al. (2020) and Dinh et al. (2020) define $\mathcal{R}(\rho, \theta_n) \triangleq \lambda \|\rho - \theta_n\|_2^2$, which we shall call the $\ell_2^2$ regularization. But other regularizations may be considered, like the $\ell_2$ regularization $\mathcal{R}(\rho, \theta_n) \triangleq \lambda \|\rho - \theta_n\|_2$, or the smooth-$\ell_2$ regularization $\mathcal{R}(\rho, \theta_n) \triangleq \lambda \sqrt{1 + \|\rho - \theta_n\|_2^2}$. Note that, for all such regularizations, the limit $\lambda \to \infty$ essentially yields the classical non-personalized federated learning framework.

### 2.1 LOCAL PAC* LEARNING

In this paper, we focus on personalized learning algorithms that provably recover a user $n$'s *preferred model* $\theta_n^\dagger$, if the user provides a large enough *honest dataset* $\mathcal{D}_n$, i.e. constructed with $\theta_n^\dagger$. Such honest datasets $\mathcal{D}_n$ could typically be obtained by repeatedly drawing random queries (or features), and by using the user's preferred model $\theta_n^\dagger$ to provide (potentially noisy) answers (or labels). We refer to Section 4 for examples. The model recovery condition is then formalized as follows.

**Definition 1.** *A personalized learning algorithm is locally PAC* learning if, for any subset $H \subset [N]$ of honest users, any preferred models $\vec{\theta}^\dagger$, any $\varepsilon, \delta > 0$, and any datasets $\vec{\mathcal{D}}_{-H}$ from users $n \notin H$, there exists $\mathcal{I}$ such that, if all users $h \in H$ provide honest datasets $\mathcal{D}_h$ with at least $|\mathcal{D}_h| \geq \mathcal{I}$ data points, then, with probability at least $1 - \delta$, we have $\left\| \theta_h^* \left( \vec{\mathcal{D}} \right) - \theta_h^\dagger \right\|_2 \leq \varepsilon$ for all users $h \in H$.*

Local PAC* learning is arguably a very desirable property. Indeed, it guarantees that any honest active user will not be discouraged to participate in federated learning as they will eventually learn their preferred model by providing more and more data. In Section 4, we will show how local PAC* learning can be achieved in practice, by considering specific local loss functions $\mathcal{L}_n$.

## 2.2 Federated gradient descent

While the computation of $\rho^*$ and $\vec{\theta}^*$ could be done by a single machine, which first collects the datasets $\vec{\mathcal{D}}$ and then minimizes the global loss LOSS, modern machine learning deployments often rather rely on *federated* (stochastic) gradient descent (or variants), with a central trusted parameter server. In this setting, each user $n$ keeps their data $\mathcal{D}_n$ locally. At each iteration $t$, the parameter server sends the latest global model $\rho^t$ to the users. Each user $n$ is then expected to update its local model given the global model $\rho^t$, either by solving $\theta_n^t \triangleq \arg\min_{\theta_n} \mathcal{L}_n(\theta_n, \mathcal{D}_n) + \mathcal{R}(\rho^t, \theta_n)$ (which is what we assume in the theory part, in the manner of Dinh et al. (2020)), or by making a (stochastic) gradient step from the previous local model $\theta_n^{t-1}$ (which is what is done in our experiments, in the manner of Hanzely & Richtárik (2021)). User $n$ is then expected to report the gradient $g_n^t = \nabla_\rho \mathcal{R}(\rho^t, \theta_n^t)$ to the parameter server. The parameter server then updates the global model, using a gradient step, i.e. it computes $\rho^{t+1} \triangleq \rho^t - \eta_t \sum_{n \in [N]} g_n^t$. For simplicity, here, and since our goal is to show the vulnerability of personalized federated learning even in good conditions, we assume that the network is synchronous and that no node can crash. Note also that our setting could be generalized to fully decentralized collaborative learning, as was done by El-Mhamdi et al. (2021).

We assume that the users are only allowed to send plausible gradient gradients. More precisely, we denote $\text{GRAD}(\rho) \triangleq \overline{\{\nabla_\rho \mathcal{R}(\rho, \theta) \mid \theta \in \mathbb{R}^d\}}$ the closure set of plausible (sub)gradients at $\rho$. If user $n$'s gradient $g_n^t$ is not in the set $\text{GRAD}(\rho^t)$, the parameter server can easily detect the malicious behavior and $g_n^t$ will be ignored by the parameter server at iteration $t$. In the case of an $\ell_2^2$ regularization, where $\mathcal{R}(\rho, \theta) = \lambda \|\rho - \theta\|_2^2$, we clearly have $\text{GRAD}(\rho) = \mathbb{R}^d$ for all $\rho \in \mathbb{R}^d$. It can be easily shown that, for $\ell_2$ and smooth-$\ell_2$ regularizations, $\text{GRAD}(\rho)$ is the closed ball $\mathcal{B}(0, \lambda)$.

Nevertheless, in this setting, a strategic user $s \in [N]$ can deviate from its expected behavior, to bias the global model in their favor. We identify in particular three sorts of attacks.

**Data poisoning:** Instead of collecting an honest dataset, $s$ fabricates any *strategically crafted* dataset $\mathcal{D}_s$, and then performs all other operations as expected.

**Model attack:** At each iteration $t$, $s$ fixes $\theta_s^t \triangleq \theta_s^\spadesuit$, where $\theta_s^\spadesuit$ is any *strategically crafted* model. All other operations would then be executed as expected.

**Gradient attack:** At each iteration $t$, $s$ sends any (plausible) *strategically crafted* gradient $g_s^t$.

Gradient attacks are intuitively the most harmful attacks, as the strategic user can then adapt their attack based on what they observe during the learning process. However, because of this, gradient attacks are more likely to be flagged as suspicious behaviors. At the other end of the spectrum, data poisoning is a much safer attack, as the strategic user can always report their entire dataset, and prove that they rigorously performed the expected computations. In fact, data poisoning can be executed, even if users directly provide the data to a (trusted) central authority, which is then tasked to execute (federated) gradient descent. This is typically what is done to construct recommendation algorithms on social medias, where users' data are their online activities (what they view, like and share). Crucially, especially in applications with no clear ground truth, such as content moderation or language processing, the strategic user can always argue that their dataset is an "honest" dataset; not a strategically crafted one. Ignoring the strategic user's data on the basis that it is an "outlier" may then be regarded as *unethical*, as it can be argued to amount to rejecting minorities' viewpoints.

## 3 The equivalence between data poisoning and gradient attacks

We now present our main result. The threat model we considered is closely related to "model-targeted attacks", studied in Suya et al. (2021). Recall also that, in this theory part, at each iteration, local models $\theta_n^t$ are fully optimized, given $\rho^t$, in the manner of Dinh et al. (2020).

**Theorem 1** (Equivalence between gradient attacks and data poisoning)**.** *Assume local PAC\* learning, and $\ell_2^2$ or smooth-$\ell_2$ regularization. Assume also that $\mathcal{L}_n$ is convex and $L$-smooth for all users*

$n \in [N]$, and that the learning rate $\eta_t$ is constant and small enough. Consider any datasets $\vec{\mathcal{D}}_{-s}$ provided by users $n \neq s$. Then, for any target model $\theta_s^\dagger \in \mathbb{R}^d$, there exists a converging gradient attack of strategic user $s$ (i.e. $g_s^t$ converges) such that $\rho^t \to \theta_s^\dagger$, if and only if, for any $\varepsilon > 0$, there exists a dataset $\mathcal{D}_s$ such that $\left\| \rho^*(\vec{\mathcal{D}}) - \theta_s^\dagger \right\|_2 \leq \varepsilon$.

Note that smoothness is used as a sufficient condition to prove the convergence of federated gradient descent. We now present our proof, which goes through the intermediary step of model attacks.

## 3.1 EQUIVALENCE BETWEEN DATA POISONING AND MODEL ATTACKS

In this section, we prove the equivalence between data poisoning and model attacks.

**Lemma 1** (Reduction from model attack to data poisoning). *Consider any data $\vec{\mathcal{D}}$ and user $s \in [N]$. Assume the global loss with datasets $\vec{\mathcal{D}}$ has a global minimum $(\rho^*, \vec{\theta}^*)$. Then $(\rho^*, \vec{\theta}^*_{-s})$ is also a global minimum of the modified loss with datasets $\vec{\mathcal{D}}_{-s}$ and strategic reporting $\theta_s^\spadesuit \triangleq \theta_s^*(\vec{\mathcal{D}})$.*

*Proof.* The proof is almost straightforward. It is given in Appendix B.1. □

Now, intuitively, by virtue of local PAC* learning, strategic user $s$ can essentially guarantee that the personalized learning framework will be learning $\theta_s^* \approx \theta_s^\spadesuit$. But, a priori, it may seem unclear if this will imply a biasing of the global model essentially identical to the one obtained through the model attack that imposes $\theta_s^* = \theta_s^\spadesuit$. In the sequel, we show that the answer is yes.

**Lemma 2** (Reduction from data poisoning to model attack). *Assume $\ell_2^2$, $\ell_2$ or smooth-$\ell_2$ regularization, and assume local PAC* learning. Consider any datasets $\mathcal{D}_{-s}$ and any attack model $\theta_s^\spadesuit$ such that the modified loss $\mathrm{LOSS}_s$ has a unique minimum $\rho^*(\theta_s^\spadesuit, \vec{\mathcal{D}}_{-s}), \vec{\theta}^*_{-s}(\theta_s^\spadesuit, \vec{\mathcal{D}}_{-s})$. Then, for any $\varepsilon > 0$, there exists a dataset $\mathcal{D}_s$ such that we have*

$$\left\| \rho^*(\vec{\mathcal{D}}) - \rho^*(\theta_s^\spadesuit, \vec{\mathcal{D}}_{-s}) \right\|_2 \leq \varepsilon \ \text{ and } \ \forall n \neq s, \ \left\| \theta_n^*(\vec{\mathcal{D}}) - \theta_n^*(\theta_s^\spadesuit, \vec{\mathcal{D}}_{-s}) \right\|_2 \leq \varepsilon. \tag{2}$$

Our proof in fact holds for all continuous regularizations $\mathcal{R}$ with $\mathcal{R}(\rho, \theta) \to \infty$ as $\|\rho - \theta\|_2 \to \infty$. Moreover, note that the approximation guarantee holds for local models too.

*Sketch of proof.* Given local PAC* learning, for a large dataset $\mathcal{D}_s$ constructed from $\theta_s^\spadesuit$, strategic user $s$ can guarantee $\theta_s^*(\vec{\mathcal{D}}) \approx \theta_s^\spadesuit$. By carefully bounding the effect of the approximation on the loss using the Heine-Cantor theorem, we show that this implies $\rho^*(\vec{\mathcal{D}}) \approx \rho^*(\theta_s^\spadesuit, \vec{\mathcal{D}}_{-s})$ and $\theta_n^*(\vec{\mathcal{D}}) \approx \theta_n^*(\theta_s^\spadesuit, \vec{\mathcal{D}}_{-s})$ for all $n \neq s$ too. The precise analysis, given in Appendix B.2, is nontrivial. □

## 3.2 EQUIVALENCE BETWEEN MODEL ATTACKS AND GRADIENT ATTACKS

For the last few years, gradient attacks have been widely studied by the Byzantine learning literature. Recently, Shejwalkar et al. (2021) argued that they are not a realistic threat model. Below, we prove that, in our setting, they are actually as concerning as model attacks (and, thus, as data poisoning).

**Lemma 3** (Reduction from model attack to gradient attack). *Assume that $\mathcal{L}_n$ is convex and $L$-smooth for all nodes $n \in [N]$, and that we use $\ell_2^2$ or smooth-$\ell_2$ regularization. If $g_s^t$ converges and if $\eta_t = \eta$ is a constant small enough, then $\rho^t$ will converge too. Denote $\rho^\infty$ its limit. Then for any $\varepsilon > 0$, there is $\theta_s^\spadesuit \in \mathbb{R}^d$ such that $\left\| \rho^\infty - \rho^*(\theta_s^\spadesuit, \vec{\mathcal{D}}_{-s}) \right\|_2 \leq \varepsilon$.*

*Sketch of proof.* Denote $g_s^\infty$ the limit of $g_s^t$. Gradient descent then behaves as though it was minimizing the loss plus $\rho^T g_s^\infty$ (and ignoring $\mathcal{R}(\rho, \theta_s)$). Essentially, classical gradient descent theory then guarantees the convergence of $\rho^t$ to $\rho^\infty$, though the precise proof is nontrivial. Then, since GRAD is closed and $g_s^\infty \in$ GRAD, we can construct $\theta_s^\spadesuit$ which approximately yields the gradient $g_s^\infty$. The full proof (which also yields the necessary upper bound on $\eta$) is given in Appendix B.3. □

Since any model attack can clearly be achieved by the corresponding honest gradient attack, we conclude that model attacks and gradient attacks are essentially equivalent. In light of our previous results, this implies that gradient attacks are essentially equivalent to data poisoning (Theorem 1).

## 4 EXAMPLES OF LOCALLY PAC* LEARNING SYSTEMS

To the best of our knowledge, though similar to collaborative PAC learning (Blum et al., 2017), local PAC* learnability is a new concept in the context of personalized federated learning. It is thus important to show that it is not meaningless. To achieve this, in this section, we provide *sufficient* conditions for a personalized learning model to be locally PAC* learnable. First, we construct local losses $\mathcal{L}_n$ as sums of losses per input, i.e.

$$\mathcal{L}_n(\theta_n, \mathcal{D}_n) = \nu \|\theta_n\|_2^2 + \sum_{x \in \mathcal{D}_n} \ell(\theta_n, x), \tag{3}$$

for some "loss per input" function $\ell$ and some weight $\nu > 0$. Appendix C gives theoretical and empirical arguments are provided for using such a sum (as opposed to an expectation). Remarkably, for linear or logistic regression, given such a loss, local PAC* learning can then be guaranteed.

**Theorem 2** (Personalized least square linear regression is locally PAC* learning). *Consider $\ell_2^2$, $\ell_2$ or smooth-$\ell_2$ regularization. Assume that, to generate a data $x_i$, a user with preferred parameter $\theta^\dagger \in \mathbb{R}^d$ first independently draws a random vector query $\mathcal{Q}_i \in \mathbb{R}^d$ from a sub-Gaussian query distribution $\tilde{\mathcal{Q}}$, with parameter $\sigma_{\mathcal{Q}}$ and positive definite matrix $\Sigma = \mathbb{E}\left[\mathcal{Q}_i \mathcal{Q}_i^T\right]$. Assume that the user labels $\mathcal{Q}_i$ with a real-valued answer $\mathcal{A}_i = \mathcal{Q}_i^T \theta^\dagger + \xi_i$, where $\xi_i$ is a zero-mean sub-Gaussian random noise with parameter $\sigma_\xi$, independent from $\mathcal{Q}_i$ and other data points. Finally, assume that $\ell(\theta, \mathcal{Q}_i, \mathcal{A}_i) = \frac{1}{2}(\theta^T \mathcal{Q}_i - \mathcal{A}_i)^2$. Then the personalized learning algorithm is locally PAC* learning.*

**Theorem 3** (Personalized logistic regression is locally PAC*-learning). *Consider $\ell_2^2$, $\ell_2$ or smooth-$\ell_2$ regularization. Assume that, to generate a data $x_i$, a user with preferred parameter $\theta^\dagger \in \mathbb{R}^d$ first independently draws a random vector query $\mathcal{Q}_i \in \mathbb{R}^d$ from a query distribution $\tilde{\mathcal{Q}}$, whose support $\mathrm{SUPP}(\tilde{\mathcal{Q}})$ is bounded and spans the full vector space $\mathbb{R}^d$. Assume that the user then labels $\mathcal{Q}_i$ with answer $\mathcal{A}_i = 1$ with probability $\sigma(\mathcal{Q}_i^T \theta^\dagger)$, and labels it $\mathcal{A}_i = -1$ otherwise, where $\sigma(z) = (1 + e^{-z})^{-1}$ is the sigmoid logistic function. Finally, assume that $\ell(\theta, \mathcal{Q}_i, \mathcal{A}_i) = -\ln(\sigma(\mathcal{A}_i \theta^T \mathcal{Q}_i))$. Then the personalized learning algorithm is locally PAC* learning.*

### 4.1 PROOF SKETCH

In this section, we outline the proofs of theorems 2 and 3, as they are interesting in their own sake. The proofs both leverage the following notion, which intuitively means "robust PAC* learning".

**Definition 2** (Gradient-PAC*). *Denote $\mathcal{E}(\mathcal{D}, \theta^\dagger, \mathcal{I}, A, B, \alpha)$ the event*

$$\forall \theta \in \mathbb{R}^d, \ \left(\theta - \theta^\dagger\right)^T \nabla \mathcal{L}\left(\theta, \mathcal{D}\right) \geq A\mathcal{I} \min\left\{\left\|\theta - \theta^\dagger\right\|_2, \left\|\theta - \theta^\dagger\right\|_2^2\right\} - B\mathcal{I}^\alpha \left\|\theta - \theta^\dagger\right\|_2. \tag{4}$$

*The loss $\ell$ is gradient-PAC* if, for any $\mathcal{K} > 0$, there exist constants $A_\mathcal{K}, B_\mathcal{K} > 0$ and $\alpha_\mathcal{K} < 1$, such that for any preferred model $\theta^\dagger \in \mathbb{R}^d$ with $\left\|\theta^\dagger\right\|_2 \leq \mathcal{K}$, assuming that the dataset $\mathcal{D}$ is obtained by honestly collecting and labeling $\mathcal{I}$ data points according to the preferred model $\theta^\dagger$, the probability of the event $\mathcal{E}(\mathcal{D}, \theta^\dagger, \mathcal{I}, A_\mathcal{K}, B_\mathcal{K}, \alpha_\mathcal{K})$ goes to 1 as $\mathcal{I} \to \infty$.*

Intuitively, this definition asserts that, as we collect more data from an honest user, then, with high probability, the gradient of the loss at any point $\theta$ too far from $\theta^\dagger$ will point away from $\theta^\dagger$. In particular, gradient descent is then essentially guaranteed to draw $\theta$ closer to $\theta^\dagger$. The right-hand side of equation 4 is subtly chosen to be strong enough to yield local PAC* learning guarantees, and weak enough to be verified by linear and logistic regression, as proved by the following lemma.

**Lemma 4.** *Logistic and linear regression, defined in theorems 2 and 3, are gradient PAC* learning.*

*Sketch of proof.* In the case of linear regression, remarkably, the discrepancy between the empirical and the expected loss functions depends only on a few key random variables, such as $\min \mathrm{SP}\left(\frac{1}{\mathcal{I}} \sum \mathcal{Q}_i \mathcal{Q}_i^T\right)$ and $\sum \xi_i \mathcal{Q}_i$, which can be controlled by appropriate concentration bounds. Meanwhile, for logistic regression, for $|b| \leq \mathcal{K}$, we observe that $(a - b)(\sigma(a) - \sigma(b)) \geq c_\mathcal{K} \min(|a - b|, |a - b|^2)$. Essentially, this proves that gradient-PAC* would hold if the empirical loss was replaced by the expected loss (times $\mathcal{I}$). The actual proofs, however, are nontrivial, especially in the case of logistic regression, which leverages topological considerations to derive a critical uniform concentration bound. The full proofs are given in appendices D.2 and D.3. □

Now, under very mild assumptions on the regularization $\mathcal{R}$ (not even convexity!), which are verified by the $\ell_2^2$, $\ell_2$ and smooth-$\ell_2$ regularizations, we prove that the gradient-PAC* learnability through $\ell$ suffices to guarantee that personalized learning will be locally PAC* learning.

**Lemma 5.** *Assume $\mathcal{R}$ is (sub)differentiable and nonnegative, and $\mathcal{R}(\rho, \theta) \to \infty$ as $\|\rho - \theta\|_2 \to \infty$. If $\ell$ is gradient-PAC* and nonnegative, then personalized learning is locally PAC*-learning.*

*Sketch of proof.* Once other users' datasets are fixed, the learning of an honest user's model has a fixed biased due to $\mathcal{R}$. But as the user provides more data, by gradient-PAC*, the local loss becomes predominant, which guarantees local PAC*-learning. Appendix E provides a full proof. $\square$

Combining the two lemmas clearly yields theorems 2 and 3 as special cases. Note that our result actually applies to a more general set of regularizations and a more general set of local losses.

### 4.2 THE CASE OF NEURAL NETWORKS

Neural networks generally do *not* verify gradient PAC* learning. After all, because of symmetries like neuron swapping, different values of the parameters compute the same neural network function. Thus the notion of a "preferred model" $\theta^\dagger$ is arguably ill-defined for neural networks[2]. Nevertheless, we may consider a strategic user who only aims to bias the parameters of the last layer. In particular, assuming that all layers but the last one of a neural network are pretrained and fixed, then our theory may apply to the parameters of the last layer, if it performs classification or regression.

## 5 A PRACTICAL DATA POISONING ATTACK

We now construct a practical data poisoning attack. We do so by introducing a new effective gradient attack, and by then leveraging our equivalence to turn it into a data poisoning attack.

### 5.1 THE COUNTER-GRADIENT ATTACK

We now define a simple, general and practical gradient attack, which we call the counter-gradient attack (CGA). Intuitively, this attack estimates the sum $g_{-s}^{\dagger, t}$ of the gradients of other users based on its value at the previous iteration, which can be inferred from the way the global model $\rho^{t-1}$ was updated into $\rho^t$. More precisely, apart from initialization $\hat{g}_{-s}^1 \triangleq 0$, CGA makes the estimation

$$\hat{g}_{-s}^t \triangleq \frac{\rho^{t-1} - \rho^t}{\eta_{t-1}} - g_s^{t-1} = g_{-s}^{\dagger, t-1}. \tag{5}$$

Strategic user $s$ then reports the plausible gradient that moves the global model closest to the user's target model $\theta_s^\dagger$, assuming others report $\hat{g}_{-s}^t$. In other words, at every iteration, CGA reports

$$g_s^t \in \underset{g \in \text{GRAD}(\rho)}{\arg\min} \left\| \rho^t - \eta_t(\hat{g}_{-s}^t + g) - \theta_s^\dagger \right\|_2. \tag{6}$$

Note that, to perform this attack, user $s$ only needs to know the previous and current learning rates $\eta_{t-1}$ and $\eta_t$, the previous and current global models $\rho^{t-1}$ and $\rho^t$, and their target model $\theta_s^\dagger$.

**Computation of CGA.** Define $h_s^t \triangleq g_s^{t-1} + \frac{\rho^t - \theta_s^\dagger}{\eta_t} - \frac{\rho^{t-1} - \rho^t}{\eta_{t-1}}$. For convex sets $\text{GRAD}(\rho^t)$, it is straightforward to see that CGA boils down to computing the orthogonal projection of $h_s^t$ on $\text{GRAD}(\rho^t)$. This yields very simple computations for $\ell_2^2$, $\ell_2$ and smooth-$\ell_2$ regularizations.

**Proposition 1.** *For $\ell_2^2$ regularization, CGA reports $g_s^t = h_s^t$. For $\ell_2$ or smooth-$\ell_2$ regularization, CGA reports $g_s^t = h_s^t \min\{1, \lambda/\|h_s^t\|_2\}$.*

*Proof.* Equation (6) boils down to minimizing the distance between $\frac{\rho^t - \theta_s^\dagger}{\eta_t} - \hat{g}_{-s}^t$ and $\text{GRAD}(\rho)$, which is the (convex) ball $\mathcal{B}(0, \lambda)$. This minimum is the orthogonal projection on $\mathcal{B}(0, \lambda)$. $\square$

---

[2]Evidently, our definition could be easily modified to give importance to the computed function, rather than to the parameters of the model.

**Theoretical analysis.** We now prove that CGA is perfectly successful against $\ell_2^2$ regularization. To do so, we suppose that, at each iteration $t$ and for each user $n \neq s$, the local models $\theta_n$ are fully optimized with respect to $\rho^t$, and the honest gradients of $g_n^{\dagger,t}$ are used for the gradient descent of $\rho$.

**Theorem 4.** *Consider $\ell_2^2$ regularization. Assume that $\ell$ is convex and $L_\ell$-smooth, and that $\eta_t = \eta$ is small enough. Then CGA is converging and optimal, as $\rho^t \to \theta_s^\dagger$.*

*Sketch of proof.* The main challenge is to guarantee that the other users' gradients $g_n^{\dagger,t}$ for $n \neq s$ remain sufficiently stable over time to guarantee convergence, which can be done by leveraging $L$-smoothness. The full proof, with the necessary upper-bound on $\eta$, is given in Appendix F. $\square$

The analysis of the convergence against smooth-$\ell_2$ is unfortunately significantly more challenging. Here, we simply make a remark about what CGA yields in this case, if it converges.

**Proposition 2.** *If CGA against smooth-$\ell_2$ regularization converges for $\eta_t = \eta$, then it either achieves perfect manipulation, or it is eventually partially honest, in the sense that the gradient by CGA correctly points towards $\theta_s^\dagger$.*

*Proof.* Denote $P$ the projection onto the closed ball $\mathcal{B}(0, \lambda)$. If CGA converges, then, by Proposition 1, $P\left(g_s^\infty + \frac{\rho^\infty - \theta_s^\dagger}{\eta}\right) = g_s^\infty$. This implies in particular that $\rho^\infty - \theta_s^\dagger$ and $g_s^\infty$ must be colinear. If perfect manipulation is not achieved (i.e. $\rho^\infty \neq \theta_s^\dagger$), then we must have $g_s^\infty = \lambda \frac{\rho^\infty - \theta_s^\dagger}{\left\|\rho^\infty - \theta_s^\dagger\right\|_2}$. $\square$

**Empirical evaluation of CGA.** We deployed CGA to bias the federated learning of MNIST. We consider a strategic user whose target model is one that labels 0's as 1's, 1's as 2's, and so on, until 9's that are labeled as 0's. In particular, this target model has a nil accuracy. Figure 1 shows that such a user effectively hacks the $\ell_2^2$ regularization against 10 honest users who each have 6,000 data points of MNIST, in the case where local models only undergo a single gradient step at each iteration, but fails to hack the $\ell_2$ regularization. Further details on the experiment are given in Appendix G. We also ran a similar successful attack on the last layer of a deep neural network trained on cifar-10, which is detailed in Appendix J. The Appendix also discusses the extent to which the attack may be turned into data poisoning.

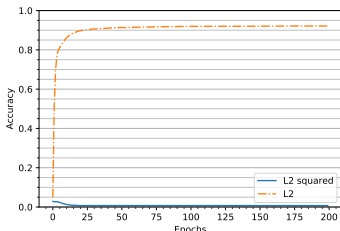

Figure 1: Accuracy of the global model under attack by CGA.

## 5.2 FROM GRADIENT ATTACK TO MODEL ATTACK AGAINST $\ell_2^2$

We now show how to turn a gradient attack into model attack, against $\ell_2^2$ regularization. Assume that we found a gradient $g_s^\infty$ such that $\rho^\infty = \theta_s^\dagger$. It is trivial to transform it into a model attack by setting $\theta_s^\spadesuit \triangleq \theta_s^\dagger - \frac{1}{2} g_s^\infty$, as guaranteed by the following result, and as depicted by Figure 2.

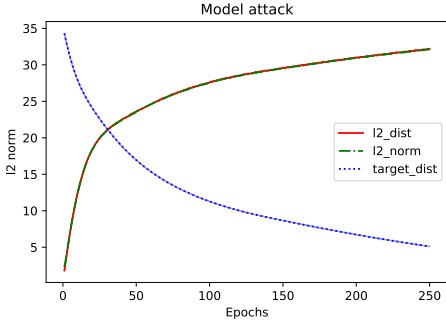

(a) Distance between the global model $\rho^t$ and the target model $\theta_s^\dagger$ (target_dist).

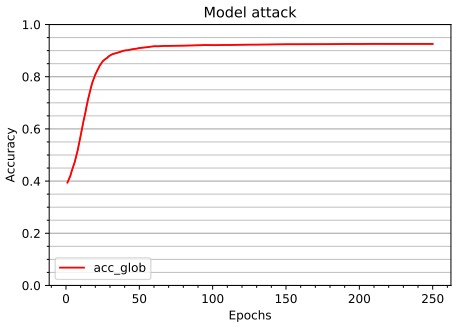

(b) Accuracy of $\rho^t$ according to $\theta_s^\dagger$ (which relabels $0 \to 1 \to 2 \to ... \to 9 \to 0$).

Figure 2: Successful model attack against $\ell_2^2$ by combining CGA and Proposition 3.

**Proposition 3.** *Consider the $\ell_2^2$ regularization. Suppose that $g_s^t \to g_s^\infty$ and $\rho^t \to \theta_s^\dagger$, with a constant learning rate $\eta_t = \eta$. Then, under the model attack $\theta_s^\spadesuit \triangleq \theta_s^\dagger - \frac{1}{2\lambda}g_s^\infty$, the gradient at $\rho = \theta_s^\dagger$ vanishes, i.e. $\nabla_\rho \mathrm{Loss}(\theta_s^\dagger, \vec{\theta}_{-s}^*(\theta_s^\dagger, \vec{\mathcal{D}}_{-s}), \theta_s^\spadesuit, \mathcal{D}_{-s}) = 0$.*

*Proof.* Given that the learning rate is constant, the convergence $\rho^t \to \theta_s^\dagger$ implies that the sum of honest users' gradients at $\rho = \theta_s^\dagger$ must equal $-g_s^\infty$. Therefore, to achieve $\rho^* = \theta_s^\dagger$ with a model attack, it suffices to send a model $\theta_s^\spadesuit$ such that the gradient of $\lambda \left\| \rho - \theta_s^\spadesuit \right\|_2^2$ with respect to $\rho$ at $\rho = \theta_s^\dagger$ equals $g_s^\infty$. Since the gradient is $\lambda(\theta_s^\dagger - \theta_s^\spadesuit)$, $\theta_s^\spadesuit \triangleq \theta_s^\dagger - \frac{1}{2\lambda}g_s^\infty$ does the trick. $\qquad \square$

### 5.3 FROM MODEL ATTACK TO DATA POISONING AGAINST $\ell_2^2$

**The case of linear regression.** In linear regression, any model attack can be turned into a *single data* poisoning attack, as proved by the following theorem whose proof is given in Appendix H.

**Theorem 5.** *Consider the $\ell_2^2$ regularization and linear regression. For any data $\mathcal{D}_{-s}$ and any target value $\theta_s^\dagger$, there is a datapoint $(\mathcal{Q}, \mathcal{A})$ to be injected by user $s$ such that $\rho^*(\{(\mathcal{Q}, \mathcal{A})\}, \mathcal{D}_{-s}) = \theta_s^\dagger$.*

**The case of linear classification.** We now consider linear classification, with the case of MNIST. By Theorem 2, any model attack can be turned into data poisoning, provided sufficiently many (random) data points are labeled by the strategic user. However, this may require creating too many data labelings, especially if the norm of $\theta_s^\spadesuit$ is large (which is the case if $s$ is alone against many active users), as suggested by Theorem 3.

For efficient data poisoning, define the indifference affine subspace $V \subset \mathbb{R}^d$ as the set of images with equiprobable labels. Intuitively, labeling images close to $V$ is very informative, as it informs us directly about the separating hyperplanes. To generate images, we first draw random images, which we then project orthogonally on $V$. We then add a small noise, before probabilistically labeling the image with model $\theta_s^\spadesuit$. Note that this leads us to consider images not in $[0, 1]^d$. Figure 3 shows the effectiveness of the resulting data poisoning attack, with only 2,000 data points, as opposed to the other nodes' 6,000 data points. More details and explanations are provided in Appendix I.

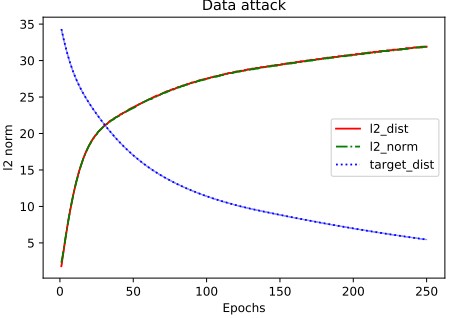
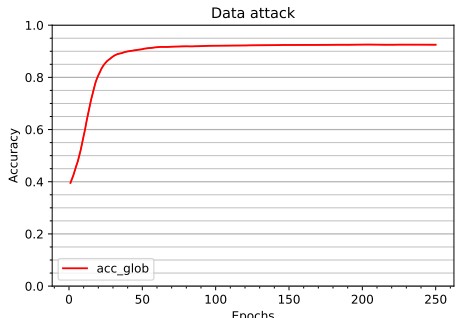

(a) Distance between the global model $\rho^t$ and the target model $\theta_s^\dagger$ (target_dist).

(b) Accuracy of $\rho^t$ according to $\theta_s^\dagger$ (which relabels $0 \to 1 \to 2 \to ... \to 9 \to 0$).

Figure 3: Successful data attack against $\ell_2^2$ by the efficient data generation scheme.

## 6 CONCLUDING REMARKS

We showed in this paper that, unlike what has been argued by, e.g., Shejwalkar et al. (2021), the gradient attack threat is not unrealistic. For personalized federated learning with local PAC* guarantees, gradient attacks are in fact just as realistic and harmful as strategic data reporting. More generally, our work stresses how critical Byzantine learning research is. For instance, El-Mhamdi et al. (2021) proved lower bounds on what *any* collaborative learning algorithm can guarantee in heterogeneous environments, under Byzantine gradient attacks. Our work implies that, at least for certain personalized federated learning problems, these lower bounds also hold for data poisoning attacks, which are known to be common for many high-risk applications. Arguably, a lot more security measures are urgently needed to make large-scale learning algorithms safe.

## ETHICS STATEMENT

The safety of algorithms is arguably a prerequisite to their ethics. After all, an arbitrarily manipulable large-scale algorithm will unavoidably endanger the targets of the entities that successfully design such algorithms. Typically, unsafe large-scale recommendation algorithms may be hacked by health disinformation campaigns that aim to promote non-certified products, e.g., by falsely pretending that they cure COVID-19. Such algorithms must not be regarded as ethical, even if they were designed with the best intentions. We believe that our work helps understand the vulnerabilities of such algorithms, and will motivate further research in the ethics and security of machine learning.

## REPRODUCIBILITY STATEMENT

All our experiments are run on the classical datasets MNIST and FashionMNIST. We provide all of the source codes to reproduce the experiments:

- The sum versus expectation experiments can be run by executing this file:
  `https://www.dropbox.com/sh/qdgmz9air24nhyr/AAAtycEkxc_1hGbvU5YG18z4a?dl=0`
- The counter-gradient attack experiments can be run by executing this file:
  `https://www.dropbox.com/sh/bycqkccgmk4muzn/AACRD1yeTglLSHEd1OOAzmVqa?dl=0`
- The data poisoning attack experiments can be run by executing this file:
  `https://www.dropbox.com/sh/qodnl6ivzti8hch/AADgX4EYuSOotiMCAHyTIiGMa?dl=0`
- The cifar10 on VGG 13-BN experiments can be run by executing this file:
  `https://www.dropbox.com/sh/5mw5c9dt25eiav7/AAC0jP3e2kuh_zvLudWbmDl-a?dl=0`

The experiments are seeded and the CuDNN backend is configured in deterministic mode in order to reduce the sources of non-determinism. We also turn of the benchmark mode. Executing the codes will generate the figures and statistics of our main paper, and most of the figures of our Appendix. Our other figures can be obtained by adjusting the hyperparameters of our codes.

The full description of the architecture and optimisation algorithm used is described in Appendix C. The experimental setup details of each experiment are provided in the Appendix, along with additional results. The Appendix also contains the full proofs of our theorems.

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

# Appendices

## A CONVEXITY LEMMAS

### A.1 GENERAL LEMMAS

**Definition 3.** *We say that $f : \mathbb{R}^d \to \mathbb{R}$ is locally strongly convex if, for any convex compact set $C \subset \mathbb{R}^d$, there exists $\mu > 0$ such that $f$ is $\mu$-strongly convex on $C$, i.e. for any $x, y \in C$ and any $\lambda \in [0, 1]$, we have*

$$f(\lambda x + (1 - \lambda)y) \leq \lambda f(x) + (1 - \lambda)f(y) - \frac{\mu}{2}\lambda(1 - \lambda)\left\| x - y \right\|_2^2. \tag{7}$$

*It is well-known that if $f$ is differentiable, this condition amounts to saying that $\left\| \nabla f(x) - \nabla f(y) \right\|_2 \geq \mu \left\| x - y \right\|_2$ for all $x, y \in C$. And if $f$ is twice differentiable, then it amounts to saying $\nabla^2 f(x) \succeq \mu I$ for all $x \in C$.*

**Lemma 6.** *If $f$ is locally strongly convex and $g$ is convex, then $f + g$ is locally strongly convex.*

*Proof.* Indeed, $(f + g)(\lambda x + (1 - \lambda)y) \leq \lambda f(x) + (1 - \lambda)f(y) - \frac{\mu}{2}\lambda(1 - \lambda)\left\| x - y \right\|_2^2 + \lambda g(x) + (1 - \lambda)g(y) = \lambda(f + g)(x) + (1 - \lambda)(f + g)(y) - \frac{\mu}{2}\lambda(1 - \lambda)\left\| x - y \right\|_2^2.$ □

**Definition 4.** *We say that $f : \mathbb{R}^d \to \mathbb{R}$ is $L$-smooth if it is differentiable and if its gradient is $L$-Lipschitz continuous, i.e. for any $x, y \in \mathbb{R}^d$,*

$$\left\| \nabla f(x) - \nabla f(y) \right\|_2 \leq L \left\| x - y \right\|_2. \tag{8}$$

**Lemma 7.** *If $f$ is $L_f$-smooth and $g$ is $L_g$-smooth, then $f + g$ is $(L_f + L_g)$-smooth.*

*Proof.* Indeed, $\left\| \nabla(f + g)(x) - \nabla(f + g)(y) \right\|_2 \leq \left\| \nabla f(x) - \nabla f(y) \right\|_2 + \left\| \nabla g(x) - \nabla g(y) \right\|_2 \leq L_f \left\| x - y \right\|_2 + L_g \left\| x - y \right\|_2 = (L_f + L_g)\left\| x - y \right\|_2.$ □

**Lemma 8.** *Suppose that $f : \mathbb{R}^d \times \mathbb{R}^{d'} \mapsto \mathbb{R}$ is locally strongly convex and $L$-smooth, and that, for any $x \in X$, where $X \subset \mathbb{R}^d$ is a convex compact subset, the map $y \mapsto f(x, y)$ has a minimum $y^*(x)$. Note that local strong convexity guarantees the uniqueness of this minimum. Then, there exists $K$ such that the function $y^*$ is $K$-Lipschitz continuous on $X$.*

*Proof.* The existence and uniqueness of $y^*(x)$ hold by strong convexity. Fix $x, x'$. By optimality of $y^*$, we know that $\nabla_y f(x, y^*(x)) = \nabla_y f(x', y^*(x')) = 0$. We then have the following bounds

$$\mu \left\| y^*(x) - y^*(x') \right\|_2 \leq \left\| \nabla_y f(x, y^*(x)) - \nabla_y f(x, y^*(x')) \right\|_2 = \left\| \nabla_y f(x, y^*(x')) \right\|_2 \tag{9}$$

$$= \left\| \nabla_y f(x, y^*(x')) - \nabla_y f(x', y^*(x')) \right\|_2 \tag{10}$$

$$\leq \left\| \nabla f(x, y^*(x')) - \nabla f(x', y^*(x')) \right\|_2 \tag{11}$$

$$\leq L \left\| (x - x', y^*(x') - y^*(x')) \right\|_2 = L \left\| x - x' \right\|_2, \tag{12}$$

where we first used the local strong convexity assumption, then the fact that $\nabla_y f(x, y^*(x)) = 0$, then the fact that $\nabla_y f(x', y^*(x')) = 0$, and then the $L$-smooth assumption. □

**Lemma 9.** *Suppose that $f : \mathbb{R}^d \times \mathbb{R}^{d'} \mapsto \mathbb{R}$ is locally strongly convex and $L$-smooth, and that, for any $x \in X$, where $X \subset \mathbb{R}^d$ is a convex compact subset, the map $y \mapsto f(x, y)$ has a minimum $y^*(x)$. Define $g(x) \triangleq \min_{y \in Y} f(x, y)$. Then $g$ is convex and differentiable on $X$ and $\nabla g(x) = \nabla_x f(x, y^*(x))$.*

*Proof.* First we prove that $g$ is convex. Let $x_1, x_2 \in \mathbb{R}^d$, and $\lambda_1, \lambda_2 \in [0, 1]$ with $\lambda_1 + \lambda_2 = 1$. For any $y_1, y_2 \in \mathbb{R}^{d'}$, we have

$$g(\lambda_1 x_1 + \lambda_2 x_2) = \min_{y \in \mathbb{R}^{d'}} f(\lambda_1 x_1 + \lambda_2 x_2, y) \tag{13}$$

$$\leq f(\lambda_1 x_1 + \lambda_2 x_2, \lambda_1 y_1 + \lambda_2 y_2) \tag{14}$$

$$\leq \lambda_1 f(x_1, y_1) + \lambda_2 f(x_2, y_2). \tag{15}$$

Taking the infimum of the right-hand side over $y_1$ and $y_2$ yields $g(\lambda_1 x_1 + \lambda_2 x_2) \leq \lambda_1 g(x_1) + \lambda_2 g(x_2)$, which proves the convexity of $g$.

Now denote $h(x) = \nabla_x f(x, y^*(x))$. We aim to show that $\nabla g(x) = h(x)$. Let $\varepsilon \in \mathbb{R}^d$ small enough so that $x + \varepsilon \in X$. Now note that we have

$$g(x + \varepsilon) = \min_{y \in \mathbb{R}^{d'}} f(x + \varepsilon, y) \leq f(x + \varepsilon, y^*(x)) \tag{16}$$

$$= f(x, y^*(x)) + \varepsilon^T \nabla_x f(x, y^*(x)) + o(\|\varepsilon\|_2) \tag{17}$$

$$= g(x) + \varepsilon^T h(x) + o(\|\varepsilon\|_2), \tag{18}$$

which shows that $h(x)$ is a superderivative of $g$ at $x$. We now show that it is also a subderivative. To do so, first note that its value at $x + \varepsilon$ is approximately the same, i.e.

$$\|h(x + \varepsilon) - h(x)\|_2 \leq \|\nabla_x f(x + \varepsilon, y^*(x + \varepsilon)) - \nabla_x f(x, y^*(x + \varepsilon))\|_2$$
$$+ \|\nabla_x f(x, y^*(x + \varepsilon)) - \nabla_x f(x, y^*(x))\|_2 \tag{19}$$

$$\leq L \|\varepsilon\|_2 + L \|y^*(x + \varepsilon) - y^*(x)\|_2 \leq \left( L + \frac{L^2}{\mu} \right) \|\varepsilon\|_2, \tag{20}$$

where we used the $L$-smoothness of $f$ and Lemma 8. Now notice that

$$g(x) = \min_{y \in \mathbb{R}^{d'}} f(x, y) \leq f(x, y^*(x + \varepsilon)) = f((x + \varepsilon) - \varepsilon, y^*(x + \varepsilon)) \tag{21}$$

$$= f(x + \varepsilon, y^*(x + \varepsilon)) - \varepsilon^T \nabla_x f(x + \varepsilon, y^*(x + \varepsilon)) + o(\|\varepsilon\|_2) \tag{22}$$

$$= g(x + \varepsilon) - \varepsilon^T h(x) - \varepsilon^T (h(x + \varepsilon) - h(x)) + o(\|\varepsilon\|_2), \tag{23}$$

But we know that $\|h(x + \varepsilon) - h(x)\|_2 = \mathcal{O}(\|\varepsilon\|_2)$. Rearranging the terms then yields

$$g(x + \varepsilon) \geq g(x) + \varepsilon^T h(x) - o(\|\varepsilon\|_2), \tag{24}$$

which shows that $h(x)$ is also a subderivative. Therefore, we know that $g(x + \varepsilon) = g(x) + \varepsilon^T h(x) + o(\|\varepsilon\|_2)$, which boils down to saying that $g$ is differentiable in $x \in X$, and that $\nabla g(x) = h(x)$. $\square$

**Lemma 10.** *Suppose that $f : X \times \mathbb{R}^{d'} \to \mathbb{R}$ is $\mu$-strongly convex, where $X \subset \mathbb{R}^d$ is closed and convex. Then $g : X \to \mathbb{R}$, defined by $g(x) = \inf_{y \in Y} f(x, y)$, is well-defined and $\mu$-strongly convex too.*

*Proof.* The function $y \mapsto f(x, y)$ is still strongly convex, which means that it is at least equal to a quadratic approximation around 0, which is a function that goes to infinity in all directions as $\|y\|_2 \to \infty$. This proves that the infimum must be reached within a compact set, which implies the existence of a minimum. Thus $g$ is well-defined. Moreover, for any $x_1, x_2 \in X, y_1, y_2 \in \mathbb{R}^{d'}$, and $\lambda_1, \lambda_2 \geq 0$ with $\lambda_1 + \lambda_2 = 1$, we have

$$g(\lambda_1 x_1 + \lambda_2 x_2) = \inf_y f(\lambda_1 x_1 + \lambda_2 x_2, y) \tag{25}$$

$$\leq f(\lambda_1 x_1 + \lambda_2 x_2, \lambda_1 y_1 + \lambda_2 y_2) \tag{26}$$

$$\leq \lambda_1 f(x_1, y_1) + \lambda_2 f(x_2, y_2) - \frac{\mu}{2} \lambda_1 \lambda_2 \|(x_1 - x_2, y_1 - y_2)\|_2^2 \tag{27}$$

$$\leq \lambda_1 f(x_1, y_1) + \lambda_2 f(x_2, y_2) - \frac{\mu}{2} \lambda_1 \lambda_2 \|x_1 - x_2\|_2^2, \tag{28}$$

where we used the $\mu$-strong convexity of $f$. Taking the infimum over $y_1, y_2$ implies the $\mu$-strong convexity of $g$. $\square$

### A.2 APPLICATIONS TO LOSS

Now instead of proving our theorems for different cases separately, we make the following assumptions on the components of the global loss that encompasses both $\ell_2^2$ and smooth-$\ell_2$ regularization, a well as linear regression and logistic regression.

**Assumption 1.** *Assume that $\ell$ is convex and $L_\ell$-smooth, and that $\mathcal{R}(\rho, \theta) = \mathcal{R}_0(\rho - \theta)$, where $\mathcal{R}_0 : \mathbb{R}^d \to \mathbb{R}$ is locally strongly convex (i.e. strongly convex on any convex compact set), $L_{\mathcal{R}_0}$-smooth and satisfy $\mathcal{R}_0(z) = \Omega(\|z\|_2)$ as $\|z\|_2 \to \infty$.*

**Lemma 11.** *Under Assumption 1,* LOSS *is locally strongly convex and L-smooth.*

*Proof.* All terms of LOSS are $L_0$-smooth, for an appropriate value of $L_0$. By Lemma 7, their sum is thus also $L$-smooth, for an appropriate value of $L$. Now, given Lemma 6, to prove that LOSS is locally strongly convex, it suffices to prove that $\nu \sum \|\theta_n\|_2^2 + \mathcal{R}_0(\rho - \theta_1)$ is locally strongly convex. Consider any convex compact set $C \subset \mathbb{R}^{d \times (1+N)}$. Since $\mathcal{R}_0$ is locally strongly convex, we know that there exists $\mu > 0$ such that $\nabla^2 \mathcal{R}_0 \succeq \mu I$. As a result,

$$(\rho, \vec{\theta})^T \left( \nabla^2 \text{LOSS} \right) (\rho, \vec{\theta}) \geq \nu \sum_{n \in [N]} \|\theta_n\|_2^2 + \mu \|\rho - \theta_1\|_2^2 \tag{29}$$

$$= \nu \|\theta_1\|_2^2 + \mu \|\rho\|_2^2 + \mu \|\theta_1\|_2^2 - 2\mu \rho^T \theta_1 + \nu \sum_{n \neq 1} \|\theta_n\|_2^2. \tag{30}$$

Now define $\alpha \triangleq \sqrt{\frac{2\mu}{\nu + 2\mu}}$. Clearly, $0 < \alpha < 1$. Moreover, $0 \leq \left\| \frac{1}{\alpha} \theta_1 - \alpha \rho \right\|_2^2 = \frac{1}{\alpha^2} \|\theta_1\|_2^2 + \alpha^2 \|\rho\|_2^2 - 2\rho^T \theta_1$. Therefore $2\rho^T \theta_1 \leq \alpha^2 \|\rho\|_2^2 + \frac{1}{\alpha^2} \|\theta_1\|_2^2$, which thus implies

$$(\rho, \vec{\theta})^T \left( \nabla^2 \text{LOSS} \right) (\rho, \vec{\theta}) \geq \left( \nu + \mu \left( 1 - \alpha^{-2} \right) \right) \|\theta_1\|_2^2 + \mu \left( 1 - \alpha^2 \right) \|\rho\|_2^2 + \nu \sum_{n \neq 1} \|\theta_n\|_2^2 \tag{31}$$

$$\geq \frac{\nu}{2} \|\theta_1\|_2^2 + \frac{2\nu\mu}{\nu + 2\mu} \|\rho\|_2^2 + \nu \sum_{n \neq 1} \|\theta_n\|_2^2 \geq \min \left\{ \frac{\nu}{2}, \frac{2\nu\mu}{\nu + 2\mu} \right\} \left\| (\rho, \vec{\theta}) \right\|_2^2, \tag{32}$$

which proves that $\nabla^2 \text{LOSS} \succeq \kappa I$, with $\kappa > 0$. This shows that LOSS is locally strongly convex. $\square$

**Lemma 12.** *Under Assumption 1,* $\rho \mapsto \vec{\theta}^*(\rho, \vec{\mathcal{D}})$ *is Lipchitz continuous on any compact set.*

*Proof.* Define $f_n(\rho, \theta_n) \triangleq \nu \|\theta_n\|_2^2 + \sum_{x \in \mathcal{D}_n} \ell(\theta_n, x) + \lambda \|\rho - \theta_n\|_2^2$. If $\ell$ is $L$-smooth, then $f_n$ is clearly $(|\mathcal{D}_n| L + \nu + \lambda)$-smooth. Moreover, if $\ell$ is convex, then for any $\rho$, the function $\theta_n \mapsto f_n(\rho, \theta_n)$ is at least $\nu$-strongly convex. Thus Lemma 8 applies, which guarantees that $\rho \mapsto \vec{\theta}^*(\rho, \vec{\mathcal{D}})$ is Lipchitz. $\square$

**Lemma 13.** *Under Assumption 1,* $\rho \mapsto \text{LOSS}(\rho, \vec{\theta}^*(\rho, \vec{\mathcal{D}}), \vec{\mathcal{D}})$ *is L-smooth and locally strongly convex.*

*Proof.* By Lemma 11, the global loss is known to be $L$-smooth, for some value of $L$ and locally strongly convex. Denoting $f : \rho \mapsto \text{LOSS}(\rho, \vec{\theta}^*(\rho, \vec{\mathcal{D}}), \vec{\mathcal{D}})$, we then have

$$\|\nabla f(\rho) - \nabla f(\rho')\|_2 \leq \left\| \nabla_\rho \text{LOSS}(\rho, \vec{\theta}^*(\rho, \vec{\mathcal{D}}), \vec{\mathcal{D}}) - \nabla_\rho \text{LOSS}(\rho', \vec{\theta}^*(\rho', \vec{\mathcal{D}}), \vec{\mathcal{D}}) \right\|_2 \tag{33}$$

$$\leq L \left\| (\rho, \vec{\theta}^*(\rho, \vec{\mathcal{D}})) - (\rho', \vec{\theta}^*(\rho', \vec{\mathcal{D}})) \right\|_2 \tag{34}$$

$$\leq L \|\rho - \rho'\|_2, \tag{35}$$

which proves that $f$ is $L$-smooth.

For strong convexity, note that since the global loss function is locally strongly convex, for any compact convex set $C$, there exists $\mu$ such that $\text{LOSS}(\rho, \vec{\theta}, \vec{\mathcal{D}})$ is $\mu$-strongly convex on $C = (C_1, C_2) \subset (\mathbb{R}^d, \mathbb{R}^{N \times d})$, therefore, by Lemma 10, $f(\rho)$ will also be $\mu$-strongly convex on $C_1$ which means that $f(\rho)$ is locally strongly convex. $\square$

# B   PROOF OF THE EQUIVALENCE

## B.1   PROOF OF THE REDUCTION FROM MODEL ATTACK TO DATA POISONING

*Proof of Lemma 1.* We omit making the dependence of the optima on $\vec{\mathcal{D}}$ explicit, and we consider any other models $\rho$ and $\vec{\theta}_{-s}$. We have the following inequalities:

$$\text{LOSS}_s(\rho^*, \vec{\theta}^*_{-s}, \theta^{\spadesuit}_s, \vec{\mathcal{D}}) = \text{LOSS}(\rho^*, \vec{\theta}^*, \vec{\mathcal{D}}) - \mathcal{L}(\theta^*_s, \mathcal{D}_s) \tag{36}$$

$$\leq \text{LOSS}(\rho, (\vec{\theta}_{-s}, \theta^*_s), \vec{\mathcal{D}}) - \mathcal{L}(\theta^*_s, \mathcal{D}_s) = \text{LOSS}_s(\rho, \vec{\theta}_{-s}, \theta^{\spadesuit}_s, \vec{\mathcal{D}}), \tag{37}$$

where we used the optimality of $(\rho^*, \vec{\theta}^*)$ in the second line, and where we repeatedly used the fact that $\theta_s^* = \theta_s^{\spadesuit}$. This proves that $(\rho^*, \vec{\theta}_{-s}^*)$ is a global minimum of the modified loss. $\qquad\square$

## B.2 Proof of the reduction from data poisoning to model attack

First, we define the following modified loss function:

$$\text{Loss}_s(\rho, \vec{\theta}_{-s}, \theta_s^{\spadesuit}, \vec{\mathcal{D}}_{-s}) \triangleq \text{Loss}(\rho, (\theta_s^{\spadesuit}, \vec{\theta}_{-s}), (\emptyset, \vec{\mathcal{D}}_{-s})) \tag{38}$$

where $\vec{\theta}_{-s}$ and $\vec{\mathcal{D}}_{-s}$ are variables and datasets for users $n \neq s$. We then define $\rho^*(\theta_s^{\spadesuit}, \vec{\mathcal{D}}_{-s})$ and $\vec{\theta}_{-s}^*(\theta_s^{\spadesuit}, \vec{\mathcal{D}}_{-s})$ as a minimum of the modified loss function, and $\theta_s^*(\theta_s^{\spadesuit}, \vec{\mathcal{D}}_{-s}) \triangleq \theta_s^{\spadesuit}$. We now prove a slightly more general version of Lemma 2, which applies to a larger class of regularizations. It also shows how to construct the strategic's user data poisoning attack.

**Lemma 14** (Reduction from data poisoning to model attack)**.** *Assume local PAC\* learning. Suppose also that $\mathcal{R}$ is continuous and that $\mathcal{R}(\rho, \theta) \to \infty$ when $\|\rho - \theta\|_2 \to \infty$. Consider any datasets $\mathcal{D}_{-s}$ and any attack model $\theta_s^{\spadesuit}$ such that the modified loss $\text{Loss}_s$ has a unique minimum $\rho^*(\theta_s^{\spadesuit}, \vec{\mathcal{D}}_{-s}), \vec{\theta}_{-s}^*(\theta_s^{\spadesuit}, \vec{\mathcal{D}}_{-s})$. Then, for any $\varepsilon, \delta > 0$, there exists $\mathcal{I}$ such that if user $s$'s dataset $\mathcal{D}_s$ contains at least $\mathcal{I}$ inputs drawn from model $\theta_s^{\spadesuit}$, then, with probability at least $1 - \delta$, we have*

$$\left\|\rho^*(\vec{\mathcal{D}}) - \rho^*(\theta_s^{\spadesuit}, \vec{\mathcal{D}}_{-s})\right\|_2 \leq \varepsilon \text{ and } \forall n \neq s, \ \left\|\theta_n^*(\vec{\mathcal{D}}) - \theta_n^*(\theta_s^{\spadesuit}, \vec{\mathcal{D}}_{-s})\right\|_2 \leq \varepsilon. \tag{39}$$

Clearly, $\ell_2^2$, $\ell_2$ and smooth-$\ell_2$ are continuous regularizations, and verify $\mathcal{R}(\rho, \theta) \to \infty$ when $\|\rho - \theta\|_2 \to \infty$. Moreover, setting $\delta \triangleq 1/2$ shows that the probability that the dataset $\mathcal{D}_s$ satisfies the inequalities of Lemma 14 is positive. This implies in particular that there must be a dataset $\mathcal{D}_s$ that satisfies these inequalities. ALl in all, this shows that Lemma 14 implies Lemma 2.

*Proof of Lemma 14.* Let $\varepsilon, \delta > 0$ and $\theta_s^{\spadesuit} \in \mathbb{R}^d$. Denote $\rho^{\spadesuit} \triangleq \rho^*(\theta_s^{\spadesuit}, \vec{\mathcal{D}}_{-s})$ and $\vec{\theta}^{\spadesuit} \triangleq \vec{\theta}^*(\theta_s^{\spadesuit}, \vec{\mathcal{D}}_{-s})$ the result of strategic user $s$'s model attack. We define the compact set $C$ by

$$C \triangleq \left\{\rho, \vec{\theta}_{-s} \ \middle| \ \|\rho - \rho^{\spadesuit}\|_2 \leq \varepsilon \wedge \forall n \neq s, \ \|\theta_n - \theta_n^{\spadesuit}\|_2 \leq \varepsilon\right\} \tag{40}$$

We define $D \triangleq \overline{\mathbb{R}^{d \times N} - C}$ the closure of the complement of $C$. Clearly, $\rho^{\spadesuit}, \vec{\theta}_{-s}^{\spadesuit} \notin D$. We aim to show that, when strategic user $s$ reveals a large dataset $\mathcal{D}_s$ whose answers are provided using the attack model $\theta_s^{\spadesuit}$, then the same holds for any global minimum of the global loss $\rho^*(\vec{\mathcal{D}}), \vec{\theta}_{-s}^*(\vec{\mathcal{D}}) \in C$. Note that, to prove this, it suffices to prove that the modified loss takes too large values, even when $\theta_s^{\spadesuit}$ is replaced by $\theta_s^*(\vec{\mathcal{D}})$.

Let us now formalize this. Denote $L^{\spadesuit} \triangleq \text{Loss}_s(\rho^{\spadesuit}, \vec{\theta}_{-s}^{\spadesuit}, \theta_s^{\spadesuit}, \vec{\mathcal{D}}_{-s})$. We define

$$\eta \triangleq \inf_{\rho, \vec{\theta}_{-s} \in D} \text{Loss}_s(\rho, \vec{\theta}_{-s}, \theta_s^{\spadesuit}, \vec{\mathcal{D}}_{-s}) - L^{\spadesuit}. \tag{41}$$

By a similar argument as that of Lemma 5, using the assumption $\mathcal{R} \to \infty$ at infinity, we know that the infimum is actually a minimum. Moreover, given that the minimum of the modified loss $\text{Loss}_s$ is unique, we know that the value of the loss function at this minimum is different from its value at $\rho^{\spadesuit}, \vec{\theta}_{-s}^{\spadesuit}$. As a result, we must have $\eta > 0$.

Now, since the function $\mathcal{R}$ is differentiable, it must be continuous. By the Heine–Cantor theorem, it is thus uniformly continuous on all compact sets. Thus, there must exist $\kappa > 0$ such that, for all models $\theta_s$ satisfying $\left\|\theta_s - \theta_s^{\spadesuit}\right\|_2 \leq \kappa$, we have

$$\left|\mathcal{R}(\theta_s, \rho^{\spadesuit}) - \mathcal{R}(\theta_s^{\spadesuit}, \rho^{\spadesuit})\right| \leq \eta/3. \tag{42}$$

Now, Lemma 5 guarantees the existence of $\mathcal{I}$ such that, if user $s$ provides a dataset $\mathcal{D}_s$ of least $\mathcal{I}$ answers with the model $\theta_s^{\spadesuit}$, then with probability at least $1 - \delta$, we will have $\left\|\theta_s^*(\vec{\mathcal{D}}) - \theta_s^{\spadesuit}\right\|_2 \leq \min(\kappa, \varepsilon)$. Under this event, we then have

$$\text{Loss}_s\left(\rho^{\spadesuit}, \vec{\theta}_{-s}^{\spadesuit}, \theta_s^*(\vec{\mathcal{D}}), \vec{\mathcal{D}}_{-s}\right) \leq L^{\spadesuit} + \eta/3. \tag{43}$$

Then

$$\inf_{\rho, \vec{\theta}_{-s} \in D} \text{Loss}_s(\rho, \vec{\theta}_{-s}, \theta_s^*(\vec{\mathcal{D}}), \vec{\mathcal{D}}_{-s}) \geq \inf_{\rho, \vec{\theta}_{-s} \in D} \text{Loss}_s(\rho, \vec{\theta}_{-s}, \theta_s^{\spadesuit}, \vec{\mathcal{D}}_{-s}) - \eta/3 \tag{44}$$

$$\geq L^{\spadesuit} + \eta - \eta/3 \geq L^{\spadesuit} + 2\eta/3 \tag{45}$$

$$> \text{Loss}_s\left(\rho^{\spadesuit}, \vec{\theta}_{-s}^{\spadesuit}, \theta_s^*(\vec{\mathcal{D}}), \vec{\mathcal{D}}_{-s}\right). \tag{46}$$

This shows that there is a high probability event under which the minimum of $\rho, \vec{\theta}_{-s} \mapsto \text{Loss}_s\left(\rho, \vec{\theta}_{-s}, \theta_s^*(\vec{\mathcal{D}}), \vec{\mathcal{D}}_{-s}\right)$ cannot be reached in $D$. This is equivalent to what the theorem we needed to prove states. $\qquad \square$

### B.3 Proof of reduction from model attack to gradient attack

In this section, we prove a slightly more general result than Lemma 3. Namely, instead of working with specific regularizations, we consider a more general class of regularizations, identified by Assumption 1.

**Lemma 15** (Reduction from model attack to gradient attack). *Under Assumption 1, if $g_s^t$ converges and if $\eta_t = \eta$ is a constant small enough, then $\rho^t$ will converge too. Denote $\rho^\infty$ its limit. Then for any $\varepsilon > 0$, there is $\theta_s^{\spadesuit} \in \mathbb{R}^d$ such that $\left\| \rho^\infty - \rho^*(\theta_s^{\spadesuit}, \vec{\mathcal{D}}_{-s}) \right\|_2 \leq \varepsilon$.*

Note that since $\ell_2^2$ and smooth-$\ell_2$ regularizations satisfy Assumption 1, Lemma 15 clearly implies Lemma 3. We now introduce the key objects of the proof of Lemma 15.

Denote $g_s^\infty$ the limit of the attack gradients $g_s^t$. We now define

$$\text{Loss}_s^1(\rho) \triangleq \inf_{\vec{\theta}_{-s}} \left\{ \text{Loss}(\rho, \vec{\theta}, \vec{\mathcal{D}}) - \mathcal{L}_s(\theta_s, \mathcal{D}_s) - \mathcal{R}(\rho, \theta_s) \right\} + \rho^T g_s^\infty \tag{47}$$

$$= \inf_{\vec{\theta}_{-s}} \left\{ \sum_{n \neq s} \mathcal{L}_n(\theta_n, \mathcal{D}_n) + \sum_{n \neq s} \mathcal{R}(\rho, \theta_n) \right\} + \rho^T g_s^\infty, \tag{48}$$

and prove that $\rho^t$ will converge to the minimizer of $\text{Loss}_s^1(\rho)$. By Lemma 13, we show that $\text{Loss}_s^1(\rho)$ is both locally strongly convex and $L$-smooth.

Now define $\zeta_s^t \triangleq g_s^t - g_s^\infty$. We then know $\zeta_s^t \to 0$ and $\nabla \text{Loss}_s^1(\rho^t)$ is the sum of all gradient vectors received from all users assuming the strategic user $s$ sends the vector $g_s^\infty$ in all iterations. Thus, at iteration $t$ of the optimization algorithm, we will take one step in the direction $G^t \triangleq \nabla \text{Loss}_s^1(\rho^t) + \zeta_s^t$, i.e.,

$$\rho^{t+1} = \rho^t - \eta_t G^t. \tag{49}$$

We now prove the following lemma that bounds the difference between the function value in two successive iterations.

**Lemma 16.** *If $\text{Loss}_s^1(\rho)$ is $L$-smooth and $\eta_t \leq 1/L$, we have*

$$\text{Loss}_s^1(\rho^{t+1}) - \text{Loss}_s^1(\rho^t) \leq -\frac{\eta_t}{2} \left\| G^t \right\|_2^2 + \eta_t \zeta_s^{t^T} G^t. \tag{50}$$

*Proof.* Since $\text{Loss}_s^1$ is $L$-smooth, we have

$$\text{Loss}_s^1(\rho^{t+1}) \leq \text{Loss}_s^1(\rho^t) + (\rho^{t+1} - \rho^t)^T \nabla \text{Loss}_s^1(\rho^t) + \frac{L}{2} \left\| \rho^{t+1} - \rho^t \right\|_2^2. \tag{51}$$

Now plugging $\rho^{t+1} - \rho^t = -\eta_t G^t$ and $\nabla \text{Loss}_s^1(\rho^t) = G^t - \zeta_s^t$ into the inequality implies

$$\text{Loss}_s^1(\rho^{t+1}) - \text{Loss}_s^1(\rho^t) \leq \left( -\eta_t G^t \right)^T \left( G^t - \zeta_s^t \right) + \frac{L}{2} \left\| -\eta_t G^t \right\|_2^2 \tag{52}$$

$$\leq -\frac{\eta_t}{2} \left\| G^t \right\|_2^2 + \eta_t \zeta_s^{t^T} G^t, \tag{53}$$

where we used the fact $\eta_t \leq 1/L$. $\qquad \square$

### B.3.1 THE GLOBAL MODEL REMAINS BOUNDED

**Lemma 17.** *There is $M$ such that, for all $t$, $\mathrm{Loss}_s^1(\rho^t) \leq M$.*

*Proof.* Consider the closed ball $\mathcal{B}(\rho^*, 1)$ centered on $\rho^*$ and of radius 1. By Lemma 13, we know that $\mathrm{Loss}_s^1$ is locally strongly convex and thus there exists a $\mu_1 > 0$ such that $\mathrm{Loss}_s^1$ is $\mu_1$-strongly convex on $\mathcal{B}(\rho^*, 1)$. Now consider a point $\rho_1$ on the boundary of $\mathcal{B}(\rho^*, 1)$. By strong convexity we have

$$\left\| \nabla \mathrm{Loss}_s^1(\rho_1) \right\|_2^2 \geq (\rho_1 - \rho^*)^T \nabla \mathrm{Loss}_s^1(\rho_1) \geq \mu_1 \left\| \rho_1 - \rho^* \right\|_2^2 = \mu_1. \tag{54}$$

Now similarly, by the convexity of $\mathrm{Loss}_s^1$ on $\mathbb{R}^d$, for any $\rho \in \mathbb{R}^d - \mathcal{B}(\rho^*, 1)$, we have $\left\| \nabla \mathrm{Loss}_s^1(\rho_1) \right\|_2 \geq \sqrt{\mu_1}$. Now since $\zeta_s^t \to 0$, there exists an iteration $T_1$ after which ($t \geq T_1$), we have $\|\zeta_s^t\|_2 \leq \frac{1}{4}\sqrt{\mu_1}$, and thus $\|G^t\|_2 \geq \left\| \nabla \mathrm{Loss}_s^1(\rho^t) \right\|_2 - \|\zeta_s^t\|_2 \geq \frac{3}{4}\sqrt{\mu_1}$. Thus, Lemma 16 implies that for $t \geq T_1$, if $\|\rho^t - \rho^*\|_2 \geq 1$, then

$$\mathrm{Loss}_s^1(\rho^{t+1}) - \mathrm{Loss}_s^1(\rho^t) \leq -\frac{\eta}{2} \left\| G^t \right\|_2^2 + \eta \zeta_s^{tT} G^t \tag{55}$$

$$\leq -\frac{\eta}{2} \left\| G^t \right\|_2^2 + \eta \left\| \zeta_s^t \right\|_2 \left\| G^t \right\|_2 \tag{56}$$

$$\leq -\frac{\eta}{2} \left\| G^t \right\|_2 \left( \left\| G^t \right\|_2 - 2 \left\| \zeta_s^t \right\|_2 \right) \tag{57}$$

$$\leq -\frac{\eta}{2}\frac{3}{4}\sqrt{\mu_1}\left(\frac{3}{4}\sqrt{\mu_1} - \frac{2}{4}\sqrt{\mu_1}\right) \leq -\frac{3\eta}{32}\mu_1 < 0. \tag{58}$$

Thus, for $\|\rho^t - \rho^*\|_2 \geq 1$, the loss cannot increase at the next iteration.

Now consider the case $\|\rho^t - \rho^*\|_2 < 1$ for $t \geq T_1$. The smoothness of $\mathrm{Loss}_s^1$ implies $\left\| \nabla \mathrm{Loss}_s^1(\rho^t) \right\|_2 < L$. Therefore,

$$\left\| \rho^{t+1} - \rho^* \right\|_2 = \left\| \rho^t - \eta(\nabla \mathrm{Loss}_s^1(\rho^t) + \zeta_s^t) - \rho^* \right\|_2 \tag{59}$$

$$\leq \left\| \rho^{t+1} - \rho^* \right\|_2 + \eta(L + \frac{1}{4}\sqrt{\mu_1}) \leq 1 + \eta(L + \frac{1}{4}\sqrt{\mu_1}). \tag{60}$$

Now we define $M_1 \triangleq \max_{\rho \in \mathcal{B}\left(\rho^*, 1+\eta(L+\frac{1}{4}\sqrt{\mu_1})\right)} \mathrm{Loss}_s^1(\rho)$, the maximum function value in the closed ball $\mathcal{B}\left(\rho^*, 1 + \eta(L + \frac{1}{4}\sqrt{\mu_1})\right)$. Therefore, we have $\mathrm{Loss}_s^1(\rho^{t+1}) \leq M_1$. So far we proved that for $t \geq T_1$, in each iteration of gradient descent either the function value will not increase or it will be upper-bounded by $M_1$. This implies that for all $t$, the function value $\mathrm{Loss}_s^1(\rho^t)$ is upper-bounded by

$$M \triangleq \max \left\{ \max_{t \leq T_1} \left\{ \mathrm{Loss}_s^1(\rho^t) \right\}, M_1 \right\}. \tag{61}$$

This concludes the proof. □

**Lemma 18.** *There is a compact set $X$ such that, for all $t$, $\rho^t \in X$.*

*Proof.* Now since $\mathrm{Loss}_s^1$ is $\mu_1$-strongly convex in $\mathcal{B}(\rho^*, 1)$, for any point $\rho \in \mathbb{R}^d$ such that $\|\rho - \rho^t\|_2 = 1$, we have

$$\mathrm{Loss}_s^1(\rho) \geq \mathrm{Loss}_s^1(\rho^*) + \frac{\mu_1}{2} \|\rho - \rho^*\|_2^2 = \mathrm{Loss}_s^1(\rho^*) + \frac{\mu_1}{2}. \tag{62}$$

But now by the convexity of $\mathrm{Loss}_s^1$ in $\mathbb{R}^d$, for any $\rho$ such that $\|\rho - \rho^*\|_2 \geq 1$, we have

$$\mathrm{Loss}_s^1(\rho) \geq \mathrm{Loss}_s^1(\rho^*) + \|\rho - \rho^*\|_2 \frac{\mu_1}{2}. \tag{63}$$

This implies that if $\|\rho^t - \rho^*\|_2 > \frac{2}{\mu_1}\left(M_2 - \mathrm{Loss}_s^1(\rho^*)\right)$, then $\mathrm{Loss}_s^1(\rho^t) > M_2$. Therefore, we must have $\|\rho^t - \rho^*\|_2 \leq \frac{2}{\mu_1}\left(M_2 - \mathrm{Loss}_s^1(\rho^*)\right)$, for all $t \geq 0$. This describes a closed ball, which is a compact set. □

B.3.2  CONVERGENCE OF THE GLOBAL MODEL UNDER CONVERGING GRADIENT ATTACK

**Lemma 19.** *Suppose $u_t \geq 0$ verifies $u_{t+1} \leq \alpha u_t + \delta_t$, with $\delta_t \to 0$. Then $u_t \to 0$.*

*Proof.* We now show that for any $\varepsilon > 0$, there exists an iteration $T(\varepsilon)$, such that for $t \geq T(\varepsilon)$, we have $u_t \leq \varepsilon$. For this, note that by induction, we observe that, for all $t \geq 0$,

$$u_{t+1} \leq u_0 \alpha^{t+1} + \sum_{\tau=0}^{t} \alpha^\tau \delta_{t-\tau}. \tag{64}$$

Since $\delta_t \to 0$, there exists an iteration $T_2(\varepsilon)$ such that for all $t \geq T_2(\varepsilon)$, we have $\delta_t \leq \frac{\varepsilon(1-\alpha)}{2}$. Therefore, for $t \geq T_2(\varepsilon)$, we have

$$u_{t+1} \leq u_0 \alpha^{t+1} + \sum_{\tau=0}^{t-T_2(\varepsilon)} \alpha^\tau \delta_{t-\tau} + \sum_{\tau=t-T_2(\varepsilon)+1}^{t} \alpha^\tau \delta_{t-\tau} \tag{65}$$

$$\leq u_0 \alpha^{t+1} + \frac{\varepsilon(1-\alpha)}{2} \sum_{\tau=0}^{t-T_2(\varepsilon)} \alpha^\tau + \sum_{s=0}^{T_2(\varepsilon)-1} \alpha^{t-s} \delta_s \tag{66}$$

$$\leq \left( u_0 + \sum_{s=0}^{T_2(\varepsilon)-1} \alpha^{-s-1} \delta_s \right) \alpha^{t+1} + \frac{\varepsilon(1-\alpha)}{2} \sum_{\tau=0}^{\infty} \alpha^\tau. \tag{67}$$

Denoting $M_0(\varepsilon) \triangleq \sum_{s=0}^{T_2(\varepsilon)-1} \alpha^{-s-1} \delta_s$, we then have

$$u_{t+1} \leq (u_0 + M_0(\varepsilon)) \alpha^{t+1} + \frac{\varepsilon}{2}. \tag{68}$$

Therefore, for $t \geq \frac{\ln \frac{\varepsilon}{2(u_0 + M_0(\varepsilon))}}{\ln \alpha}$, we have

$$u_{t+1} \leq \frac{\varepsilon}{2} + \frac{\varepsilon}{2} = \varepsilon. \tag{69}$$

This proves that $u_t \to 0$. $\qquad\square$

We first prove the first part of Lemma 3.

**Lemma 20.** *Under Assumption 1 and $\eta_t = \eta \leq 1/L$, if $g_s^t$ converges, then so does $\rho^t$.*

*Proof.* Define $X$ based on Lemma 18. Since $\text{Loss}_s^1$ is locally strongly convex, there exists $\mu_2 > 0$ such that $\text{Loss}_s^1$ is $\mu_2$-strongly convex in a convex compact set $X$ containing $\rho^t$ for all $t \geq 0$. By the strong convexity of $\text{Loss}_s^1(\rho)$, we have

$$\text{Loss}_s^1(\rho^t) - \text{Loss}_s^1(\rho^*) \leq (\rho^t - \rho^*)^T \nabla \text{Loss}_s^1(\rho^t) - \frac{\mu_2}{2} \left\| \rho^t - \rho^* \right\|_2^2 \tag{70}$$

$$= (\rho^t - \rho^*)^T \left( G^t - \zeta_s^t \right) - \frac{\mu_2}{2} \left\| \rho^t - \rho^* \right\|_2^2. \tag{71}$$

Now, using the fact

$$(\rho^t - \rho^*)^T G^t = \frac{1}{\eta} (\rho^t - \rho^*)^T (\rho^t - \rho^{t+1}) \tag{72}$$

$$= \frac{1}{2\eta} \left( \left\| \rho^t - \rho^* \right\|_2^2 + \left\| \rho^t - \rho^{t+1} \right\|_2^2 - \left\| \rho^{t+1} - \rho^* \right\|_2^2 \right) \tag{73}$$

$$= \frac{1}{2\eta} \left( \eta^2 \left\| G^t \right\|_2^2 + \left\| \rho^t - \rho^* \right\|_2^2 - \left\| \rho^{t+1} - \rho^* \right\|_2^2 \right) \tag{74}$$

$$= \frac{\eta}{2} \left\| G^t \right\|_2^2 + \frac{1}{2\eta} \left( \left\| \rho^t - \rho^* \right\|_2^2 - \left\| \rho^{t+1} - \rho^* \right\|_2^2 \right), \tag{75}$$

we have

$$\text{Loss}_s^1(\rho^t) - \text{Loss}_s^1(\rho^*) \leq \tag{76}$$

$$\frac{\eta}{2} \left\| G^t \right\|_2^2 + \frac{1}{2\eta} \left( \left\| \rho^t - \rho^* \right\|_2^2 - \left\| \rho^{t+1} - \rho^* \right\|_2^2 \right) - (\rho^t - \rho^*)^T \zeta_s^t - \frac{\mu_2}{2} \left\| \rho^t - \rho^* \right\|_2^2. \tag{77}$$

But now note that $\text{Loss}_s^1(\rho^t) - \text{Loss}_s^1(\rho^*) \geq \text{Loss}_s^1(\rho^t) - \text{Loss}_s^1(\rho^{t+1})$. Thus, combining Equation (77) and Lemma 16 yields

$$- \eta \zeta_s^{t^T} G^t \leq \frac{1}{2\eta} \left( \left\| \rho^t - \rho^* \right\|_2^2 - \left\| \rho^{t+1} - \rho^* \right\|_2^2 \right) - (\rho^t - \rho^*)^T \zeta_s^t - \frac{\mu_2}{2} \left\| \rho^t - \rho^* \right\|_2^2. \tag{78}$$

By rearranging the terms, we then have

$$\left\| \rho^{t+1} - \rho^* \right\|_2^2 \leq (1 - \mu_2 \eta) \left\| \rho^t - \rho^* \right\|_2^2 - \eta \left( \rho^{t+1} - \rho^* \right)^T \zeta_s^t \tag{79}$$

$$\leq (1 - \mu_2 \eta) \left\| \rho^t - \rho^* \right\|_2^2 + \eta \left\| \rho^{t+1} - \rho^* \right\|_2 \left\| \zeta_s^t \right\|_2. \tag{80}$$

Now note that $\eta \leq 1/L < 1/\mu_2$ and thus $0 < 1 - \mu_2 \eta < 1$. We now define two sequences $u_t \triangleq \left\| \rho^t - \rho^* \right\|_2$ and $\delta_t = \eta \left\| \zeta_s^t \right\|_2$. We already know that $\delta_t \to 0$, and we want to show $u_t$ also converges to 0. By Equation (80), we have

$$u_{t+1}^2 \leq (1 - \eta \mu_2) u_t^2 + \delta_t u_{t+1}, \tag{81}$$

which implies

$$\left( u_{t+1} - \frac{\delta_t}{2} \right)^2 = u_{t+1}^2 - u_{t+1} \delta_t + \frac{\delta_t^2}{4} \leq (1 - \eta \mu_2) u_t^2 + \frac{\delta_t^2}{4}, \tag{82}$$

and thus

$$u_{t+1} \leq \sqrt{(1 - \eta \mu_2) u_t^2 + \frac{\delta_t^2}{4}} + \frac{\delta_t}{2} \leq \sqrt{(1 - \eta \mu_2) u_t^2} + \frac{\delta_t}{2} + \frac{\delta_t}{2} \leq \left( 1 - \frac{\eta \mu_2}{2} \right) u_t + \delta_t. \tag{83}$$

Lemma 19 allows to conclude. $\qquad\square$

### B.3.3 REDUCTION FROM MODEL ATTACK TO CONVERGING GRADIENT ATTACK

*Proof of Lemma 15.* Lemma 20 already provided the convergence part of Lemma 3. We now move forward to proving the second part of the theorem, showing that for any $\varepsilon > 0$, there exists $\theta_s^\spadesuit \in \mathbb{R}^d$ such that $\left\| \rho^\infty - \rho^*(\theta_s^\spadesuit, \vec{\mathcal{D}}_{-s}) \right\|_2 \leq \varepsilon$. We define

$$\text{Loss}_s^2(\rho, \theta_s) \triangleq \inf_{\vec{\theta}_{-s}} \left\{ \text{Loss}(\rho, \vec{\theta}, \vec{\mathcal{D}}) - \mathcal{L}_s(\theta_s, \mathcal{D}_s) \right\} \tag{84}$$

$$= \text{Loss}_s^1(\rho) + \mathcal{R}(\rho, \theta_s) - \rho^T g_s^\infty, \tag{85}$$

and $\rho^*(\theta_s)$, its minimizer. Therefore, we have

$$\nabla_\rho \text{Loss}_s^2(\rho, \theta_s) = \nabla_\rho \text{Loss}_s^1(\rho) + \nabla_\rho \mathcal{R}(\rho, \theta_s) - g_s^\infty. \tag{86}$$

By Lemma 13, we know that $\text{Loss}_s^2$ is locally strongly convex. Therefore, there exists $\mu_3 > 0$ such that $\text{Loss}_s^2(\rho)$ is $\mu_3$-strongly convex in $\{\rho | \left\| \rho - \rho^*(\theta_s) \right\|_2 \leq 1\}$. Therefore, for any $0 < \varepsilon < 1$, if $\left\| \rho^\infty - \rho^*(\theta_s) \right\|_2 > \varepsilon$, we then have

$$\varepsilon \left\| \nabla_\rho \text{Loss}_s^2(\rho^\infty, \theta_s^\spadesuit) \right\|_2 \geq (\rho^\infty - \rho^*(\theta_s))^T \nabla_\rho \text{Loss}_s^2(\rho^\infty, \theta_s^\spadesuit) \tag{87}$$

$$\geq \mu_3 \left\| \rho^\infty - \rho^*(\theta_s) \right\|_2^2 = \mu_3 \geq \mu_3 \varepsilon^2, \tag{88}$$

and thus $\left\| \nabla_\rho \text{Loss}_s^2(\rho^\infty, \theta_s^\spadesuit) \right\|_2 \geq \mu_3 \varepsilon$.

Now since $g_s^\infty \in \text{GRAD}(\rho^\infty)$ there exists $\theta_s^\spadesuit \in \mathbb{R}^d$ such that[3] $\left\| \nabla_\rho \mathcal{R}(\rho^\infty, \theta_s^\spadesuit) - g_s^\infty \right\|_2 \leq \frac{\mu_3 \varepsilon}{2}$ which yields

$$\left\| \nabla_\rho \text{Loss}_s^2(\rho^\infty, \theta_s^\spadesuit) \right\|_2 = \left\| \nabla_\rho \text{Loss}_s^1(\rho^\infty) + \nabla_\rho \mathcal{R}(\rho^\infty, \theta_s^\spadesuit) - g_s^\infty \right\|_2 \tag{89}$$

$$= \left\| \nabla_\rho \mathcal{R}(\rho^\infty, \theta_s^\spadesuit) - g_s^\infty \right\|_2 \leq \frac{\mu_3 \varepsilon}{2}, \tag{90}$$

which is a contradiction. Therefore, we must have $\left\| \rho^\infty - \rho^*(\theta_s^\spadesuit, \vec{\mathcal{D}}_{-s}) \right\|_2 \leq \varepsilon$. $\qquad\square$

---

[3]In fact, if $g_s^\infty$ belongs to the interior of $\text{GRAD}(\rho^\infty)$, we can guarantee $\nabla_\rho \mathcal{R}(\rho^\infty, \theta_s^\spadesuit) = g_s^\infty$.

## C   Sum over expectations

In this section, we provide both theoretical and empirical results to argue for using a sum-based local loss over an expectation-based local loss.

### C.1   Theoretical arguments

Indeed, intuitively, if one considers an expectation $\mathbb{E}_{x \sim \mathcal{D}_n} [\ell(\theta_n, x)]$ rather than a sum, as is done by Hanzely et al. (2020), Dinh et al. (2020) and El-Mhamdi et al. (2021), then the weight of an honest active user's local loss will not increase as a node provides more and more data, which will hinder the ability of $\theta_n$ to fit the user's local data. In fact, intuitively, using an expectation wrongly yields the same influence to any two nodes, even when one (honest) node provides a much larger dataset $\mathcal{D}_n$ than the other, and should thus intuitively be regarded as "more reliable".

There is another theoretical argument for using the sum rather than the expectation. Namely, if the loss is regarded as a Bayesian negative log-posterior, given a prior $\exp \left( - \sum_{n \in [N]} \nu \|\theta_n\|_2 - \sum_{n \in [N]} \mathcal{R}(\rho, \theta_n) \right)$ on the local and global models, then the term that fits local data should equal the negative log-likelihood of the data, given the models $(\rho, \vec{\theta})$. Assuming that the distribution of each data point $x \in \mathcal{D}_n$ is independent from all other data points, and depends only on the local model $\theta_n$, this negative log-likelihood yields a sum over data points; not an expectation.

### C.2   Empirical results

We also empirically compared the performances of sum as opposed to the expectation. To do so, we constructed a setting where 10 "idle" users draw randomly 10 data points from the FashionMNIST dataset, while one "active" user has all of the FashionMNIST dataset (60,000 data points). We then learned local and global models, with $\mathcal{R}(\rho, \theta) \triangleq \lambda \|\rho - \theta\|_2^2$, $\lambda = 1$. We compared two different classifiers to which we refer as a "linear model" and "2-layers neural network", both using *CrossEntropy* loss. The linear model has $(784+1) \times 10$ parameters. The neural network has 2 layers of 784 parameters with bias, with *ReLU* activation in between, adding up to $((784 + 1) \times 784 + (784 + 1) \times 10$.

Note also that, in all our experiments, we did not consider any local regularization, i.e. we set $\nu \triangleq 0$. All our experiments are seeded with seed 999.

#### C.2.1   Noisy FashionMNIST

To see a strong difference between sum and average, we made the FashionMNIST dataset harder to learn, by randomly labeling 60% of the training set. Table 1 reports the accuracy of local and global models in the different settings. Our results clearly and robustly indicate that the use of sums outperforms the use of expectations.

|  | $\mathbb{E}L$ | $\Sigma L$ | $\mathbb{E}NN$ | $\Sigma NN$ |
|---|---|---|---|---|
| idle user's model | 0.52 | 0.80 | 0.55 | 0.79 |
| active user's model | 0.58 | 0.80 | 0.56 | 0.79 |
| global model | 0.55 | 0.80 | 0.58 | 0.79 |

Table 1: Accuracy of trained models, depending on the use of expectation (denoted $\mathbb{E}$) or sum ($\Sigma$), and on the use of linear classifier ($L$) or a 2-layer neural net ($NN$). Here, all users are honest and an $\ell_2^2$ regularization is used, but there is a large heterogeneity in the amount of data per user.

On each of the following plots, we display the top-1 accuracy on the MNIST test dataset (10 000 images) for the active user, for the global model and for one of the idle users (in Table 1, the mean accuracy for idle users is reported), as we vary the value of $\lambda$. Intuitively, $\lambda$ models how much we want the local models to be similar.

In the case of learning FashionMNIST, given that the data is i.i.d., larger values of $\lambda$ are more meaningful (though our experiments show that they may hinder convergence speed). However, in settings where users have different data distributions, e.g. because the labels depend on users' preferences, then smaller values of $\lambda$ may be more relevant.

Note that the use of a common value of $\lambda$ in both cases is slightly misleading, as using the sum intuitively decreases the comparative weight of the regularization term. To reduce this effect, for this experiment only, we divide the local losses by the average of the number of data points per node for the sum version. This way, if the number of points is equal for all nodes, the two losses will be exactly the same. What's more, our experiments seem to robustly show that using the sum consistently outperforms the expectation, for both a linear classifier and a 2-layer neural network, for the problem of noisy FashionMNIST classification.

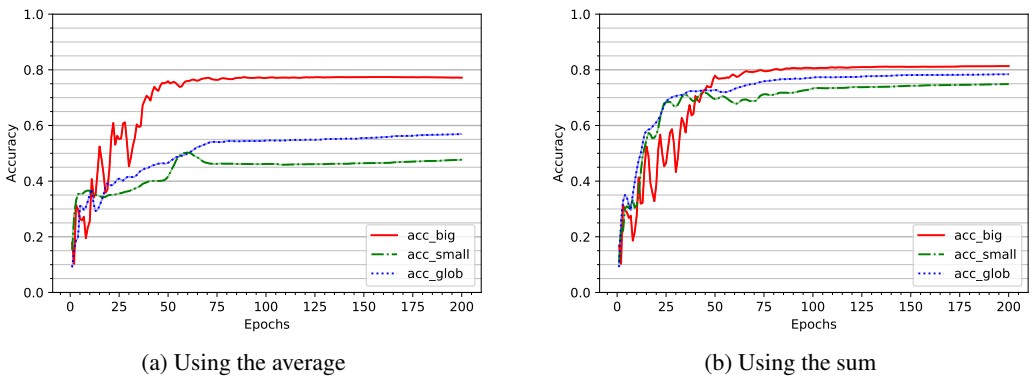

(a) Using the average          (b) Using the sum

Figure 4: Linear model on noisy FashionMNIST, for $\lambda = 0.01$.

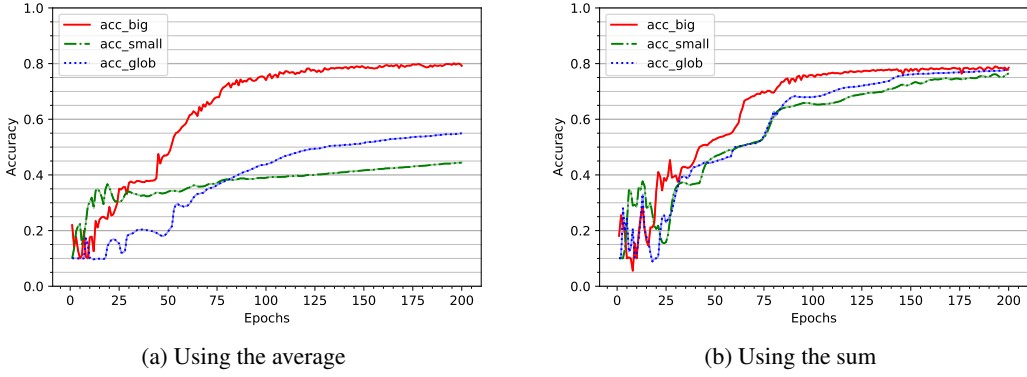

(a) Using the average          (b) Using the sum

Figure 5: 2-layer neural network on noisy FashionMNIST, for $\lambda = 0.01$.

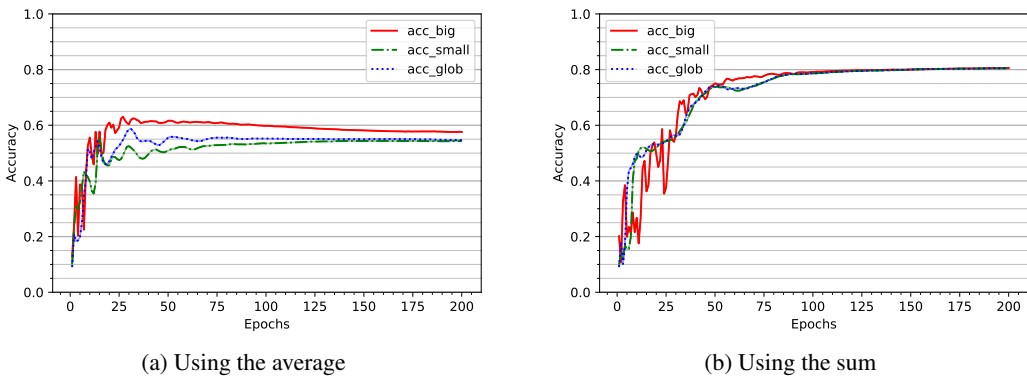

(a) Using the average

(b) Using the sum

Figure 6: Linear model on noisy FashionMNIST, for $\lambda = 0.1$.

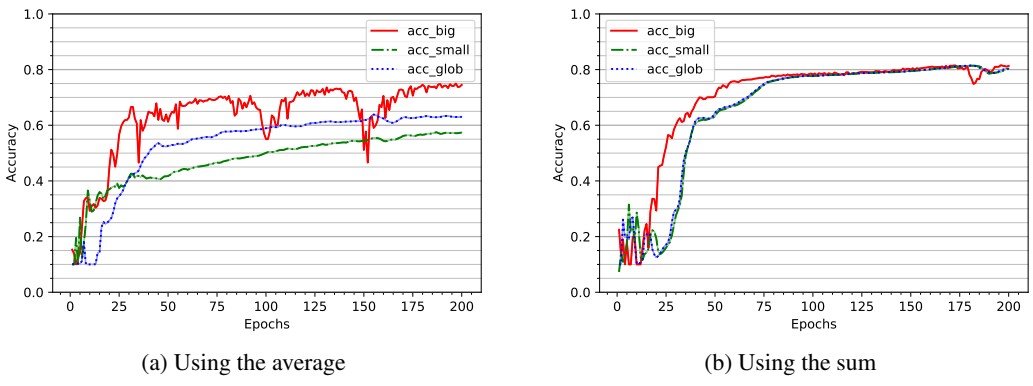

(a) Using the average

(b) Using the sum

Figure 7: 2-layer neural network on noisy FashionMNIST, for $\lambda = 0.1$.

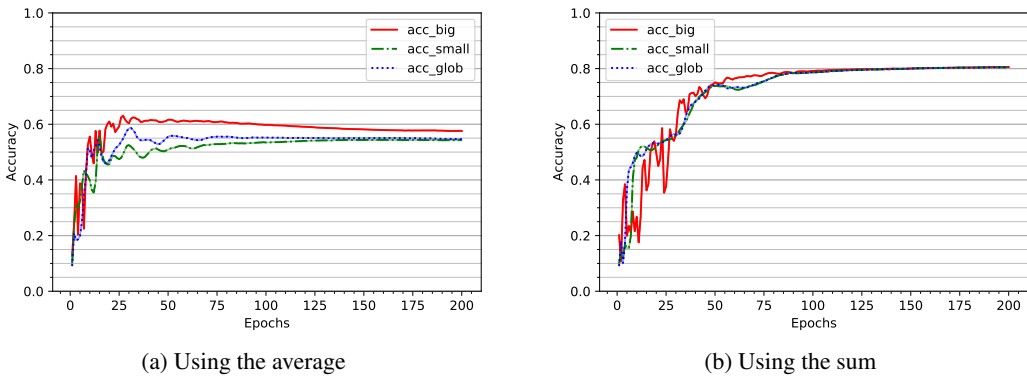

(a) Using the average

(b) Using the sum

Figure 8: Linear model on noisy FashionMNIST, for $\lambda = 1$.

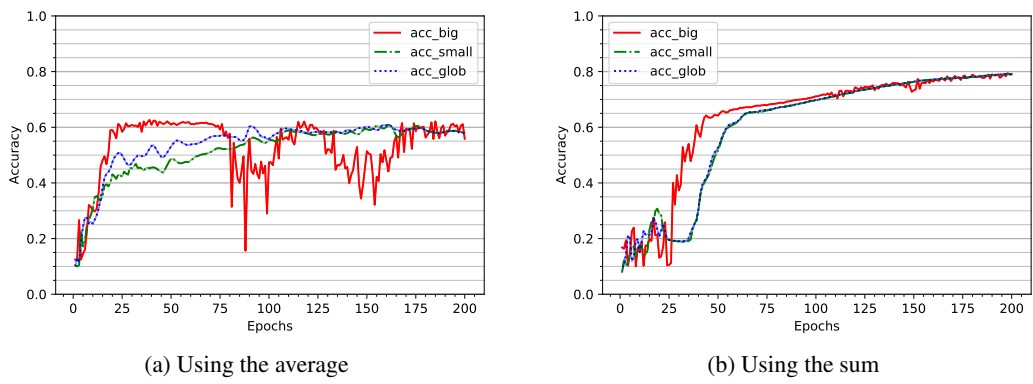

(a) Using the average

(b) Using the sum

Figure 9: 2-layer neural network on noisy FashionMNIST, for $\lambda = 1$.

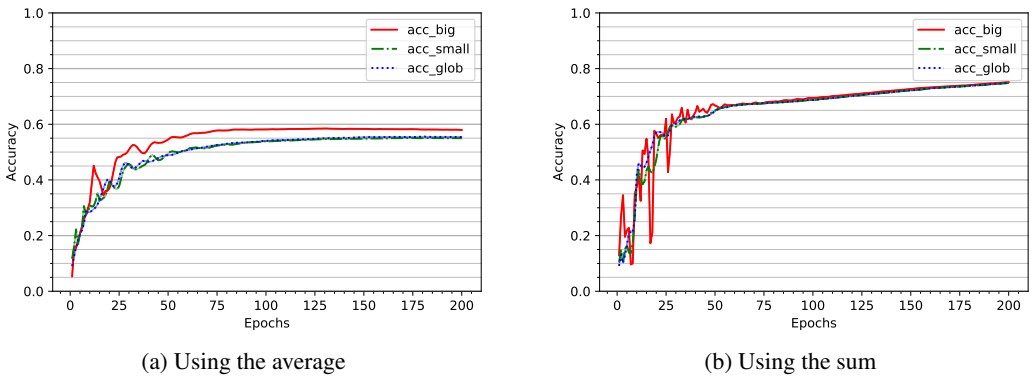

(a) Using the average

(b) Using the sum

Figure 10: Linear model on noisy FashionMNIST, for $\lambda = 10$.

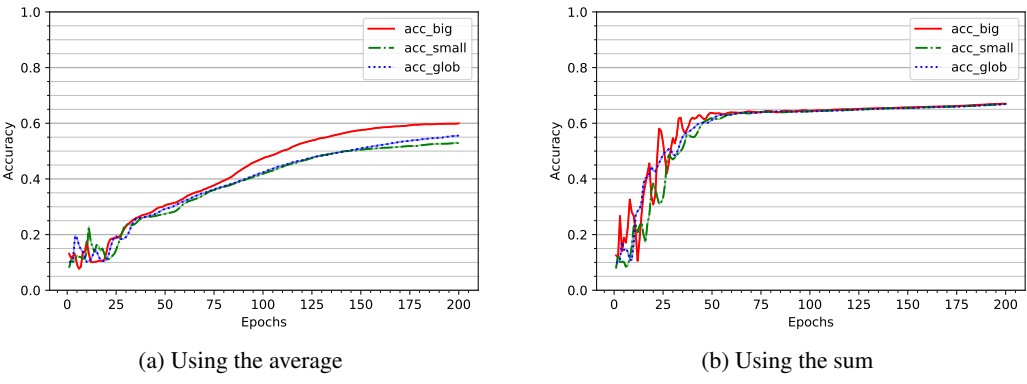

(a) Using the average

(b) Using the sum

Figure 11: 2-layer neural network on noisy FashionMNIST, for $\lambda = 10$.

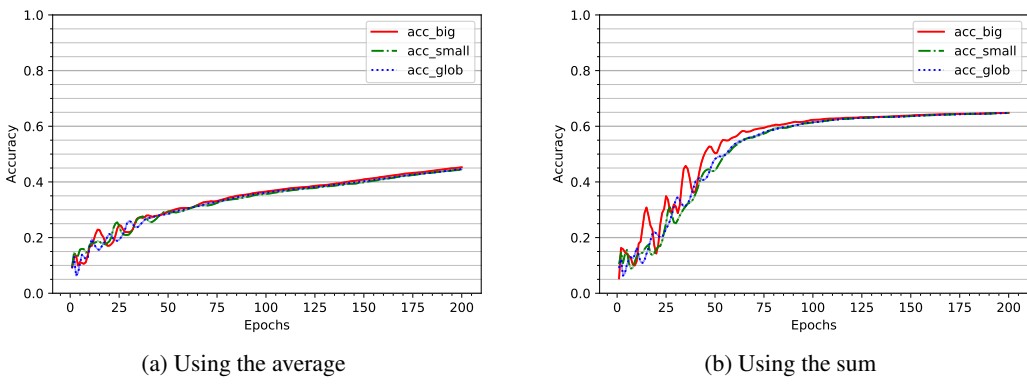

(a) Using the average          (b) Using the sum

Figure 12: Linear model on noisy FashionMNIST, for $\lambda = 100$.

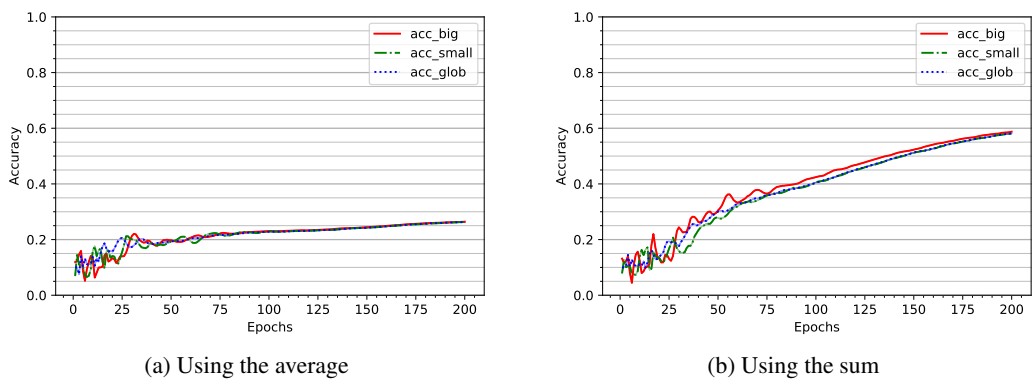

(a) Using the average          (b) Using the sum

Figure 13: 2-layer neural network on noisy FashionMNIST, for $\lambda = 100$.

### C.2.2 FASHIONMNIST WITHOUT NOISE

Recall that we introduced noise into FashionMNIST to make the problem harder to learn and observe a clear difference between the average and the sum. In this section, we present results of our experiments when the noise is removed.

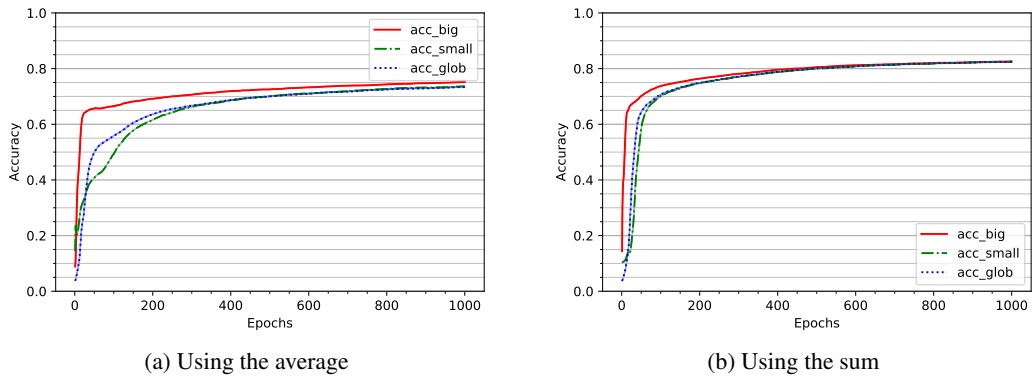

(a) Using the average          (b) Using the sum

Figure 14: Linear model on FashionMNIST (without noise), for $\lambda = 1$.

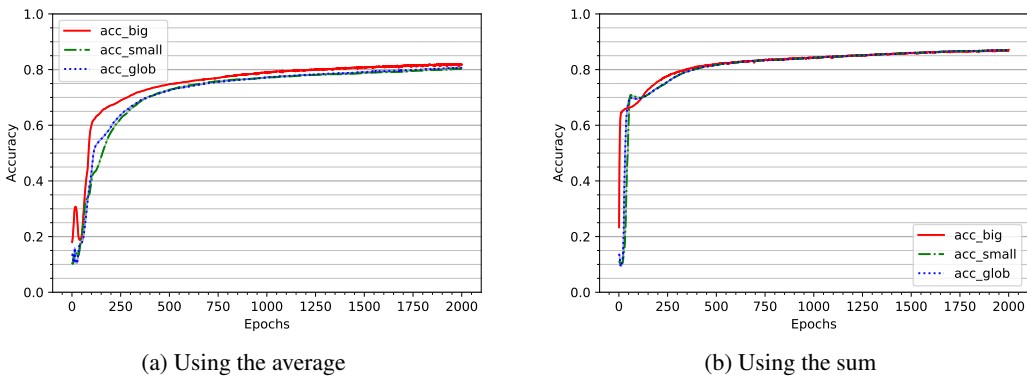

(a) Using the average

(b) Using the sum

Figure 15: 2-layer neural network on FashionMNIST (without noise), for $\lambda = 1$.

Even without noise, the difference between using the sum and using the expectation still seems important. We acknowledge, however, that the plots suggest that even though we ran this experiment for 10 times more (and 5 times more for the linear model) than other experiments, we might not have reached convergence yet, and that the use of the expectation might still eventually gets closer to the case of sum. We believe that the fact that the difference between sum and expectation in the absence of noise is weak is due to the fact that the FashionMNIST dataset is sufficiently linearly separable. Thus, we achieve a near-zero loss in both cases, which make the sum and the expectation close at optimum.

Even in this case, however, we observed that the sum clearly outperforms the expectation especially, in the first epochs. We argue that the reason for this is the following. By taking the average in local losses, the weights of the data of idle nodes are essentially blown out of proportion. As a result, the optimizer will very quickly fit these data. However, the signal from the data of the active node will then be too weak, so that the optimizer has to first almost perfectly fit the idle nodes' data before it can catch the signal of the active node's data and hence the average achieves weaker convergence performances than the sum.

## D  LINEAR REGRESSION AND CLASSIFICATION ARE GRADIENT PAC*

Throughout this section, we use the following terminology.

**Definition 5.** *Consider a parameterized event $\mathcal{E}(\mathcal{I})$. We say that the event $\mathcal{E}$ occurs with high probability if $\mathbb{P}\left[\mathcal{E}(\mathcal{I})\right] \to 1$ as $\mathcal{I} \to \infty$.*

### D.1  PRELIMINARIES

Define $\|\Sigma\|_2 \triangleq \max_{\|x\|_2 \neq 0}(\|\Sigma x\|_2 / \|x\|_2)$ the $\ell_2$ operator norm of the matrix $\Sigma$. For symmetric matrices $\Sigma$, this is also the largest eigenvalue in absolute value.

**Theorem 6** (Covariance concentration, Theorem 6.5 in Wainwright (2019)). *Denote $\Sigma = \mathbb{E}\left[\mathcal{Q}_i \mathcal{Q}_i^T\right]$, where $\mathcal{Q}_i \in \mathbb{R}^d$ is from a $\sigma_{\mathcal{Q}}$-sub-Gaussian random distribution $\tilde{\mathcal{Q}}$. Then, there are universal constants $c_1$, $c_2$ and $c_3$ such that, for any set $\{\mathcal{Q}_i\}_{i \in [\mathcal{I}]}$ of i.i.d. samples from $\tilde{\mathcal{Q}}$, and any $\delta > 0$, the sample covariance $\widehat{\Sigma} = \frac{1}{\mathcal{I}} \sum \mathcal{Q}_i \mathcal{Q}_i^T$ satisfies the bound*

$$\mathbb{P}\left[\frac{1}{\sigma_{\mathcal{Q}}^2} \left\|\hat{\Sigma} - \Sigma\right\|_2 \geq c_1 \left(\sqrt{\frac{d}{\mathcal{I}}} + \frac{d}{\mathcal{I}}\right) + \delta\right] \leq c_2 \exp\left(-c_3 \mathcal{I} \min(\delta, \delta^2)\right). \tag{91}$$

**Theorem 7** (Weyl's Theorem, Theorem 4.3.1 in Horn & Johnson (2012)). *Let $A$ and $B$ be Hermitian[4] and let the respective eigenvalues of $A$ and $B$ and $A + B$ be $\{\lambda_i(A)\}_{i=1}^d$, $\{\lambda_i(B)\}_{i=1}^d$, and*

---

[4]For real matrices, Hermitian is the same as symmetric.

$\{\lambda_i(A+B)\}_{i=1}^d$, *each increasingly ordered. Then*

$$\lambda_i(A+B) \leq \lambda_{i+j}(A) + \lambda_{d-j}(B), \quad j = 0, 1, ..., d-i, \tag{92}$$

*and*

$$\lambda_{i+j}(A) + \lambda_{j+1}(B) \leq \lambda_i(A+B), \quad j = 0, ..., i-1, \tag{93}$$

*for each $i = 1, ..., d$.*

**Lemma 21.** *Consider two symmetric definite positive matrices $S$ and $\Sigma$. Denote $\rho_{min}$ and $\lambda_{min}$ their minimal eigenvalues. Then $|\rho_{min} - \lambda_{min}| \leq \|S - \Sigma\|_2$.*

*Proof.* This is a direct consequence of Theorem 7, for $A = S$, $B = \Sigma - S$, $i = 1$, and $j = 0$. $\square$

**Corollary 1.** *There are universal constants $c_1$, $c_2$ and $c_3$ such that, for any $\sigma_{\mathcal{Q}}$-sub-Gaussian vector distribution $\tilde{\mathcal{Q}} \in \mathbb{R}^d$ and any $\delta > 0$, the sample covariance $\widehat{\Sigma} = \frac{1}{\mathcal{I}} \sum \mathcal{Q}_i \mathcal{Q}_i^T$ satisfies the bound*

$$\mathbb{P}\left[\frac{1}{\sigma_{\mathcal{Q}}^2}\left|\min \mathrm{SP}(\hat{\Sigma}) - \min \mathrm{SP}(\Sigma)\right| \geq c_1\left(\sqrt{\frac{d}{\mathcal{I}}} + \frac{d}{\mathcal{I}}\right) + \delta\right] \leq c_2 \exp\left(-c_3 \mathcal{I} \min(\delta, \delta^2)\right), \tag{94}$$

*where $\min \mathrm{SP}(\hat{\Sigma})$ and $\min \mathrm{SP}(\Sigma)$ are the minimal eigenvalues of $\hat{\Sigma}$ and $\Sigma$.*

*Proof.* This follows from Theorem 6 and Lemma 21. $\square$

**Lemma 22.** *With high probability, $\min \mathrm{SP}(\hat{\Sigma}) \geq \min \mathrm{SP}(\Sigma)/2$.*

*Proof.* Denote $\lambda_{min} \triangleq \min \mathrm{SP}(\Sigma)$ and $\widehat{\lambda}_{min} \triangleq \min \mathrm{SP}(\hat{\Sigma})$. Since each $\mathcal{Q}_i$ is drawn i.i.d. from a $\sigma_{\mathcal{Q}}$-sub-Gaussian, we can apply Corollary 1. Namely, there are constants $c_1$, $c_2$ and $c_3$, such that for any $\delta > 0$, we have

$$\mathbb{P}\left[\left|\widehat{\lambda}_{min} - \lambda_{min}\right| \geq c_1 \sigma_{\mathcal{Q}}^2\left(\sqrt{\frac{d}{\mathcal{I}}} + \frac{d}{\mathcal{I}}\right) + \delta \sigma_{\mathcal{Q}}^2\right] \leq c_2 \exp\left(-c_3 \mathcal{I} \min\left\{\delta, \delta^2\right\}\right). \tag{95}$$

We now set $\delta \triangleq \lambda_{min}/(4\sigma_{\mathcal{Q}}^2)$ and we consider $\mathcal{I}$ large enough so that $c_1\left(\sqrt{\frac{d}{\mathcal{I}}} + \frac{d}{\mathcal{I}}\right) \leq \lambda_{min}/(4\sigma_{\mathcal{Q}}^2)$. With high probability, we then have $\widehat{\lambda}_{min} \geq \lambda_{min}/2$. $\square$

## D.2 LINEAR REGRESSION IS GRADIENT-PAC*

In this section, we prove the first part of Lemma 4. Namely, we prove that linear regression is gradient-PAC* learning.

### D.2.1 LEMMAS FOR LINEAR REGRESSION

Before moving to the main proof that linear regression is gradient-PAC*, we first prove a few useful lemmas. These lemmas will rest on the following well-known theorems.

**Theorem 8** (Lemma 2.7.7 in Vershynin (2018)). *If $X$ and $Y$ are sub-Gaussian, then $XY$ is sub-exponential.*

**Theorem 9** (Equation 2.18 in Wainwright (2019)). *If $X_1, \ldots, X_{\mathcal{I}}$ are iid sub-exponential variables, then there exist constants $c_4$, $c_5$ such that, for all $\mathcal{I}$, we have*

$$\forall t \in [0, c_4], \ \mathbb{P}\left[|X - \mathbb{E}[X]| \geq t\mathcal{I}\right] \leq 2\exp\left(-c_5 \mathcal{I} t^2\right). \tag{96}$$

**Lemma 23.** *For all $j \in [d]$, the random variables $X_i \triangleq \xi_i \mathcal{Q}_i[j]$ are iid, sub-exponential and have zero mean.*

*Proof.* The fact that these variables are iid follows straightforwardly from the fact that the noises $\xi_i$ are iid, and the queries $\mathcal{Q}_i$ are also iid. Moreover, both are sub-Gaussian, and by Theorem 8, the product of sub-Gaussian variables is sub-exponential. Finally, we have $\mathbb{E}[X] = \mathbb{E}[\xi \mathcal{Q}[j]] = \mathbb{E}[\xi]\mathbb{E}[\mathcal{Q}[j]] = 0$, using the independence of the noise and the query, and the fact that noises have zero mean ($\mathbb{E}[\xi] = 0$). $\square$

**Lemma 24.** *There exists $B$ such that $\left\|\sum_{i\in\mathcal{I}}\xi_i\mathcal{Q}_i\right\|_2 \leq B\mathcal{I}^{3/4}$ with high probability.*

*Proof.* By Lemma 23, the terms $\xi_i\mathcal{Q}_i[j]$ are iid, sub-exponential and have zero mean. Therefore, by Theorem 9, there exist constants $c_4$ and $c_5$ such that for any coordinate $j \in [d]$ of $\xi_i\mathcal{Q}_i$ and for all $0 \leq u \leq c_4$, we have

$$\mathbb{P}\left[\left|\sum_{i\in\mathcal{I}}\xi_i\mathcal{Q}_i[j]\right| \geq \mathcal{I}u\right] \leq 2\exp\left(-c_5\mathcal{I}u^2\right). \tag{97}$$

Plugging $u = v\mathcal{I}^{(-1/4)}$ into the inequality for some small enough constant $v$, and using union bound then yields

$$\mathbb{P}\left[\left\|\sum_{i\in\mathcal{I}}\xi_i\mathcal{Q}_i\right\|_2 \geq \mathcal{I}^{(3/4)}v\sqrt{d}\right] \leq \mathbb{P}\left[\left\|\sum_{i\in\mathcal{I}}\xi_i\mathcal{Q}_i\right\|_\infty \geq \mathcal{I}^{(3/4)}v\right] \leq 2d\exp\left(-c_5\sqrt{\mathcal{I}}v^2\right). \tag{98}$$

Defining $B \triangleq v\sqrt{d}$ yields the lemma. $\qquad\square$

### D.2.2 PROOF THAT LINEAR REGRESSION IS GRADIENT-PAC*

We now move on to proving that least square linear regression is gradient-PAC*.

*Proof of Theorem 2.* Note that $\nabla_\theta\ell(\theta, \mathcal{Q}, \mathcal{A}) = (\theta^T\mathcal{Q} - \mathcal{A})\mathcal{Q}$. Thus, on input $i \in [\mathcal{I}]$, we have

$$\nabla_\theta\ell(\theta, \mathcal{Q}_i, \mathcal{A}(\mathcal{Q}_i, \theta^\dagger)) = \left((\theta - \theta^\dagger)^T\mathcal{Q}_i\right)\mathcal{Q}_i - \xi_i\mathcal{Q}_i. \tag{99}$$

Moreover, we have

$$(\theta - \theta^\dagger)^T\nabla_\theta\left(\nu\|\theta\|_2^2\right) = 2\nu(\theta - \theta^\dagger)^T\theta = 2\nu\|\theta - \theta^\dagger\|_2^2 + 2\nu(\theta - \theta^\dagger)^T\theta^\dagger. \tag{100}$$

As a result, we have

$$(\theta - \theta^\dagger)^T\nabla_\theta\mathcal{L}(\theta, \mathcal{D}) = \tag{101}$$

$$\mathcal{I}(\theta - \theta^\dagger)^T\widehat{\Sigma}(\theta - \theta^\dagger) - (\theta - \theta^\dagger)^T\left(\sum_{i\in\mathcal{I}}\xi_i\mathcal{Q}_i\right) + 2\nu\|\theta - \theta^\dagger\|_2^2 + 2\nu(\theta - \theta^\dagger)^T\theta^\dagger. \tag{102}$$

But now, with high probability, we have $(\theta-\theta^\dagger)^T\widehat{\Sigma}(\theta-\theta^\dagger) \geq (\lambda_{min}/2)\|\theta - \theta^\dagger\|_2^2$ (Lemma 22) and $\left\|\sum_{i\in\mathcal{I}}\xi_i\mathcal{Q}_i\right\|_2 \leq B\mathcal{I}^{(3/4)}$ (Lemma 24). Using the fact that $\|\theta^\dagger\|_2 \leq \mathcal{K}$ and the Cauchy-Schwarz inequality, we have

$$(\theta - \theta^\dagger)^T\nabla_\theta\mathcal{L}(\theta, \mathcal{D}) \geq (\frac{\lambda_{min}}{2}\mathcal{I} + \nu)\|\theta - \theta^\dagger\|_2^2 - (B\mathcal{I}^{(3/4)} + 2\nu\mathcal{K})\|\theta - \theta^\dagger\|_2. \tag{103}$$

Denoting $A_\mathcal{K} \triangleq \frac{\lambda_{min}}{2}$ and $B_\mathcal{K} \triangleq B + 2\nu\mathcal{K}$ and using the fact that $\mathcal{I} \geq 1$, we then have

$$(\theta - \theta^\dagger)^T\nabla_\theta\mathcal{L}(\theta, \mathcal{D}) \geq A_\mathcal{K}\mathcal{I}\|\theta - \theta^\dagger\|_2^2 - B_\mathcal{K}\mathcal{I}^{(3/4)}\|\theta - \theta^\dagger\|_2 \tag{104}$$

$$\geq A_\mathcal{K}\mathcal{I}\min\left\{\|\theta - \theta^\dagger\|_2, \|\theta - \theta^\dagger\|_2^2\right\} - B_\mathcal{K}\mathcal{I}^{(3/4)}\|\theta - \theta^\dagger\|_2, \tag{105}$$

with high probability. This corresponds to saying Assumption 2 is satisfied for $\alpha = 3/4$. $\qquad\square$

### D.3 LOGISTIC REGRESSION

In this section, we now prove the second part of Lemma 4. Namely, we prove that logistic regression is gradient-PAC* learning.

### D.3.1 Lemmas about the sigmoid function

We first prove two useful lemmas about the following logistic distance function.

**Definition 6.** *We define the logistic distance function by* $\Delta(a, b) \triangleq (a - b)(\sigma(a) - \sigma(b))$.

**Lemma 25.** *If* $a, b \in \mathbb{R}$ *such that for some* $k > 0$, $|a| \leq k$ *and* $|b| \leq k$, *then there exists some constant* $c_k > 0$ *such that*

$$\Delta(a, b) \geq c_k |a - b|^2. \tag{106}$$

*Proof.* Note that the derivative of $\sigma(z)$ is strictly positive, symmetric ($\sigma'(z) = \sigma'(-z)$) and monotonically decreasing for $z \geq 0$. Therefore, for any $z \in [-k, k]$, we know $\sigma'(z) \geq c_k \triangleq \sigma'(k)$. Thus, by the mean value theorem, we have

$$\frac{\sigma(a) - \sigma(b)}{a - b} \geq c_k. \tag{107}$$

Multiplying both sides by $(a - b)^2$ then yields the lemma. $\square$

**Lemma 26.** *If* $b \in \mathbb{R}$, *and* $|b| \leq k$, *for some* $k > 0$, *then there exists a constant* $d_k$, *such that for any* $a \in \mathbb{R}$, *we have*

$$\Delta(a, b) \geq d_k |a - b| - d_k \tag{108}$$

*Proof.* Assume $|a - b| \geq 1$ and define $d_k \triangleq \sigma(k+1) - \sigma(k)$. If $b \geq 0$, since $\sigma'(z)$ is decreasing for $z \geq 0$, we have $\sigma(b) - \sigma(b-1) \geq \sigma(b+1) - \sigma(b) \geq d_k$, and by symmetry, a similar argument holds for $b \leq 0$. Thus, we have

$$|\sigma(a) - \sigma(b)| \geq \min \{\sigma(b) - \sigma(b-1), \sigma(b+1) - \sigma(b)\} \geq d_k. \tag{109}$$

Therefore,

$$(a - b)(\sigma(a) - \sigma(b)) \geq d_k |a - b| \geq d_k |a - b| - d_k. \tag{110}$$

For the case of $|a - b| \leq 1$, we also have $(a - b)(\sigma(a) - \sigma(b)) \geq 0 \geq d_k |a - b| - d_k$. $\square$

### D.3.2 A uniform lower bound

**Definition 7.** *Denote* $\mathbb{S}^{d-1} \triangleq \{\mathbf{u} \in \mathbb{R}^d \mid \|\mathbf{u}\|_2 = 1\}$ *the hypersphere in* $\mathbb{R}^d$.

**Lemma 27.** *Assume* $\text{Supp}(\tilde{\mathcal{Q}})$ *spans* $\mathbb{R}^d$. *Then, for all* $\mathbf{u} \in \mathbb{S}^{d-1}$, $\mathbb{E}\left[|\mathcal{Q}^T\mathbf{u}|\right] > 0$.

*Proof.* Let $\mathbf{u} \in \mathbb{S}^{d-1}$. We know that there exists $\mathcal{Q}_1, \ldots, \mathcal{Q}_d \in \text{Supp}(\tilde{\mathcal{Q}})$ and $\alpha_1, \ldots, \alpha_d \in \mathbb{R}$ such that $\mathbf{u}$ is colinear with $\sum \alpha_j \mathcal{Q}_j$. In particular, we then have $\mathbf{u}^T \sum \alpha_j \mathcal{Q}_j = \sum \alpha_j (\mathcal{Q}_j^T \mathbf{u}) \neq 0$. Therefore, there must be a query $\mathcal{Q}_* \in \text{Supp}(\tilde{\mathcal{Q}})$ such that $\mathcal{Q}_*^T \mathbf{u} \neq 0$, which implies $a \triangleq |\mathcal{Q}_*^T \mathbf{u}| > 0$ By continuity of the scalar product, there must then also exist $\varepsilon > 0$ such that, for any $\mathcal{Q} \in \mathcal{B}(\mathcal{Q}_*, \varepsilon)$, we have $|\mathcal{Q}^T \mathbf{u}| \geq a/2$, where $\mathcal{B}(\mathcal{Q}_*, \varepsilon)$ is an Euclidean ball centered on $\mathcal{Q}_*$ and of radius $\varepsilon$.

But now, by definition of the support, we know that $p \triangleq \mathbb{P}[\mathcal{Q} \in \mathcal{B}(\mathcal{Q}_*, \varepsilon)] > 0$. By the law of total expectation, we then have

$$\mathbb{E}\left[|\mathcal{Q}^T\mathbf{u}|\right] = \mathbb{E}\left[|\mathcal{Q}^T\mathbf{u}| \mid \mathcal{Q} \in \mathcal{B}(\mathcal{Q}_*, \varepsilon)\right] \mathbb{P}[\mathcal{Q} \in \mathcal{B}(\mathcal{Q}_*, \varepsilon)]$$
$$+ \mathbb{E}\left[|\mathcal{Q}^T\mathbf{u}| \mid \mathcal{Q} \notin \mathcal{B}(\mathcal{Q}_*, \varepsilon)\right] \mathbb{P}[\mathcal{Q} \notin \mathcal{B}(\mathcal{Q}_*, \varepsilon)] \tag{111}$$
$$\geq ap/2 + 0 > 0, \tag{112}$$

which is the lemma. $\square$

**Lemma 28.** *Assume that, for all unit vectors* $\mathbf{u} \in \mathbb{S}^{d-1}$, *we have* $\mathbb{E}\left[|\mathcal{Q}^T\mathbf{u}|\right] > 0$, *and that* $\text{Supp}(\tilde{\mathcal{Q}})$ *is bounded by* $M_{\mathcal{Q}}$. *Then there exists* $C > 0$ *such that, with high probability,*

$$\forall \mathbf{u} \in \mathbb{S}^{d-1}, \quad \sum_{i \in \mathcal{I}} |\mathcal{Q}_i^T \mathbf{u}| \geq C\mathcal{I}. \tag{113}$$

*Proof.* By continuity of the scalar product and the expectation operator, and by compactness of $\mathbb{S}^{d-1}$, we know that

$$C_0 \triangleq \inf_{\mathbf{u} \in \mathbb{R}^d} \mathbb{E}\left[\left|\mathcal{Q}^T \mathbf{u}\right|\right] > 0. \tag{114}$$

Now define $\varepsilon \triangleq C_0/4M_{\mathcal{Q}}$. Note that $\mathbb{S}^{d-1} \subset \bigcup_{\mathbf{u} \in \mathbb{S}^{d-1}} \mathcal{B}(\mathbf{u}, \varepsilon)$. Thus we have a covering of the hypersphere by open sets. But since $\mathbb{S}^{d-1}$ is compact, we know that we can extract a finite covering. In other words, there exists a finite subset $S \subset \mathbb{S}^{d-1}$ such that $\mathbb{S}^{d-1} \subset \bigcup_{\mathbf{u} \in S} \mathcal{B}(\mathbf{u}, \varepsilon)$. Put differently, for any $\mathbf{v} \in \mathbb{S}^{d-1}$, there exists $\mathbf{u} \in S$ such that $\|\mathbf{u} - \mathbf{v}\|_2 \leq \varepsilon$.

Now consider $\mathbf{u} \in S$. Given that $\text{SUPP}(\tilde{\mathcal{Q}})$ is bounded, we know that $\left|\mathcal{Q}_i^T \mathbf{u}\right| \in [0, M_{\mathcal{Q}}]$. Moreover, such variables $\left|\mathcal{Q}_i^T \mathbf{u}\right|$ are iid. By Hoeffding's inequality, for any $t > 0$, we have

$$\mathbb{P}\left[\left|\sum_{i \in \mathcal{I}} \left|\mathcal{Q}_i^T \mathbf{u}\right| - \mathcal{I}\mathbb{E}\left[\left|\mathcal{Q}^T \mathbf{u}\right|\right]\right| \geq \mathcal{I}t\right] \leq 2\exp\left(\frac{-2\mathcal{I}t^2}{M_{\mathcal{Q}}}\right). \tag{115}$$

Choosing $t = C_0/2$ then yields

$$\mathbb{P}\left[\sum_{i \in \mathcal{I}} \left|\mathcal{Q}_i^T \mathbf{u}\right| \leq \frac{C_0 \mathcal{I}}{2}\right] \leq \mathbb{P}\left[\left|\sum_{i \in \mathcal{I}} \left|\mathcal{Q}_i^T \mathbf{u}_{\theta - \theta^\dagger}\right| - \mathcal{I}\mathbb{E}\left[\left|\mathcal{Q}^T \mathbf{u}_{\theta - \theta^\dagger}\right|\right]\right| \geq \frac{\mathcal{I}C_0}{2}\right] \tag{116}$$

$$\leq 2\exp\left(\frac{-\mathcal{I}C_0^2}{2M_{\mathcal{Q}}}\right). \tag{117}$$

Taking a union bound for $\mathbf{u} \in S$ then guarantees

$$\mathbb{P}\left[\forall \mathbf{u} \in S, \ \sum_{i \in \mathcal{I}} \left|\mathcal{Q}_i^T \mathbf{u}\right| \geq \frac{C_0 \mathcal{I}}{2}\right] \geq 1 - 2|S|\exp\left(\frac{-\mathcal{I}C_0^2}{2M_{\mathcal{Q}}}\right), \tag{118}$$

which clearly goes to 1 as $\mathcal{I} \to \infty$. Thus $\forall \mathbf{u} \in S, \ \sum_{i \in \mathcal{I}} \left|\mathcal{Q}_i^T \mathbf{u}\right| \geq \frac{C_0 \mathcal{I}}{2}$ holds with high probability.

Now consider $\mathbf{v} \in \mathbb{S}^{d-1}$. We know that there exists $\mathbf{u} \in S$ such that $\|\mathbf{u} - \mathbf{v}\|_2 \leq \varepsilon$. Then, we have

$$\sum_{i \in [\mathcal{I}]} \left|\mathcal{Q}_i^T \mathbf{v}\right| = \sum_{i \in [\mathcal{I}]} \left|\mathcal{Q}_i^T \mathbf{u} + \mathcal{Q}_i^T (\mathbf{v} - \mathbf{u})\right| \tag{119}$$

$$\geq \sum_{i \in [\mathcal{I}]} \left|\mathcal{Q}_i^T \mathbf{u}\right| - \mathcal{I}M_{\mathcal{Q}} \|\mathbf{v} - \mathbf{u}\|_2 \tag{120}$$

$$\geq \frac{C_0 \mathcal{I}}{2} - \mathcal{I}M_{\mathcal{Q}} \frac{C_0}{4M_{\mathcal{Q}}} = \frac{C_0 \mathcal{I}}{4}, \tag{121}$$

which proves the lemma. $\qquad\square$

### D.3.3 LOWER BOUND ON THE DISCREPANCY BETWEEN PREFERRED AND REPORTED ANSWERS

**Lemma 29.** *Assume that $\tilde{\mathcal{Q}}$ has a bounded support, whose interior contains the origin. Suppose also that $\left\|\theta^\dagger\right\|_2 \leq \mathcal{K}$. Then there exists $A_{\mathcal{K}}$ such that, with high probability, we have*

$$\sum_{i \in [\mathcal{I}]} \Delta(\mathcal{Q}_i^T \theta, \mathcal{Q}_i^T \theta^\dagger) \geq A_{\mathcal{K}} \mathcal{I} \min\left\{\left\|\theta - \theta^\dagger\right\|_2, \left\|\theta - \theta^\dagger\right\|_2^2\right\}. \tag{122}$$

*Proof.* Note that by Cauchy-Schwarz inequality we have

$$\left|\mathcal{Q}_i^T \theta^\dagger\right| \leq \|\mathcal{Q}_i\|_2 \left\|\theta^\dagger\right\|_2 \leq M_{\mathcal{Q}} \mathcal{K}. \tag{123}$$

Thus, Lemma 26 implies the existence of a positive constant $d_{\mathcal{K}}$, such that for all $\theta \in \mathbb{R}^d$, we have

$$\sum_{i \in \mathcal{I}} \Delta\left(\mathcal{Q}_i^T \theta, \mathcal{Q}_i^T \theta^\dagger\right) \geq \sum_{i \in \mathcal{I}} \left(d_{\mathcal{K}} \left|\mathcal{Q}_i^T \theta - \mathcal{Q}_i^T \theta^\dagger\right| - d_{\mathcal{K}}\right) \tag{124}$$

$$= -d_{\mathcal{K}} \mathcal{I} + d_{\mathcal{K}} \left\|\theta - \theta^\dagger\right\|_2 \sum_{i \in \mathcal{I}} \left|\mathcal{Q}_i^T \mathbf{u}_{\theta - \theta^\dagger}\right|, \tag{125}$$

where $\mathbf{u}_{\theta-\theta^\dagger} \triangleq (\theta - \theta^\dagger)/\left\|\theta - \theta^\dagger\right\|_2$ is the unit vector in the direction of $\theta - \theta^\dagger$.

Now, by Lemma 28, we know that, with high probability, for all unit vectors $\mathbf{u} \in \mathbb{S}^{d-1}$, we have $\sum \left|\mathcal{Q}_i^T \mathbf{u}\right| \geq C\mathcal{I}$. Thus, for $\mathcal{I}$ sufficiently large, for any $\theta \in \mathbb{R}^d$, with high probability, we have

$$\sum_{i \in \mathcal{I}} \Delta(\mathcal{Q}_i^T \theta, \mathcal{Q}_i^T \theta^\dagger) \geq \frac{d_\mathcal{K} C_{min}}{2} \mathcal{I} \left\|\theta - \theta^\dagger\right\|_2 - d_\mathcal{K}\mathcal{I}. \tag{126}$$

Defining $e_\mathcal{K} \triangleq \frac{d_\mathcal{K} C_{min}}{4}$, and $f_\mathcal{K} \triangleq \frac{4}{C_{min}}$, for $\left\|\theta - \theta^\dagger\right\|_2 > f_\mathcal{K}$, we then have

$$\sum_{i \in \mathcal{I}} \Delta(\mathcal{Q}_i^T \theta, \mathcal{Q}_i^T \theta^\dagger) \geq e_\mathcal{K}\mathcal{I} \left\|\theta - \theta^\dagger\right\|_2. \tag{127}$$

We now focus on the case of $\left\|\theta - \theta^\dagger\right\|_2 \leq f_\mathcal{K}$. The triangle inequality yields $\|\theta\|_2 \leq \left\|\theta - \theta^\dagger\right\|_2 + \left\|\theta^\dagger\right\|_2 \leq f_\mathcal{K} + \mathcal{K}$. By Cauchy-Schwarz inequality, we then have $\left|\mathcal{Q}_i^T \theta\right| \leq (f_\mathcal{K} + \mathcal{K})M_\mathcal{Q} \triangleq g_\mathcal{K}$ and $\left|\mathcal{Q}_i^T \theta^\dagger\right| \leq \mathcal{K}M_\mathcal{Q} \leq g_\mathcal{K}$. Thus, by Lemma 25, we know there exists some constant $c_\mathcal{K}$ such that

$$\sum_{i \in \mathcal{I}} \left(\sigma(\mathcal{Q}_i^T \theta) - \sigma(\mathcal{Q}_i^T \theta^\dagger)\right)\left(\mathcal{Q}_i^T \theta - \mathcal{Q}_i^T \theta^\dagger\right) \geq \sum_{i \in \mathcal{I}} c_\mathcal{K} \left|\mathcal{Q}_i^T \theta - \mathcal{Q}_i^T \theta^\dagger\right|^2 \tag{128}$$

$$= \sum_{i \in \mathcal{I}} c_\mathcal{K}(\theta - \theta^\dagger)^T \mathcal{Q}_i \mathcal{Q}_i^T (\theta - \theta^\dagger) \tag{129}$$

$$= c_\mathcal{K}(\theta - \theta^\dagger)^T \left(\sum_{i \in \mathcal{I}} \mathcal{Q}_i \mathcal{Q}_i^T\right)(\theta - \theta^\dagger). \tag{130}$$

Since distribution $\tilde{\mathcal{Q}}$ is bounded (and thus sub-Gaussian), by Theorem 6, with high probability, we have

$$(\theta - \theta^\dagger)^T \left(\sum_{i \in \mathcal{I}} \mathcal{Q}_i \mathcal{Q}_i^T\right)(\theta - \theta^\dagger) \geq \frac{\lambda_{min}}{2} \mathcal{I} \left\|\theta - \theta^\dagger\right\|_2^2, \tag{131}$$

where $\lambda_{min}$ is the smallest eigenvalue of $\mathbb{E}\left[\mathcal{Q}_i \mathcal{Q}_i^T\right]$. Thus, for $\left\|\theta - \theta^\dagger\right\|_2 \leq f_\mathcal{K}$, we have

$$\sum_{i \in \mathcal{I}} \left(\sigma(\mathcal{Q}_i^T \theta) - \sigma(\mathcal{Q}_i^T \theta^\dagger)\right)\left(\mathcal{Q}_i^T \theta - \mathcal{Q}_i^T \theta^\dagger\right) \geq \frac{\lambda_{min} c_\mathcal{K}}{2} \mathcal{I} \left\|\theta - \theta^\dagger\right\|_2^2. \tag{132}$$

Combining this with (127), and defining $A_\mathcal{K} \triangleq \min\left\{\frac{\lambda_{min} c_\mathcal{K}}{2}, e_\mathcal{K}\right\}$, we then obtain the lemma. $\square$

### D.3.4 Proof that logistic regression is gradient-PAC*

Now we proceed with the proof that logistic regression is gradient-PAC*.

*Proof of Theorem 3.* Note that $\sigma(-z) = e^{-z}\sigma(z) = 1 - \sigma(z)$ and $\sigma'(z) = e^{-z}\sigma^2(z)$. We then have

$$\nabla_\theta \ell(\theta, \mathcal{Q}, \mathcal{A}) = -\frac{\sigma'(\mathcal{A}\mathcal{Q}^T \theta)\mathcal{A}\mathcal{Q}}{\sigma(\mathcal{A}\mathcal{Q}^T \theta)} = -e^{-\mathcal{A}\mathcal{Q}^T \theta}\sigma(\mathcal{A}\mathcal{Q}^T \theta)\mathcal{A}\mathcal{Q} \tag{133}$$

$$= -\sigma(-\mathcal{A}\mathcal{Q}^T \theta)\mathcal{A}\mathcal{Q} = \left(\sigma(\mathcal{Q}^T \theta) - \mathbb{1}[\mathcal{A} = 1]\right)\mathcal{Q}, \tag{134}$$

where $\mathbb{1}\left[\mathcal{A} = 1\right]$ is the indicator function that outputs 1 if $\mathcal{A} = 1$, and 0 otherwise. As a result,

$$(\theta - \theta^\dagger)^T \nabla_\theta \mathcal{L}(\theta, \mathcal{D}) = \tag{135}$$

$$(\theta - \theta^\dagger)^T \left( \sum_{i \in \mathcal{I}} \left( \sigma(\mathcal{Q}_i^T \theta) - \mathbb{1}\left[\mathcal{A}_i = 1\right] \right) \mathcal{Q}_i \right) + 2\nu(\theta - \theta^\dagger)^T \theta \tag{136}$$

$$= (\theta - \theta^\dagger)^T \left( \sum_{i \in \mathcal{I}} \left( \sigma(\mathcal{Q}_i^T \theta) - \sigma(\mathcal{Q}_i^T \theta^\dagger) + \sigma(\mathcal{Q}_i^T \theta^\dagger) - \mathbb{1}\left[\mathcal{A}_i = 1\right] \right) \mathcal{Q}_i \right) \tag{137}$$

$$+ 2\nu \left\| \theta - \theta^\dagger \right\|_2^2 + 2\nu(\theta - \theta^\dagger)^T \theta^\dagger \tag{138}$$

$$= \sum_{i \in [\mathcal{I}]} \Delta \left( \mathcal{Q}_i^T \theta, \mathcal{Q}_i^T \theta^\dagger \right) + (\theta - \theta^\dagger)^T \left( \sum_{i \in \mathcal{I}} \left( \sigma(\mathcal{Q}_i^T \theta^\dagger) - \mathbb{1}\left[\mathcal{A}_i = 1\right] \right) \mathcal{Q}_i \right) \tag{139}$$

$$+ 2\nu \left\| \theta - \theta^\dagger \right\|_2^2 + 2\nu(\theta - \theta^\dagger)^T \theta^\dagger. \tag{140}$$

By Lemma 29, with high probability, we have

$$\sum_{i \in [\mathcal{I}]} \Delta \left( \mathcal{Q}_i^T \theta, \mathcal{Q}_i^T \theta^\dagger \right) \geq A_\mathcal{K} \mathcal{I} \min \left\{ \left\| \theta - \theta^\dagger \right\|_2, \left\| \theta - \theta^\dagger \right\|_2^2 \right\}. \tag{141}$$

To control the second term of (139), note that the random vectors $Z_i \triangleq \left( \sigma(\mathcal{Q}_i^T \theta^\dagger) - \mathbb{1}\left[\mathcal{A}_i = 1\right] \right) \mathcal{Q}_i$ are iid with norm at most $M_\mathcal{Q}$. Moreover, since $\mathbb{E}\left[\mathbb{1}\left[\mathcal{A}_i = 1\right] | \mathcal{Q}_i \right] = \sigma(\mathcal{Q}_i^T \theta^\dagger)$, by the tower rule, we have $\mathbb{E}\left[Z_i\right] = \mathbb{E}\left[\mathbb{E}\left[Z_i | \mathcal{Q}_i\right]\right] = 0$. Therefore, by applying Hoeffding's bound to every coordinate of $Z_i$, and then taking a union bound, for any $B > 0$, we have

$$\mathbb{P}\left[ \left\| \sum_{i \in \mathcal{I}} Z_i \right\|_2 \geq B\mathcal{I}^{3/4} \right] \leq 2d \exp\left( -\frac{B^2 \sqrt{\mathcal{I}}}{2d M_\mathcal{Q}^2} \right). \tag{142}$$

Applying now Cauchy-Schwarz inequality, with high probability, we have

$$\left| (\theta - \theta^\dagger)^T \left( \sum_{i \in \mathcal{I}} \left( \sigma(\mathcal{Q}_i^T \theta^\dagger) - \mathbb{1}\left[\mathcal{A}_i = 1\right] \right) \mathcal{Q}_i \right) \right| \leq B\mathcal{I}^{3/4} \left\| \theta - \theta^\dagger \right\|_2.$$

Combining this with (132) and using $\left\| \theta^\dagger \right\|_2^2 \leq \mathcal{K}$, we then have

$$(\theta - \theta^\dagger)^T \nabla_\theta \mathcal{L}(\theta, \mathcal{D}) \tag{143}$$

$$\geq (A_\mathcal{K} \mathcal{I} + \nu) \left\{ \left\| \theta - \theta^\dagger \right\|_2, \left\| \theta - \theta^\dagger \right\|_2^2 \right\} - (B\mathcal{I}^{(3/4)} + 2\nu\mathcal{K}) \left\| \theta - \theta^\dagger \right\|_2 \tag{144}$$

$$\geq A_\mathcal{K} \mathcal{I} \left\{ \left\| \theta - \theta^\dagger \right\|_2, \left\| \theta - \theta^\dagger \right\|_2^2 \right\} - B_\mathcal{K} \mathcal{I}^{(3/4)} \left\| \theta - \theta^\dagger \right\|_2, \tag{145}$$

where $B_\mathcal{K} = B + 2\nu\mathcal{K}$. This shows that Assumption 2 is satisfied for logistic loss for $\alpha = 3/4$, and $A_\mathcal{K}$ and $B_\mathcal{K}$ as previously defined.

$\square$

# E    PROOFS OF LOCAL PAC*-LEARNABILITY

Let us now prove Lemma 5. To do so, consider the preferred models $\vec{\theta^\dagger}$ and a subset $H \subset [N]$ of honest users. Denote $\vec{\mathcal{D}}_{-H}$ the datasets provided by users $n \in [N] - H$. Each honest user $h \in H$ provides an honest dataset $\mathcal{D}_h$ of cardinality at least $\mathcal{I} \geq 1$. Consider the bound $K_H \triangleq \max_{h \in H} \left\| \theta_h^\dagger \right\|_2$ on the parameter norm of honest active users $h \in H$.

## E.1    BOUNDS ON THE OPTIMA

Before proving the theorem, we prove a useful lemma that bounds the set of possible values for the global model and honest local models.

**Lemma 30.** *Assume that $\mathcal{R}$ and $\ell$ are nonnegative. For $\mathcal{I}$ large enough, if all honest active nodes $h \in H$ provide at least $\mathcal{I}$ data, then, with high probability, $\vec{\theta}_H^*$ must lie in a compact subset of $\mathbb{R}^{d \times H}$ that does not depend on $\mathcal{I}$.*

*Proof.* Denote $L^0 \triangleq \text{Loss}(0, (\vec{\theta}_H^\dagger, 0_{-H}), (\emptyset, \vec{\mathcal{D}}_{-H}))$. Essentially, we will show that, if $\vec{\theta}_H^*$ is too far from $\vec{\theta}_H^\dagger$, then the loss will take values strictly larger than $L^0$.

Assumption 2 implies the existence of an event $\mathcal{E}$ that occurs with probability at least $P_0 \triangleq P(K_H, \mathcal{I})^{|H|}$, under which, for any $\theta_h \in \mathbb{R}^d$, we have

$$\left(\theta_h - \theta_h^\dagger\right)^T \nabla \mathcal{L}_h\left(\theta_h\right) \geq A_{K_H} \mathcal{I} \min\left\{\left\|\theta_h - \theta_h^\dagger\right\|_2, \left\|\theta_h - \theta_h^\dagger\right\|_2^2\right\} - B_{K_H} \mathcal{I}^\alpha \left\|\theta_h - \theta_h^\dagger\right\|_2, \quad (146)$$

which implies

$$\mathbf{u}_{(\theta_h - \theta_h^\dagger)}^T \nabla \mathcal{L}_h\left(\theta_h\right) \geq A_{K_H} \mathcal{I} \min\left\{1, \left\|\theta_h - \theta_h^\dagger\right\|_2\right\} - B_{K_H} \mathcal{I}^\alpha. \quad (147)$$

Note also that $P_0 \to 1$ as $\mathcal{I} \to \infty$. We now integrate both sides over the line segment from $\theta_h^\dagger$ to $\theta_h$. The fundamental theorem of calculus for line integrals then yields

$$\mathcal{L}_h\left(\theta_h\right) - \mathcal{L}_h\left(\theta_h^\dagger\right) = \left\|\theta_h - \theta_h^\dagger\right\|_2 \int_{t=0}^1 \mathbf{u}_{(\theta_h - \theta_h^\dagger)}^T \nabla \mathcal{L}\left(\theta_h^\dagger + t(\theta_h - \theta_h^\dagger)\right) dt \quad (148)$$

$$\geq \left\|\theta_h - \theta_h^\dagger\right\|_2 \int_{t=0}^1 \left(A_{K_H} \mathcal{I} \min\left\{1, t\left\|\theta_h - \theta_h^\dagger\right\|_2\right\} - B_{K_H} \mathcal{I}^\alpha\right) dt \quad (149)$$

$$= \left\|\theta_h - \theta_h^\dagger\right\|_2 \int_{t=0}^1 \left(A_{K_H} \mathcal{I} \min\left\{1, t\left\|\theta_h - \theta_h^\dagger\right\|_2\right\}\right) dt - B_{K_H} \mathcal{I}^\alpha \left\|\theta_h - \theta_h^\dagger\right\|_2. \quad (150)$$

Now, if $\left\|\theta_h - \theta_h^\dagger\right\|_2 > 2$, we then have

$$\mathcal{L}_h\left(\theta_h\right) - \mathcal{L}_h\left(\theta_h^\dagger\right) \geq \left(\frac{A_{K_H} \mathcal{I}}{2} - B_{K_H} \mathcal{I}^\alpha\right) \left\|\theta_h - \theta_h^\dagger\right\|_2 \quad (151)$$

$$\geq A_{K_H} \mathcal{I} - 2 B_{K_H} \mathcal{I}^\alpha. \quad (152)$$

Now for $\mathcal{I} > \mathcal{I}_1 \triangleq \max\left\{2L^0 / A_{K_H}, (4 B_{K_H} / A_{K_H})^{\frac{1}{1-\alpha}}\right\}$, we have

$$\mathcal{L}_h\left(\theta_h\right) - \mathcal{L}_h\left(\theta_h^\dagger\right) > L^0. \quad (153)$$

This implies that if $\left\|\theta_h - \theta_h^\dagger\right\|_2 > 2$ for any $h \in H$, then we have

$$\text{Loss}(0, (\vec{\theta}_H^\dagger, 0_{-H}), \vec{\mathcal{D}}) < \text{Loss}(\rho, (\vec{\theta}_H, \vec{\theta}_{-H}), \vec{\mathcal{D}}), \quad (154)$$

regardless of $\rho$ and $\theta_{-H}$. Therefore, we must have $\left\|\theta_h^\dagger - \theta_h^*\right\|_2 \leq 2$. Such inequalities describe a bounded closed subset of $\mathbb{R}^{d \times H}$, which is thus compact. $\qquad \square$

**Lemma 31.** *Assume that $\mathcal{R}(\rho, \theta) \to \infty$ as $\|\rho - \theta\|_2 \to \infty$, and that $\left\|\theta_h^\dagger - \theta_h^*\right\|_2 \leq 2$ for all honest users $h \in H$. Then $\rho^*$ must lie in a compact subset of $\mathbb{R}^d$ that does not depend on $\mathcal{I}$.*

*Proof.* Consider an honest user $h'$. Given our assumption on $\mathcal{R} \to \infty$, we know that there exists $D_{K_H}$ such that if $\|\rho - \theta_{h'}^*\|_2 \geq D_{K_H}$, then $\mathcal{R}(\rho, \theta_{h'}^*) \geq L^0 + 1$. Thus any global optimum $\rho^*$ must satisfy $\left\|\rho^* - \theta_{h'}^\dagger\right\|_2 \leq \|\rho^* - \theta_{h'}^*\|_2 + \left\|\theta_{h'}^* - \theta_{h'}^\dagger\right\|_2 \leq D_{K_H} + 2$. $\qquad \square$

### E.2 PROOF OF LEMMA 5

*Proof of Lemma 5.* Fix $\varepsilon, \delta > 0$. We want to show the existence of some value of $\mathcal{I}(\varepsilon, \delta, \vec{\mathcal{D}}_{-H}, \vec{\theta}^{\dagger})$ that will guarantee $(\varepsilon, \delta)$-locally PAC* learning for honest users.

By lemmas 30 and 31, we know that the set $C$ of possible values for $(\rho^*, \vec{\theta}_H^*)$ is compact. Now, we define

$$E_{K_H} \triangleq \max_{(\rho, \theta) \in C} \|\nabla_\theta \mathcal{R}(\rho, \theta)\|_2 \tag{155}$$

the maximum of the norm of achievable gradients at the optimum. We know this maximum exists since $C$ is compact.

Using the optimality of $(\rho^*, \vec{\theta}^*)$, for all $h \in H$, we have

$$0 \in (\theta_h^* - \theta_h^\dagger)^T \nabla_{\theta_h} \text{Loss}(\rho^*, \vec{\theta}^*) \tag{156}$$

$$= (\theta_h^* - \theta_h^\dagger)^T \nabla \mathcal{L}_h(\theta_h^*) + (\theta_h^* - \theta_h^\dagger)^T \nabla_{\theta_h} \mathcal{R}(\rho^*, \theta_h^*) \tag{157}$$

$$\geq (\theta_h^* - \theta_h^\dagger)^T \nabla \mathcal{L}_h(\theta_h^*) - \left\|\theta_h^* - \theta_h^\dagger\right\|_2 \|\nabla_{\theta_h} \mathcal{R}(\rho^*, \theta_h^*)\|_2 \tag{158}$$

$$\geq (\theta_h^* - \theta_h^\dagger)^T \nabla \mathcal{L}_h(\theta_h^*) - E_{K_H} \left\|\theta_h^* - \theta_h^\dagger\right\|_2. \tag{159}$$

We now apply assumption 2 for $\theta = \theta_h^*$ (for $h \in H$). Thus, there exists some other event $\mathcal{E}'$ with probability at least $P_0$, under which, for all $h \in H$, we have

$$0 \geq A_{K_H} \mathcal{I} \min\left\{\left\|\theta_h^* - \theta_h^\dagger\right\|_2, \left\|\theta_h^* - \theta_h^\dagger\right\|_2^2\right\} - B_{K_H} \mathcal{I}^\alpha \left\|\theta_h^* - \theta_h^\dagger\right\|_2 - E_{K_H} \left\|\theta_h^* - \theta_h^\dagger\right\|_2. \tag{160}$$

Now if $\mathcal{I} > \mathcal{I}_2 \triangleq \max\left\{2E_{K_H}/A_{K_H}, (2B_{K_H}/A_{K_H})^{\frac{1}{1-\alpha}}\right\}$ this inequality cannot hold for $\left\|\theta_h^* - \theta_h^\dagger\right\|_2 \geq 1$. Therefore, for $\mathcal{I} > \mathcal{I}_2$, we have $\left\|\theta_h^* - \theta_h^\dagger\right\|_2 < 1$, and thus,

$$0 \geq A_{K_H} \mathcal{I} \left\|\theta_h^* - \theta_h^\dagger\right\|_2^2 - B_{K_H} \mathcal{I}^\alpha \left\|\theta_h^* - \theta_h^\dagger\right\|_2 - E_{K_H} \left\|\theta_h^* - \theta_h^\dagger\right\|_2 \tag{161}$$

and thus,

$$\left\|\theta_h^* - \theta_h^\dagger\right\|_2 \leq \frac{B_{K_H} \mathcal{I}^\alpha + E_{K_H}}{A_{K_H} \mathcal{I}}. \tag{162}$$

Now note that $\mathbb{P}[\mathcal{E} \wedge \mathcal{E}'] = 1 - \mathbb{P}[\neg \mathcal{E} \vee \neg \mathcal{E}'] \geq 1 - \mathbb{P}[\neg \mathcal{E}] - \mathbb{P}[\neg \mathcal{E}'] = 2P_0 - 1$. It now suffices to consider $\mathcal{I}$ larger than $\mathcal{I}_2$ and large enough so that $P(K_H, \mathcal{I})^{|H|} \geq 1 - \delta/2$ (whose existence is guaranteed by Assumption 2, and which guarantees $2P_0 - 1 \geq 1 - \delta$) and so that $\frac{B_{K_H}\mathcal{I}^\alpha + E_{K_H}}{A_{K_H}\mathcal{I}} \leq \varepsilon$ to obtain the theorem. $\square$

## F CONVERGENCE OF CGA AGAINST $\ell_2^2$

To write our proof, we define $\text{Loss}_{-s}^\rho : \mathbb{R}^d \to \mathbb{R}$ by

$$\text{Loss}_{-s}^\rho(\rho) \triangleq \inf_{\vec{\theta}} \left\{\text{Loss}(\rho, \vec{\theta}, \vec{\mathcal{D}}) - \mathcal{L}_s(\theta_s, \mathcal{D}_s) - \mathcal{R}(\rho, \theta_s)\right\} \tag{163}$$

$$= \inf_{\vec{\theta}} \sum_{n \neq s} \mathcal{L}(\theta_n, \mathcal{D}_n) + \lambda \sum_{n \neq s} \|\rho - \theta_n\|_2^2. \tag{164}$$

In other words, it is the loss when local models are optimized, and when the data of strategic user $s$ are removed.

**Lemma 32.** *Assuming $\ell_2^2$ regularization and convex loss-per-input functions $\ell$, for any datasets $\vec{\mathcal{D}}$, Loss is strongly convex. As a result, so is $\text{Loss}_{-s}^\rho$.*

*Proof.* Note that the global loss can be written as a sum of convex function, and of $\nu \sum \|\theta_n\|_2^2 + \|\rho - \theta_1\|_2^2$. Using tricks similar to the proof of Lemma 11, we see that the loss is strongly convex. The latter part of the lemma is then a straightforward application of Lemma 10. $\square$

We now move on to the proof of Theorem 4. Note that our statement of the proof was not fully explicit, especially about the upper bound on the constant learning rate $\eta$. Here, we prove that it holds for $\eta_t = \eta \le 1/3L$, where $L$ is a constant such that $\text{Loss}^\rho_{-s}$ is $L$-smooth. The existence of $L$ is guaranteed by Lemma 13.

*Proof of Theorem 4.* Note that by Lemma 9, $\text{Loss}^\rho_{-s}$ is convex, differentiable and $L$-smooth, and $\nabla\text{Loss}^\rho_{-s}(\rho^t) = g^{\dagger,t}_{-s}$. For $\ell^2_2$ regularization, we have $\text{GRAD}(\rho) = \mathbb{R}^d$ for all $\rho \in \mathbb{R}^d$. Then the minimum of equation 6 is zero, which is obtained when $g^t_s \triangleq \frac{\rho^t - \theta^\dagger_s}{\eta} - \hat{g}^t_{-s} = g^{t-1}_s + \frac{\rho^t - \theta^\dagger_s}{\eta} + \frac{\rho^t - \rho^{t-1}}{\eta}$. Note that

$$\rho^{t+1} = \rho^t - \eta g^{\dagger,t}_{-s} - \eta g^t_s \tag{165}$$

$$= \rho^t - \eta g^{\dagger,t}_{-s} - (\rho^t - \theta^\dagger_s) + (\rho^{t-1} - \rho^t) - \eta g^{t-1}_s \tag{166}$$

$$= \theta^\dagger_s - \eta_t(g^{\dagger,t}_{-s} + g^{t-1}_s) + \eta(g^{\dagger,t-1}_{-s} + g^{t-1}_s) \tag{167}$$

$$= \theta^\dagger_s - \eta(g^{\dagger,t}_{-s} - g^{\dagger,t-1}_{-s}). \tag{168}$$

Therefore, $\rho^{t+1} - \rho^t = \eta(g^{\dagger,t}_{-s} - g^{\dagger,t-1}_{-s}) - \eta(g^{\dagger,t-1}_{-s} - g^{\dagger,t-2}_{-s})$.

Then, using the $L$-smoothness of $\text{Loss}^\rho_{-s}$, and denoting $u_t \triangleq \left\|\rho^{t+1} - \rho^t\right\|_2$, we have $u_{t+1} \le L\eta_t u_t + L\eta_{t-1}u_{t-1}$. Now assume that $\eta \le 1/3L$. Then $u_{t+1} \le \frac{1}{3}(u_t + u_{t-1})$. We then know that $u_{t+2} \le \frac{1}{3}(u_{t+1} + u_t) \le \frac{1}{3}(\frac{1}{3}(u_t + u_{t-1}) + u_t) = \frac{4}{9}u_t + \frac{1}{9}u_{t-1}$.

Now define $v_t \triangleq u_t + u_{t-1}$. We then have $v_{t+2} \le u_{t+2} + u_{t+1} \le \frac{7}{9}u_t + \frac{4}{9}u_{t-1} \le \frac{7}{9}(u_t + u_{t-1}) \le \frac{7}{9}v_t$. By induction, we know that $v_t \le (7/9)^{(t-1)/2}\max\{v_0, v_1\} \le (\sqrt{7}/3)^t((\sqrt{7}/3)\max\{v_0, v_1\})$. Thus, defining $\alpha \triangleq \sqrt{7}/3 < 1$, there exists $C > 0$ such that $u_t \le v_t \le C\alpha^t$. This implies that $\sum\left\|\rho^{t+1} - \rho^t\right\|_2 \le \sum C\alpha^t < \infty$. Thus $\sum(\rho^{t+1} - \rho^t)$ converges, which implies the convergence of $\rho^t$ to a limit $\rho^\infty$. By $L$-smoothness, we know that $g^{\dagger,t}_{-s}$ must converge too. Taking equation 168 to the limit then implies $\rho^\infty = \theta^\dagger_s$. This shows that the strategic user achieves precisely what they want with CGA. It is thus optimal. $\qquad\square$

# G  CGA ON MNIST

In this section, CGA is executed against 10 honest users, each one having 6,000 randomly and data points of MNIST, drawn randomly and independently. CGA is run by a strategic user whose target model $\theta^\dagger_s$ labels 0's as 1's, 1's as 2's, and so on, until 9's as 0's. We learn $\theta^\dagger_s$ by relabeling the MNIST training dataset and learning from the relabeled data. We use $\lambda = 1$, Adam optimizer and a decreasing learning rate.

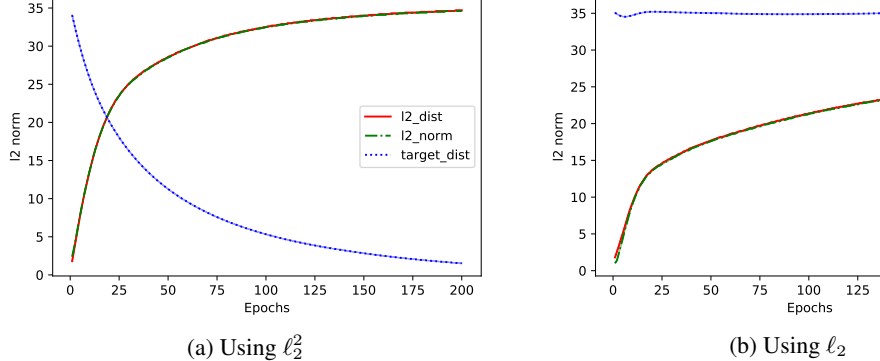

(a) Using $\ell^2_2$  (b) Using $\ell_2$

Figure 16: Norm of global model, distance to initialisation and distance to target, under attack by CGA. In particular, we see that the attack against $\ell^2_2$ is successful, as the distance between the global model and the target model goes to zero.

## H SINGLE DATA POISONING FOR LEAST SQUARE LINEAR REGRESSION

*Proof of Theorem 5.* We define the minimized loss with respect to $\rho$ and without strategic user $s$ by

$$\text{LOSS}^*_{-s}(\rho, \vec{\mathcal{D}}_{-s}) \triangleq \min_{\vec{\theta}_{-s} \in \mathbb{R}^{d \times (N-1)}} \left\{ \sum_{n \neq s} \mathcal{L}_n(\theta_n, \mathcal{D}_n) + \sum_{n \neq s} \lambda \|\theta_n - \rho\|_2^2 \right\}. \tag{169}$$

Now consider a subgradient $g \in \nabla_\rho \text{LOSS}^*_{-s}(\theta_s^\dagger, \vec{\mathcal{D}}_{-s})$ of the minimized loss at $\theta_s^\dagger$. For $x \triangleq \frac{-g}{2\lambda}$, then have $-g \in \nabla \left( \lambda \|x\|_2^2 \right)$. We then define $\theta_s^\spadesuit \triangleq \theta_s^\dagger - x$.

$$0 = g - g \in \nabla_\rho \text{LOSS}^*_{-s}(\theta_s^\dagger, \vec{\mathcal{D}}_{-s}) + \nabla_\rho \left( \lambda \|\theta_s^\spadesuit - \theta_s^\dagger\|_2^2 \right) \tag{170}$$

$$= \nabla_\rho \text{LOSS}_s(\theta_s^\dagger, \vec{\theta}^*_{-s}(\theta_s^\spadesuit, \vec{\mathcal{D}}_{-s}), \theta_s^\spadesuit, \vec{\mathcal{D}}_{-s}), \tag{171}$$

where $\text{LOSS}_s$ is defined by (38). Now consider the data point $(\mathcal{Q}, \mathcal{A}) = (g, g^T \theta_s^\spadesuit - 1)$. For $\mathcal{D}_s = \{(\mathcal{Q}, \mathcal{A})\}$, we then have $\nabla \mathcal{L}_s(\theta_s^\spadesuit, \mathcal{D}_s) = g$, which implies

$$\nabla_{\theta_s} \text{LOSS}(\theta_s^\dagger, (\theta_s^\spadesuit, \vec{\theta}^*_{-s}(\theta_s^\spadesuit, \vec{\mathcal{D}}_{-s}), \vec{\mathcal{D}}) = 0. \tag{172}$$

Combining it all together with the uniqueness of the solution then yields

$$\arg\min_{(\rho, \vec{\theta})} \left\{ \text{LOSS}(\rho, \vec{\theta}, \vec{\mathcal{D}}) \right\} = \left( \theta_s^\dagger, \left( \theta_s^\spadesuit, \vec{\theta}^*_{-s}(\theta_s^\spadesuit, \vec{\mathcal{D}}_{-s}) \right) \right), \tag{173}$$

which is what we wanted. $\qquad\qquad\square$

## I DATA POISONING AGAINST LINEAR CLASSIFICATION

### I.1 GENERATING EFFICIENT POISONING DATA

For every label $a \in \{1, \ldots, 9\}$, we define $y_a \triangleq \theta_a^\spadesuit - \theta_0^\spadesuit$, and $c_a \triangleq -(\theta_{a0}^\spadesuit - \theta_{00}^\spadesuit)$ (where $\theta_{a0}^\spadesuit$ is the bias of the linear classifier). The indifference subspace $V$ is then the set of images $\mathcal{Q} \in \mathbb{R}^d$ such that $\mathcal{Q}^T y_a = c_a$ for all $a \in \{1, \ldots, 9\}$.

To project any image $X \in \mathbb{R}^d$ on $V$, let us first construct an orthogonal basis of the vector space orthogonal to $V$, using the Gram-Schmidt algorithm. Namely, we first define $z_1 \triangleq y_1$. Then, for any answer $a \in \{1, \ldots, 9\}$, we define

$$z_a \triangleq y_a - \sum_{b < a} y_a^T z_b \frac{z_b}{\|z_b\|_2^2}. \tag{174}$$

It is easy to check that for $b < a$, we have $z_a^T z_b = 0$. Moreover, if $\mathcal{Q} \in V$, then

$$z_a^T \mathcal{Q} = y_a^T \mathcal{Q} - \sum_{b < a} \frac{(y_a^T z_b)(z_b^T \mathcal{Q})}{\|z_b\|_2^2} = c_a - \sum_{b < a} \frac{(y_a^T z_b)(z_b^T \mathcal{Q})}{\|z_b\|_2^2}. \tag{175}$$

By induction, we see that $z_a^T \mathcal{Q}$ is a constant independent from $\mathcal{Q}$. Indeed, for $a = 1$, this is clear as $z_1^T \mathcal{Q} = y_1^T \mathcal{Q} = c_1$. Moreover, for $a > 1$, then, in the computation of $z_a^T \mathcal{Q}$, $\mathcal{Q}$ always appear as $z_b^T \mathcal{Q}$ for $b < a$. Moreover, denoting $c'_a$ the constant such that $z_a^T \mathcal{Q} = c'_a$ for all $a \in \{1, \ldots 9\}$, we see that these constants can be computed by

$$c'_a = c_a - \sum_{b < a} \frac{y_a^T z_b}{\|z_b\|_2^2} c'_b. \tag{176}$$

Finally, we can simply perform repeated projection onto the hyperplanes where $a$ is equally probable as the answer 0. To do this, we first define the orthogonal projection $P(X, y, c)$ of $X \in \mathbb{R}^d$ on the hyperplane $x^T y = c$, which is given by

$$P(X, y, c) = X - (X^T y - c) \frac{y}{\|y\|_2^2}. \tag{177}$$

It is straightforward to verify that $P(X, y, c)^T y = c$ and that $P(P(X, y, c), y, c) = P(X, y, c)$. We then canonically define repeated projection by induction, as

$$P(X, (y_1, \ldots, y_{k+1}), (c_1, \ldots, c_{k+1})) \triangleq P(P(X, (y_1, \ldots, y_k), (c_1, \ldots, c_k)), y_{k+1}, c_{k+1}). \quad (178)$$

Now consider any image $X \in \mathbb{R}^d$. Its projection can be obtained by setting

$$\mathcal{Q} \triangleq P(X, (z_1, \ldots, z_9), (c_1', \ldots c_9')) + \xi. \quad (179)$$

Note that to avoid being exactly on the boundary, and thus retrieve information about the scales of $\theta^{\spadesuit}$ and on which side of the boundary favors which label, we add a small noise $\xi$, to make sure $\mathcal{Q}$ does not lie exactly on $V$ (which would lead to multiple solutions for the learning), but small enough so that the probabilities of the different label remain close to $0.1$ (the equiprobable probability).

We acknowledge that images obtained this way may not be in $[0, 1]^d$, like the images of the MNIST dataset. In general, one could search for points $\mathcal{Q} \in V \cap [0, 1]^d$. Note that in theory, by Theorem 3 (or a generalization of it), labeling random images in $[0, 1]^d$ should suffice. However, in the case where $V \cap [0, 1]^d$ is empty (typically if no image in $[0, 1]$ is argued by model $\theta_s^{\spadesuit}$ as realistically a 9), this procedure may require the labeling of significantly more images to be successful.

## I.2 A BRIEF THEORY OF DATA POISONING FOR LINEAR CLASSIFICATION

Using the efficient poisoning data fabrication, we thus have a set of images $(\mathcal{Q}, p(\mathcal{Q}))$, where $p_a(\mathcal{Q})$ is the probability assigned to image $\mathcal{Q}$ and label $a$. This defines the following local loss for the strategic node:

$$\mathcal{L}_s(\theta_s, \mathcal{D}_s) = \sum_{(\mathcal{Q}, p(\mathcal{Q})) \in \mathcal{D}_s} \sum_{a \in \{0, 1, \ldots, 9\}} p_a(\mathcal{Q}) \ln \sigma_a(\theta_s, \mathcal{Q}), \quad (180)$$

where $\sigma_a(\theta_s, \mathcal{Q}) = \frac{\exp(\theta_{sa}^T \mathcal{Q} + \theta_{sa0})}{\sum \exp(\theta_{sb}^T \mathcal{Q} + \theta_{sb0})}$ is the probability that image $\mathcal{Q}$ has label $a$, according to the model $\theta_s$. We acknowledge that such labelings of queries is unusual. Evidently, in practice, an image may be labeled $N$ times, and the number of labels $N_a$ it received can be set to be approximately $N_a \approx N p_a(\mathcal{Q})$.

It is noteworthy that the gradient of the loss function is then given by

$$\left(\theta_s - \theta_s^{\spadesuit}\right)^T \nabla_{\theta_s} \mathcal{L}_s(\theta_s, \mathcal{D}_s) = \sum_{\mathcal{Q} \in \mathcal{D}_s} \sum_{a \in \{0, 1, \ldots, 9\}} \left(\sigma_a(\theta_s, \mathcal{Q}) - \sigma_a(\theta_s^{\spadesuit}, \mathcal{Q})\right) \left(\theta_{sa} - \theta_{sa}^{\spadesuit}\right)^T \mathcal{Q}^+, \quad (181)$$

where we defined $\mathcal{Q}^+ \triangleq (1, \mathcal{Q})$ (which allows to factor in the bias of the model. This shows that $\nabla_{\theta_s} \mathcal{L}_s(\theta_s, \mathcal{D}_s)$ points systematically away from $\theta_s^{\spadesuit}$, and thus that gradient descent will move towards $\theta_s^{\spadesuit}$.

In fact, if the set of images $\mathcal{Q}$ cover all dimensions (which occurs if there are $\Omega(d)$ images, which is the case for 2,000 images, since $d = 784$), then gradient descent will always move the model in the direction of $\theta_s^{\spadesuit}$, which will be the minimum. Moreover, by overweighting each data $(\mathcal{Q}, p(\mathcal{Q}))$ by a factor $\alpha$ (as though the image $\mathcal{Q}$ was labeled $\alpha$ times), we can guarantee gradient-PAC* learning, which means that we will have $\theta_s^* \approx \theta_s^{\spadesuit}$, even in the personalized federated learning framework. This shows why data poisoning should work in theory, with relatively few data injections.

Note that the number of other users does make learning harder. Indeed, the gradient of the regularization $\mathcal{R}(\rho, \theta_s)$ at $\rho = \theta_s^{\dagger}$ and $\theta_s = \theta_s^{\spadesuit}$ is equal to $2\lambda \left\| \theta_s^{\dagger} - \theta_s^{\spadesuit} \right\|_2$. As the number $N - 1$ of other users grows, we should expect this distance to grow roughly proportionally to $N$. In order to make strategic user $s$ robustly learn $\theta_s^{\spadesuit}$, the norm of the gradient of the local loss $\mathcal{L}_s$ at $\theta_s^{\dagger}$ must be vastly larger than $2\lambda \left\| \theta_s^{\dagger} - \theta_s^{\spadesuit} \right\|_2$. This means that the value of $\alpha$ (or, equivalently, the number of data injected in $\mathcal{D}_s$) must also grow proportionally to $N$.

## I.3 INITIALIZATION OF THE LEARNING ALGORITHM

The convergence to the optimum is slow. But given that the problem is convex, we focus here mostly on showing that the minimum is indeed a poisoned model. To boost the convergence, we initialize our learning algorithm at a point close to what we expect to be the minimum, by taking this minimum and adding a Gaussian noise, and then we observe the convergence to this minimum.

## J    CIFAR-10 ON VGG 13-BN EXPERIMENTS

We considered VGG 13-BN, which was pretrained on cifar-10 by Phan (2021). We now assume that 10 nodes are given part of the cifar-10 database, while a strategic node also joins to the personalized federated gradient descent algorithm. The strategic node's goal is to bias the global model towards a target model, which misclassifies the cifar-10 data, by reclassifying 0 into 1, 1 into 2... and 9 into 0.

### J.1    COUNTER-GRADIENT ATTACK

We first show the result of performing counter-gradient attack on the last layer of the neural network. Essentially, images are now reduced to their vector embedding, and the last layer performs a simple linear classification akin to the case of MNIST (see Appendix G).

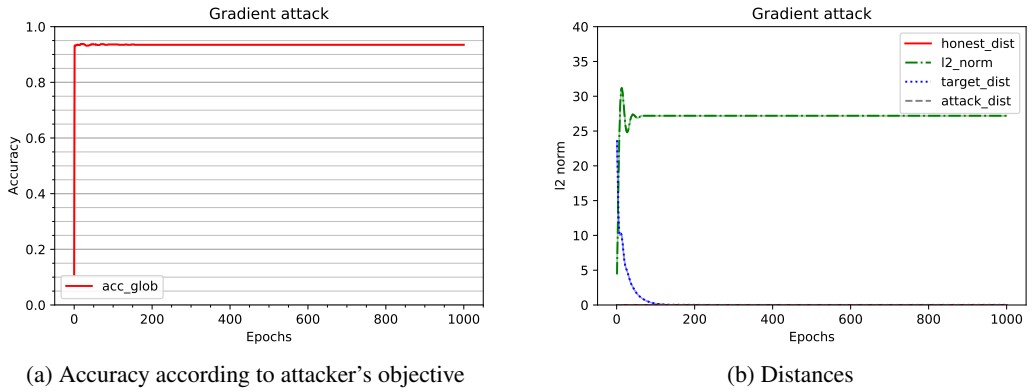

(a) Accuracy according to attacker's objective                    (b) Distances

Figure 17: CGA on cifar-10.

### J.2    RECONSTRUCTING A MODEL ATTACK

Reconstructing an attack model whose effect is equivalent to the counter-gradient attack is identical to what was done in the case of MNIST (see Section 5.2).

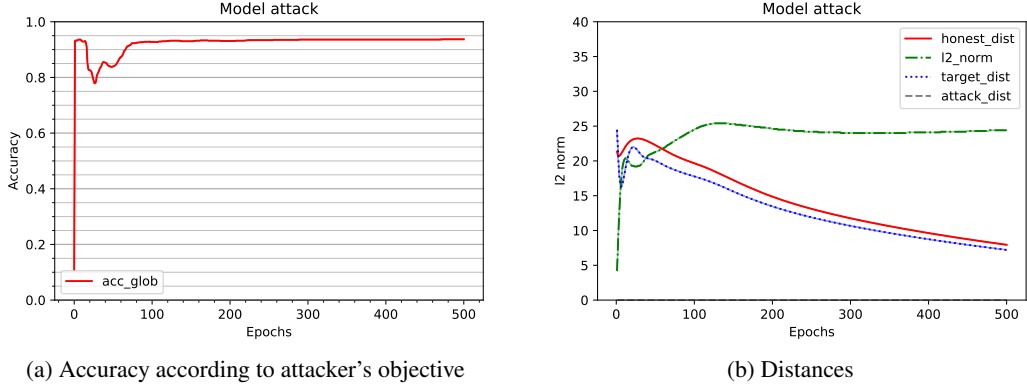

(a) Accuracy according to attacker's objective                    (b) Distances

Figure 18: Model attack on cifar-10.

### J.3    RECONSTRUCTING DATA POISONING

This last step is however nontrivial. On one hand, we could simply use the attack model to label a large number of random images. However, this solution would likely require a large sample com-

plexity. For a more efficient data poisoning, we can construct vector embeddings on the indifference affine subspace $V$, as was done for MNIST in Section 5.3. This is what is shown below.

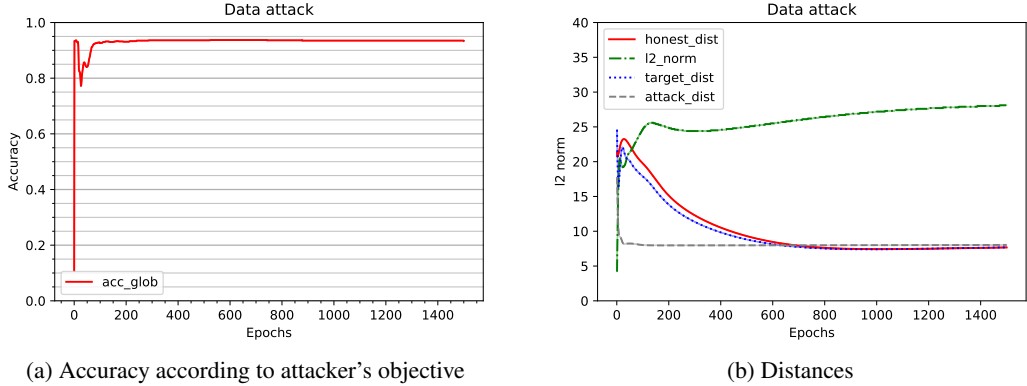

(a) Accuracy according to attacker's objective

(b) Distances

Figure 19: Data poisoning on cifar-10.

We acknowledge however that this does not quite correspond to data poisoning, as it requires reporting a vector embedding and its label, rather than an actual image and its label. The challenge is then to reconstruct an image that has a given vector embedding. We note that, while this is not a straightforward task in general, this has been shown to be at least somewhat possible for some neural networks, especially when they are designed to be interpretable (Zeiler & Fergus, 2014; Wang et al., 2019c; Mai et al., 2019).

