# OpenReview forum: "An Equivalence Between Data Poisoning and Byzantine Gradient Attacks"
_ICLR.cc/2022/Conference — ICLR 2022 Submitted_

### Official Review · Reviewer_bDXf · 2021-11-01

**Correctness:** 3
**Technical Novelty And Significance:** 3
**Empirical Novelty And Significance:** 2
**Recommendation:** 3
**Confidence:** 4

**Main Review:**

Pros:

The paper studies an interesting problem, since both Byzantine attacks and data poisoning attacks have been extensive studied in the literature, but little is known about how these fields relate. The paper provides a substantial technical analysis, as well as a practical Byzantine attack.


##########################################################################

Cons:

Unfortunately, several concern prevent me from recommending acceptance.

- Firstly, I find the comparison to related work very insufficient.

For the theoretical analysis a number of recent works, for example [1,2,3,4], have explored (robust) machine learning from multiple datasets from a PAC perspective. These essentially correspond to the notion of PAC* learnability presented here, with and without data poisoning. It is unexplained how these works compare to the current one.

For the gradient attack that is designed, there is no comparison to prior attacks as well. It is also unclear what this attack has to do with the rest of the paper - can the authors explain?

- The paper is also slightly hard to follow, in particular the results of Theorem 4,5 and 6 are quite hard for me to interpret. Since there is no previously defined notions of success of for each of the attacks, it is hard to understand why the results imply any type of equivalence between the notions. Further discussion immediately after these results will therefore be helpful.

#########################################################################

References:

[1] A. Blum, N. Haghtalab, A. D. Procaccia, M. Qiao: Collaborative PAC Learning, NIPS 2017
[2] M. Qiao: Do outliers ruin collaboration?, ICML 2018
[3] N. Konstantinov, E. Frantar, D. Alistarh, C.H. Lampert: On the Sample Complexity of Adversarial Multi-Source PAC Learning, ICML 2020
[4] A. Jain and A. Orlitsky: A General Method for Robust Learning from Batches, NeurIPS 2020



**Summary Of The Paper:**

The paper studies the relationship between Byzantine gradient attacks in distributed optimization and data poisoning. Some theoretical evidence is presented about the equivalence of these. A new gradient-based Byzantine attack is also presented.

**Summary Of The Review:**

While the studied topic and the proposed notions and results are interesting, I believe that the paper will benefit from substantial further comparison to existing work and also from further discussion of the results from Theorems 4,5,6.

---

> ### Author Response · Authors · 2021-11-15
> **Response to Reviewer bDXf**
>
> We thank the reviewer for their review. We have made significant revisions to our paper to address their remarks. Below, we respond to the reviewer’s specific questions.
>
> > Firstly, I find the comparison to related work very insufficient. For the theoretical analysis a number of recent works, for example [1,2,3,4], have explored (robust) machine learning from multiple datasets from a PAC perspective. These essentially correspond to the notion of PAC* learnability presented here, with and without data poisoning. It is unexplained how these works compare to the current one.
>
> Thank you for suggesting the references! We happily discussed their relationship to our work in the updated version of our paper (see Page 3).
>
> Indeed, these papers share similarities with our framework. The main difference, however, is that all of the references assume that all honest users (or data sources) have the same true labeling function (which we call “true model” or "preferred model"). We stress that, in many applications, like language processing or recommendation systems, this is an unrealistic assumption. In fact, a key motivation of personalized federated learning is to allow each user to learn a model that fits their true labeling function.
> Another important difference is that, despite a very interesting and valuable theory, the algorithms proposed by these references usually cannot be used in practice, e.g, see section 5 of [2] which is probably the most relevant to the topic of our paper (again from a learning theory perspective). In particular, they fail to appropriately model the class of gradient-based learning algorithms that has become widespread in recent years.
>
> > For the gradient attack that is designed, there is no comparison to prior attacks as well. It is also unclear what this attack has to do with the rest of the paper - can the authors explain?
>
> As opposed to the gradient attacks presented in [5,6], we assume here that the strategic user’s goal is not to harm the system; rather they aim to bias the learning towards some target model. As a result, our attack is hardly comparable with prior attacks.
>
> [5] Gilad Baruch, Moran Baruch, Yoav Goldberg: A Little Is Enough: Circumventing Defenses For Distributed Learning. NeurIPS 2019.
>
> [6] Cong Xie, Oluwasanmi Koyejo, Indranil Gupta: Fall of Empires: Breaking Byzantine-tolerant SGD by Inner Product Manipulation. UAI 2019.
>
> >The paper is also slightly hard to follow, in particular the results of Theorem 4,5 and 6 are quite hard for me to interpret. Since there is no previously defined notions of success of for each of the attacks, it is hard to understand why the results imply any type of equivalence between the notions. Further discussion immediately after these results will therefore be helpful.
>
> We have significantly revised our paper to make it clearer. In particular, our main theorem is now simplified, and presented as Theorem 1 in the revised version. It essentially aggregates what used to be Theorems 4, 5 and 6, which are now presented as lemmas.
> Essentially, the new Theorem 1 proves the equivalence between data poisoning and gradient attack (and model attack). In other words, for any definition of “sucessful” (that relates to the learned global model), there exists a successful data poisoning attack if and only if there exists a successful (converging) gradient attack.
> This theorem is also illustrated by our empirical results, which show how to turn in practice any gradient attack into a data poisoning attack.

---

### Official Review · Reviewer_a13A · 2021-11-04

**Correctness:** 4
**Technical Novelty And Significance:** 2
**Empirical Novelty And Significance:** 3
**Recommendation:** 5
**Confidence:** 3

**Main Review:**

The considered problem is of great contemporary importance in the deployment of large-scale machine learning systems. 1. The formalization of the PAC learnability is intuitive and concrete examples linear regression, logistic regression, are analyzed to illustrate this definition. 2. The equivalence between data poisoning, model attacks, and gradient attacks are carefully analyzed. 3. The proposed CGA attack has theoretical guarantee and demonstrates good empirical performance.

In general, I think the assumptions used in this paper are strong. 1. The sufficient condition proposed for PAC* learnability is not intuitive to me, is more like a sufficient condition for the two concrete examples analyzed, and lack intuitions for a wider class of functions. Like, does this condition or the PAC* learnability holds for any non-convex loss used in practice? 2. It might also help to have more detailed explanations for the considered data poisoning, model attacks and gradient attacks, like giving concrete examples to illustrate. The assumptions for theorem 6 to hold also seem strong to me, it is not clear to me how to interpret the assumed property of R, especially R(\rho, \theta) = R_0(\rho - \theta). It might be clearer to draw a diagram to show under which conditions the claimed equivalences hold, since the claim in the paper title and abstract is fairly strong. 3. I also would suggest giving proof outlines for main theorems for the paper. 4. Another question is how should we relate the proposed CGA with the previous sections, is there any motivations therein?


**Summary Of The Paper:**

The paper formalizes the concept of PAC*-learnability for a generalized personalized federated learning model and provides a sufficient condition. Under this model, converging gradient attack is equivalent to data poisoning. The paper also proposes the counter-gradient attack that is effective and data-efficient.

**Summary Of The Review:**

I would recommend that the paper is marginally below the acceptance threshold. Mainly because I don’t quite follow the intuitions of some assumptions. Refined analysis with weaker assumptions and clearer presentation would be appreciated.

---

> ### Author Response · Authors · 2021-11-15
> **Response to Reviewer a13A**
>
> We thank the reviewer for their thorough review, which has enabled us to improve the paper in a newly uploaded version. Below, we respond to the reviewer’s specific remarks.
>
> > In general, I think the assumptions used in this paper are strong. 1. The sufficient condition proposed for PAC* learnability is not intuitive to me, is more like a sufficient condition for the two concrete examples analyzed, and lack intuitions for a wider class of functions.
>
> We have significantly rewritten the section of the paper (Section 4 in the revised paper) on the PAC* learnability (now called “local PAC* learnability") of linear and logistic regression, to make clear that our goal is to prove this result. In particular, the sufficient condition discussed by the reviewer is now just a lemma.
>
> > Like, does this condition or the PAC* learnability holds for any non-convex loss used in practice?
>
> In general, no, it does not hold. We added a new brief section to the revised paper (Section 4.2) which discusses the case of neural networks. While we acknowledge that the definition should be adapted to neural networks, it is nevertheless very relevant, if we focus on the last layer of a pretrained neural network, designed to perform classification or regression.
>
> We are currently working on some experiments to evaluate this claim in practice, and we hope to manage to do this and update the paper before the response deadline.
>
> > 2. It might also help to have more detailed explanations for the considered data poisoning, model attacks and gradient attacks, like giving concrete examples to illustrate. The assumptions for theorem 6 to hold also seem strong to me, it is not clear to me how to interpret the assumed property of R, especially R(\rho, \theta) = R_0(\rho - \theta). It might be clearer to draw a diagram to show under which conditions the claimed equivalences hold, since the claim in the paper title and abstract is fairly strong.
>
> To simplify the presentation of the paper, we have restated our main theorem (now Theorem 1 in the revised paper) in a simpler setting. Namely, we merely consider $l_2^2$ and smooth-$l2$ regularization, for which the assumed property on $R$ clearly holds.
> Note that these assumptions are needed even to define the key objects of the paper. Namely, it allows us to guarantee that the infimum of the loss function is reached within a bounded region, which guarantees its existence, which then allows us to guarantee that $\rho^*$ is well-defined.
> Similarly, the smoothness assumptions are used to easily guarantee the convergence of the gradient descent algorithm.
>
> > 3. I also would suggest giving proof outlines for main theorems for the paper.
>
> In the revised version, we have made an effort to better highlight the main theorems, and to more clearly explain the main steps of their proofs. Unfortunately, the space limit is preventing us from dwelling too much on the proof ideas.
>
> >4. Another question is how should we relate the proposed CGA with the previous sections, is there any motivations therein?
>
> We have significantly reorganized the paper to make this clearer. CGA is now part of the global data poisoning attack. Our motivation is to show how easy it is to attack federated gradient descent with gradient attacks, and how this can be leveraged to then design a data poisoning attack.
>
> > I would recommend that the paper is marginally below the acceptance threshold. Mainly because I don’t quite follow the intuitions of some assumptions. Refined analysis with weaker assumptions and clearer presentation would be appreciated.
>
> We have made efforts to clarify the paper, by stating more specific-case theorems, which are easier to state, and by moving their general forms to the Appendix. In particular, our more convoluted assumptions are often no longer needed, as they are satisfied by the specific cases we consider. We hope that this helps make the presentation significantly clearer.

---

### Official Review · Reviewer_oiiD · 2021-11-08

**Correctness:** 4
**Technical Novelty And Significance:** 3
**Empirical Novelty And Significance:** 2
**Recommendation:** 5
**Confidence:** 4

**Main Review:**

The goal of this paper to compare two threat models in federated learning 1) data poisoning attacks that inject a number of poisoning examples to the training set 2) gradient attacks that change the update gradients sent during the federated learning operation by malicious agents. They specifically study the personalized federated learning setting where N machines collaborate to train N personalized models and a single global model. In this setting, the loss function is defined in a way that the global model and personalized models have dependencies to each other through an additive regularization that controls how far the personalized models can be from the global model.

The paper first defines two notions that they call PAC^* Learning and gradient PAC* learning for the personalized federated learning. The notion of PAC* learning is about finding a set of personalized models that are close to the set of true models with respect according to some metric. Note that this is different from PAC learning because they care about distance, rather than accuracy. They also define gradient-PAC*, which requires the gradients of the personalized models with respect to any global model \rho to be close to that of true models.

Then, they introduce a hypothetical threat model called a model attack, where they fix one of the personalized models to a specially crafted adversarial model. Then they show that, if the optimization of loss function has PAC* learning abilities, then the adversary can simulate the effect of the model attack using data poisoning attack, by poisoning all of the data for the machine, if the number of data-points provided by that machine is larger than some threshold. This result is expected, because as the number of data-points grow, the dependence between the personalized model and the global model decrease (because of the diminishing effect of the regularization). Then, by increasing the number of data points, because of PAC* learning, the adversary can change the personalized model to a adversarially crafted model, by using queries that are labeled accordingly. At the same time, the global model and other personalized models would not change much because of the diminish dependency of the global and personalized models.

Now the remaining step is to show that the gradient attack can be simulated by a model attack. In order to show this, they need some strong assumptions. For instance, they need the loss function to be strongly convex. They need the poisonous gradients to be valid gradients according to the global model. and also they require the poisonous gradients to converge. This step is technically interesting and novel, in my opinion.

Finally, they design a data poisoning attacks that are inspired by the steps mentioned above. They empirically show that this attack is effective in linear regression.


Limitations:

- The final equivalence result of the paper only applies to simple settings with strongly convex loss functions and seems to only work for GD optimization.

- The presentation of the work is not in publication stage. It is really hard to understand the main claims of the paper.

- Some discussions about the implication of the results are missing


Comments/Questions to authors:

1- I have read through the paper multiple times, but I still do not understand the exact threat model of gradient attack. I think you need to define this explicitly.

2- The notions of PAC* learning are not defined properly. There are terms such as "true model"  and "honest dataset" that are not defined. These terms are not standard and must be defined. Also, there is a probability \delta, what is this probability over?

3- Section 3 seems to be completely out of context. Why do you need to justify the choice of loss function before describing the main results? I was really confused by this section. After reading the paper, I realize that you need to define the loss this way in order for Theorem 5 to hold? If this is correct, you need to explicitly mention it. I also find Table 1 confusing.

4- Defining the loss function as a summation of the loss is fine, but your notion of PAC* learning talks about large number of data points. Wouldn't large number of data points make the dependence of global and personalized models weaker as more data is provided? How is it justified to keep the regularization the same as more data is provided?

5- Again, section 4 seems to be out of place. It is a bit odd to see the main results of the paper in section 5. I would recommend to  change the order of sections 4 and 5.

6- The equivalence between the model attacks and data poisoning attack depend on the number of data points. I think the paper needs discussion on this connection. Could the power of model attacks be significantly higher than data poisoning attacks when the number of data points provided by each party is a fixed and perhaps small number?

7- Theorem 6 seems like the main theorem in the paper to me. But I cannot verify the proof. Specifically, I cannot verify equation (120) in page 33. Can you explain why this is the case? Aren't \rho and \theta_n getting optimized together. In your formulation it seems like \theta_n is optimized until convergence for each intermediate \rho.

**Summary Of The Paper:**

This paper shows an equivalence between data poisoning attacks and gradient attacks that attack gradient descent on personalized federated learning setting for certain convex loss functions.

**Summary Of The Review:**

I find the topic of this paper extremely interesting. Understanding the relation between gradient attacks and data poisoning attacks are very important. This paper takes an initial step in understanding the relation between these two attacks for simple models such as linear regression. However, as stated in my comments, I have serious concerns about the presentation of this work. The main ideas presented are not easy to comprehend. The exact threat models are not also clearly specified. I also have some concerns about the proofs (In particular about equation 120, page 33). I will be happy to increase my score if authors can provide sufficient response to my concerns.

---

> ### Author Response · Authors · 2021-11-15
> **Response to Reviewer oiiD (1/2)**
>
> >The final equivalence result of the paper only applies to simple settings with strongly convex loss functions and seems to only work for GD optimization.
>
> We acknowledge that we consider the GD optimization, as our analysis means to relate gradient attacks to data poisoning. We added a section (Section 2.2 in the revised version), which clarifies our setting.
> Note however that, in the context of a neural network, an attacker may focus its attack only on the parameters of the last layer, which makes the loss relevant to this attack essentially convex.
> We added a new brief section to the revised paper (Section 4.2) which discusses this point.
> We are currently working on some experiments to evaluate this claim in practice, and we hope to manage to do this and update the paper before the response deadline
>
> >1- I have read through the paper multiple times, but I still do not understand the exact threat model of gradient attack. I think you need to define this explicitly.
>
> We have clarified all attack models in a dedicated section in the revised paper (Section 2.2).
> Essentially, we consider the classical federated gradient descent scheme, in a synchronous environment with no crash, where a parameter server manages the global model, and updates it at each iteration based on the users’ gradients.
> We hope that our new presentation of the setting makes the paper significantly more understandable.
>
> >2- The notions of PAC* learning are not defined properly. There are terms such as "true model" and "honest dataset" that are not defined. These terms are not standard and must be defined. Also, there is a probability \delta, what is this probability over?
>
> Before defining PAC* learning (Section 2.1 in the revised version, which was also renamed “local PAC* learning” for more clarity), we added the following paragraph:
>
> “In this paper, we focus on personalized learning algorithms that provably recover a user $n$'s \emph{preferred model} $\theta_n^\dagger$, if the user provides a large enough \emph{honest dataset} $\mathcal D_n$, i.e. constructed with $\theta_n^\dagger$. Such honest datasets $\mathcal D_n$ could typically be obtained by repeatedly drawing random queries (or features), and by using the user's preferred model $\theta_n^\dagger$ to provide answers (or labels). We refer to Section 4 for examples. The model recovery condition is then formalized as follows.”
>
> > 3- Section 3 seems to be completely out of context. Why do you need to justify the choice of loss function before describing the main results? I was really confused by this section. After reading the paper, I realize that you need to define the loss this way in order for Theorem 5 to hold? If this is correct, you need to explicitly mention it. I also find Table 1 confusing.
>
> In the revised paper, we postponed the justification of the loss function to Section 4, so that the main result of the paper (Theorem 1 in the revised version) could be presented first.
> What is now Section 4 aimed to prove that our PAC* learning assumption was not unrealistic as it applies to classical learning frameworks. However, we acknowledge that our justification of our loss function may have been too lengthy and too general. Instead, in the updated version, we postponed some discussions to the Appendix (especially about using sums instead of expectations, including Table 1). We also made Section 3 much easier to read by stating more specific results rather than our general results.
>
> >4- Defining the loss function as a summation of the loss is fine, but your notion of PAC* learning talks about large number of data points. Wouldn't large number of data points make the dependence of global and personalized models weaker as more data is provided? How is it justified to keep the regularization the same as more data is provided?
>
> Indeed, as discussed in the paper, taking a sum allows to make a user n’s model more independent from other users, when the user n has a large amount of data. This is quite intuitive since when a user has a large dataset, the model trained on their local date would be more reliable. Thus this user does not need to be affected by other users’ data points. This intuition is discussed in “sum over average” section (which has been moved to appendix).
> Note that decreasing the regularization is then not needed, as the data fitting term will already grow proportionally to the number of data. For a large number of data, the regularization will then have a minimal impact (which is formally proved by our PAC* learning theorems).
>
> >5- Again, section 4 seems to be out of place. It is a bit odd to see the main results of the paper in section 5. I would recommend to change the order of sections 4 and 5.
>
> We fully agree with the reviewer. Thanks to their useful remarks, we have made significant revisions to the paper to improve its structure. We hope that the revised version is now significantly easier to follow.

---

> > ### Author Response · Authors · 2021-11-15
> > **Response to Reviewer oiiD (2/2)**
> >
> > > 6- The equivalence between the model attacks and data poisoning attack depend on the number of data points. I think the paper needs discussion on this connection. Could the power of model attacks be significantly higher than data poisoning attacks when the number of data points provided by each party is a fixed and perhaps small number?
> >
> > Our theoretical results indeed require a potentially large number of poisoning data provided by the strategic user.
> > However, remarkably, our experiments show that, in fact, a surprisingly small number of poisoning data suffices. Namely, in our experiments, a single strategic user with only 2,000 data points successfully biases the global model to a zero-accuracy model (in fact, to any desirable model), despite collaborating with 10 other users who are given 6,000 honest data each.
> >
> > > 7- Theorem 6 seems like the main theorem in the paper to me. But I cannot verify the proof. Specifically, I cannot verify equation (120) in page 33. Can you explain why this is the case? Aren't \rho and \theta_n getting optimized together. In your formulation it seems like \theta_n is optimized until convergence for each intermediate \rho.
> >
> > **Note that equation (120) is equation (49, page 21) in the updated version of the paper.**
> >
> > Yes, $\theta_n$ is optimized until convergence. As discussed in the paper, we consider a gradient descent approach similar to Dinh et al. (2020) where at each iteration, first the local models are optimized and then we take one gradient step for the global model. The difference with Dinh et al. (2020) is that we do not consider batches. This actually makes the attack even more difficult as in our framework the gradients coming from the honest nodes are very accurate (computed on the whole dateset).
> >
> > Now note that $\nabla Loss_s^1(\rho^t)$ is the sum of all gradient vectors received from all users assuming the strategic user $s$ sends the vector $g_s^\infty$ in all iterations. To compensate the error we add the difference between $g_s^t$ and $g_s^\infty$ to $\nabla Loss_s^1(\rho)$.
> >
> > We added this clarification to the paper (in Section 2.2) and we hope it makes our proof easier to verify.
> >
> > > I find the topic of this paper extremely interesting. Understanding the relation between gradient attacks and data poisoning attacks are very important. This paper takes an initial step in understanding the relation between these two attacks for simple models such as linear regression. However, as stated in my comments, I have serious concerns about the presentation of this work. The main ideas presented are not easy to comprehend. The exact threat models are not also clearly specified. I also have some concerns about the proofs (In particular about equation 120, page 33). I will be happy to increase my score if authors can provide sufficient response to my concerns.
> >
> > Thank you again to the reviewer. We hope that our replies addressed their concerns.

---

> > > ### Comment · Reviewer_oiiD · 2021-11-22
> > > **Thanks for the response**
> > >
> > >
> > >
> > > I appreciate authors effort in improving the paper. I believe the new version is much easier to understand.
> > >
> > > The authors response clarified the federated learning setup for me. However, I think the setup is not aligned with what happens in practice. First, if each user has so much data (to satisfy the PAC* requirements), and wants the local models to be independent from others, why should they participate in the federated learning? They can just train their own models.  Second, training local models to convergence at each iteration is not a common setting. It almost feels like the setting is specifically designed for the equivalence result of this paper to hold. Given this, I think the contribution of the paper is a bit narrow at this point and would stay with my current score.

---

> > > > ### Author Response · Authors · 2021-11-23
> > > > **Thank you for your time! We addressed your concerns**
> > > >
> > > > >I appreciate authors effort in improving the paper. I believe the new version is much easier to understand. The authors response clarified the federated learning setup for me. However, I think the setup is not aligned with what happens in practice.
> > > >
> > > > We thank the reviewer for the time and valuable feedback. We believe both of the shortcomings mentioned by the reviewer are due to a misunderstanding. We will elaborate on each of them in the following:
> > > >
> > > > >First, if each user has so much data (to satisfy the PAC* requirements), and wants the local models to be independent from others, why should they participate in the federated learning? They can just train their own models.
> > > >
> > > > Note that we do not assume that each honest user must have many data points. In fact, we do not make any assumption on the number of data points provided by the users and there might exist some users in the system without any data point in the system (in this case the local model of these users would be equal to the global model).
> > > > PAC* (Definition 1) states that if a user has enough data, the system will eventually learn their preferred model for them. Our main result is that this desirable feature of federated learning necessarily leads to a vulnerability, or more precisely, to an equivalence between gradient attacks and data poisoning.
> > > >
> > > > >Second, training local models to convergence at each iteration is not a common setting. It almost feels like the setting is specifically designed for the equivalence result of this paper to hold. Given this, I think the contribution of the paper is a bit narrow at this point and would stay with my current score.
> > > >
> > > > Training local models to convergence at each iteration is not a requirement for our equivalence. Our equivalence result holds as long as the global model is updated based on the gradient vectors received from the users, and this is the case for all federated learning algorithms that we are aware of. We will make this clearer in the next version of the paper. Training local models to convergence (which is based on [ Dinh et al. (2020)]) is only required to guarantee the convergence of the counter gradient attack.

---

### Official Review · Reviewer_xXNa · 2021-11-08

**Correctness:** 4
**Technical Novelty And Significance:** 3
**Empirical Novelty And Significance:** 4
**Recommendation:** 6
**Confidence:** 3

**Main Review:**

Strengths:
+ Reveals a new insight on the equivalence between Byzantine gradient and data poisoning attacks, which is an important and timely topic.
+ A novel PAC framework for analyzing personalized federated learning performance.

Weaknesses:
- Gradient attack model is limited.
- Experiments are limited to simple datasets.

**Summary Of The Paper:**

This paper reveals the inherent equivalence between gradient and data poisoning attacks in personalized federated learning settings. The authors showed that any gradient attack can be transformed into data poisoning in a personalized federated learning system that provides PAC guarantees. This new insight challenges the view that (Byzantine) gradient attacks are unrealistic. The authors built this equivalence by constructing a model attack for personalized federated learning models. The authors showed the effectiveness of this attack both theoretically and empirically.

**Summary Of The Review:**

This paper reveals an equivalence between Byzantine gradient and data poisoning attacks in the context of personalized federated learning, which is an important and timely topic. This insight suggests that the claims in the existing literature that Byzantine gradient attacks are unrealistic is misleading. To my knowledge, this result is new and its contributions to the field are significant. However, this paper also has several issues, which are listed as follows:

1. The attack model considered in this paper is somewhat simplistic in that the authors only considered the single-strategic-user case. Although it is understandable that this renders the problem more tractable for theoretical analysis, the results may not be very useful because Byzantine gradient attacks are not necessarily from a single malicious user. In fact, most works on Byzantine gradient attacks allow multiple Byzantine workers. It's unclear whether the results in this paper could be extended to multi-attacker scenarios.

2. The authors proposed a PAC framework as a foundation to evaluate the performance of various Byzantine and data poisoning attacks. To my knowledge, this is also a new and interesting contribution. The authors demonstrated the relevance of this PAC framework by showing that linear regression and classifications are PAC learnable under this framework. But I wonder whether this PAC framework continues to be meaningful for more complex learning models than linear models. Having further discussions on this aspect would be very interesting.

3. Most of the experiments in this paper are conducted on MNIST and Fashion-MNIST datasets, which are relatively simple. It would be more interesting to demonstrate the claimed equivalence between attacks on more sophisticated datasets. Also related to the previous comment, most of the experiments are based on linear models (a simple two-layer neural network is also used). I think this paper could benefit from more experiments with more sophisticated learning models.

---

> ### Author Response · Authors · 2021-11-15
> **Response to Reviewer xXNa**
>
> We thank the reviewer for their thorough review, which has enabled us to significantly improve the paper in a newly uploaded version. Below, we respond to the reviewer’s specific remarks.
>
> > The attack model considered in this paper is somewhat simplistic in that the authors only considered the single-strategic-user case. Although it is understandable that this renders the problem more tractable for theoretical analysis, the results may not be very useful because Byzantine gradient attacks are not necessarily from a single malicious user. In fact, most works on Byzantine gradient attacks allow multiple Byzantine workers. It's unclear whether the results in this paper could be extended to multi-attacker scenarios.
>
> Our main focus is on the vulnerability of established personalized learning framework, with l_2^2 regularization. We prove **the stronger result** that they are vulnerable **even** when a single user behaves strategically; clearly the vulnerability remains if several strategic users collude (multi-attacker scenario).
>
> > The authors proposed a PAC framework as a foundation to evaluate the performance of various Byzantine and data poisoning attacks. To my knowledge, this is also a new and interesting contribution. The authors demonstrated the relevance of this PAC framework by showing that linear regression and classifications are PAC learnable under this framework. But I wonder whether this PAC framework continues to be meaningful for more complex learning models than linear models. Having further discussions on this aspect would be very interesting.
>
> The reviewer makes a very relevant remark. We added a new brief section to the revised paper (Section 4.2) which discusses the case of neural networks. While we acknowledge that the definition should be adapted to neural networks, it is nevertheless very relevant, if we focus on the last layer of a pretrained neural network, designed to perform classification or regression.
>
> We are currently working on some experiments to evaluate this claim in practice, and we hope to mange to do this and update the paper before the response deadline.
>
> > Most of the experiments in this paper are conducted on MNIST and Fashion-MNIST datasets, which are relatively simple. It would be more interesting to demonstrate the claimed equivalence between attacks on more sophisticated datasets. Also related to the previous comment, most of the experiments are based on linear models (a simple two-layer neural network is also used). I think this paper could benefit from more experiments with more sophisticated learning models.
>
> We are currently working on some experiments using more sophisticated datasets, e.g., Cifar10 and more complex models.

---

> > ### Comment · Reviewer_xXNa · 2021-11-30
> > **Response to Rebuttal**
> >
> > Thanks for the response to my review. After reading the rebuttal, I'm still not very convinced by the argument that a single strategic malicious user is stronger than multiple malicious (independent) users, at least it's not immediately clear. The rebuttal addressed my other concerns. Overall, I found this paper interesting and reveal some fresh insights between gradient and data poisoning attacks, but there remain some limitations. Thus, I will keep my original rating "6".

---

> > > ### Author Response · Authors · 2021-12-09
> > > **Clarification on single attacker.**
> > >
> > > We thank the reviewer for their feedback on our rebuttal.
> > > On the remaining concern about single vs multiple attackers: Note that in the paper we prove a single attacker can bais the model to any desired point. Now if we consider $f$ attackers, they can still bias the model to any desired point and our results hold. In the worst case, $(f-1)$ attackers can act as honest and only one of them performs the attack. However, we agree this is not the best strategy for the attackers. The best strategy for them would be to all perform the attack at the same. We will add this remark and adapt our results accordingly (this change is quite straightforward). Thank you again for the time you spent on our paper.

---

### Official Review · Reviewer_gjXp · 2021-11-11

**Correctness:** 3
**Technical Novelty And Significance:** 2
**Empirical Novelty And Significance:** 1
**Recommendation:** 5
**Confidence:** 3

**Main Review:**

Model poisoning are generally perceived as stronger attacks (in terms of poisoning impact) than data poisoning attacks, simply because data poisoning only indirectly manipulates a client’s model update. Therefore, showing an equivalence as the paper argues would be very interesting. However, in my view, the paper is written in somewhat confusing manner, and lacks in details. Therefore, it is difficult to understand and assess the claims. (More detailed comments are given below.) Moreover, the paper does not compare the proposed Counter-Gradient attack with existing attacks. There are a large number of existing attacks and defenses, and it would be important to place the proposed attack in the larger context by comparing and contrasting with prior attacks and defenses.

Detailed Comments:

1. Definition 1 relies on ‘true models’ \theta^{perp}. It will be good to give more details about what do the authors mean by true models. Propositions 1 and 2 on computing poisoned gradients also use true models. It is not clear how users will know about these true models? It will be important to elaborate this further.

2. Is the difference between using sum vs. expectation in the loss not related to empirical vs. population risk? Also, computing the sum  of losses over the entire user dataset is essentially gradient descent, which is known to be computationally burdensome. It will be helpful to comment on these points.

3. In Definition 2, a loss function is defined to be PAC-learnable. This is a bit confusing. PAC-learning is usually defined for a class of functions. It would be good to explicitly define the function class.

4. In Theorem 2, how is \tilde{Q} defined and how is the Supp() defined?

5. Theorem 1 restricts the queries to be i.i.d. sub-Gaussian. This seems to be quite restrictive. It will be good to comment on this.

6. Theorem 5 requires that a user’s dataset contains at least \mathcal{I} inputs drawn from a certain model. This is quite confusing. What does it mean to draw inputs from a model?

7. The paper says that [Shejwalkar et al. 2021] argued that model poisoning attacks are not realistic, and claims in the conclusion that their findings reverses this argument. It is not clear why an equivalence result would reverse the claim in [Shejwalkar et al. 2021]. This is because, as per my understanding, [Shejwalkar et al. 2021] also argue that data poisoning attacks are not very impactful (key lesson (2) on page 2). It would be important to give more details here.

**Summary Of The Paper:**

The paper studies data and model poisoning attacks in federated learning (FL). It argues that there is an equivalence between data poisoning and model poisoning attacks, by restricting attention to linear and logistic regression. The main technique is to leverage ideas from PAC-learning. Another key contribution is to propose a gradient poisoning attack.

**Summary Of The Review:**

It would be interesting to show an equivalence between model and data poisoning attacks in FL. However, the paper is written in somewhat confusing manner and lacks in details, which makes it difficult to understand and assess the claims.

---

> ### Author Response · Authors · 2021-11-15
> **Response to Reviewer gjXp (1/2)**
>
> We thank the reviewer for their review. We have made significant revisions to our paper to address their remarks. Below, we respond to the reviewer’s specific questions.
>
> > However, in my view, the paper is written in somewhat confusing manner, and lacks in details. Therefore, it is difficult to understand and assess the claims.
>
> We agree with the reviewer. We hope that our significantly revised version makes the main result of the paper a lot easier to understand and appreciate. In particular, our main result is now stated as Theorem 1.
>
> > Moreover, the paper does not compare the proposed Counter-Gradient attack with existing attacks. There are a large number of existing attacks and defenses, and it would be important to place the proposed attack in the larger context by comparing and contrasting with prior attacks and defenses.
>
> As opposed to the gradient attacks presented in [5,6], we assume here that the strategic user’s goal is not only to harm the system; rather they aim to bias the learning towards some target model. As a result, our attack is hardly comparable with prior attacks.
>
> [5] Gilad Baruch, Moran Baruch, Yoav Goldberg: A Little Is Enough: Circumventing Defenses For Distributed Learning. NeurIPS 2019.
>
> [6] Cong Xie, Oluwasanmi Koyejo, Indranil Gupta: Fall of Empires: Breaking Byzantine-tolerant SGD by Inner Product Manipulation. UAI 2019.
>
> > Definition 1 relies on ‘true models’ \theta^{perp}. It will be good to give more details about what do the authors mean by true models. Propositions 1 and 2 on computing poisoned gradients also use true models. It is not clear how users will know about these true models? It will be important to elaborate this further.
>
> Before defining PAC* learning (Section 2.1 in the revised version, which was also renamed “local PAC* learning” for more clarity), we added the following paragraph:
>
> “In this paper, we focus on personalized learning algorithms that provably recover a user $n$'s \emph{preferred model} $\theta_n^\dagger$, if the user provides a large enough \emph{honest dataset} $\mathcal D_n$, i.e. constructed with $\theta_n^\dagger$. Such honest datasets $\mathcal D_n$ could typically be obtained by repeatedly drawing random queries (or features), and by using the user's preferred model $\theta_n^\dagger$ to provide answers (or labels). We refer to Section 4 for examples. The model recovery condition is then formalized as follows.”
>
> In particular, instead of ‘true models’, we now talk of ‘preferred models’. The preferred models are the ones that users typically use to label their data. In the context of content recommendation on social media, this would correspond to their behaviors on social media (what they watch, like and share). In the context of language processing, it is the way they generate text. Importantly, we stress that a user’s preferred model may strongly differ from any other user’s preferred model.
>
> > Is the difference between using sum vs. expectation in the loss not related to empirical vs. population risk?
>
> This is not the case. Population risk, at least as defined in, e.g., [7], still corresponds to an expectation, but over a (usually unknown and continuous) out-of-sample distribution, as opposed to the in-sample empirical risk.
>
> Our discussion is however orthogonal to such considerations. Namely, we ask how the empirical (or population) risk of a given user should be compared to the empirical (or population) risk of another user in the global loss function. In particular, we argue that, as opposed to what has been done in published work, a user’s empirical (or population) risk should be weighed proportionally to the number of data collected by the user. Equivalently, this corresponds to considering sums rather than expectations.
>
> [7] Chi Jin, Lydia T. Liu, Rong Ge, Michael I. Jordan. On the Local Minima of the Empirical Risk. NeurIPS 2018.
>
> > Also, computing the sum of losses over the entire user dataset is essentially gradient descent, which is known to be computationally burdensome. It will be helpful to comment on these points.
>
> Our paper is not concerned about the optimization performance. Instead, we focus on the vulnerability of learning systems to adversaries, assuming that the optimization is successfully performed.
> Nevertheless, we point out that, while we ran experiments in the context of gradient descent, they can straightforwardly be adapted to stochastic gradient descent, with very similar empirical results.

---

> > ### Author Response · Authors · 2021-11-15
> > **Response to Reviewer gjXp (2/2)**
> >
> > >In Definition 2, a loss function is defined to be PAC-learnable. This is a bit confusing. PAC-learning is usually defined for a class of functions. It would be good to explicitly define the function class.
> >
> > Our framework technically considers more general objects than functions — typically users may want to learn distributions parameterized by $\theta$. Nevertheless, we have clarified the role of $\theta$, and thus the implicit function class that they define, in the revised paper. Namely, in the context of data labeling, $\theta$ defines the (possibly stochastic) function used to label the data (see Section 2.1 in the revised version).
> >
> > In the more specific case of linear or logistic regression, the function class then naturally becomes the set of linear function, or the set of logistic linear classifiers, as defined in Section 4 of the revised paper.
> >
> > We have also provided a short discussion in the case of neural networks (see Section 4.2).
> >
> > > In Theorem 2, how is \tilde{Q} defined and how is the Supp() defined?
> >
> > Note that Theorem 2 is Theorem 3 in the revised version of the paper.
> >
> > \tilde{Q} is the distribution of queries (input variable of data points) and Supp() is its support which is assumed to be bounded (as we discuss below the bounded support assumption is automatically satisfied in practice).
> >
> > > Theorem 1 restricts the queries to be i.i.d. sub-Gaussian. This seems to be quite restrictive. It will be good to comment on this.
> >
> > Note that Theorem 1 is Theorem 2 in the revised version of the paper.
> >
> > The sub-Gaussian family contains a very large number of random variables **including any random variable which is bounded** (See Example 2.4 in Wainwright (2019)). Therefore, this assumption is automatically satisfied for (arguably) all learning tasks (image, video, audio, ... ) as for instance for RGB image, each coordinate represents a color intensity with a value in [0, 256], so the underlying distribution would be bounded.
> >
> > The i.i.d assumption is also quite standard and widely used in machine learning.
> >
> >  > Theorem 5 requires that a user’s dataset contains at least \mathcal{I} inputs drawn from a certain model. This is quite confusing. What does it mean to draw inputs from a model?
> >
> > Note that Theorem 5 became Lemma 14 in the revised version of the paper.
> >
> >  Generative models are a common class of models [8]. They are generally defined as computing a probability distribution over a space (which may be of the form Features x Labels). Drawing an input from the model then amounts to sampling from the corresponding probability distribution.
> > We give specific example of data generation process in the context of linear and logistic regression (see Section 4).
> >
> > What is commonly done in practice, is to propose inputs (which we call queries) to different users, and to ask them to label these inputs (which we call answers). The user is then part of the data generation process. In our analysis, we then assume that there exists some model \theta that an honest user is implicitly using to determine how to answer the queries they are given.
> >
> > [8] Ian J. Goodfellow, Jean Pouget-Abadie, Mehdi Mirza, Bing Xu, David Warde-Farley, Sherjil Ozair, Aaron C. Courville, Yoshua Bengio: Generative adversarial networks. Commun. ACM
> >
> > > The paper says that [Shejwalkar et al. 2021] argued that model poisoning attacks are not realistic, and claims in the conclusion that their findings reverses this argument. It is not clear why an equivalence result would reverse the claim in [Shejwalkar et al. 2021]. This is because, as per my understanding, [Shejwalkar et al. 2021] also argue that data poisoning attacks are not very impactful (key lesson (2) on page 2). It would be important to give more details here.
> >
> > One of the main claims of Shejwalkar et al. 2021 is that data-poisoning attack is not very impactful on the model. On the other hand, they argue that while the gradient attack (model poisoning attack in their terminalogy) is much more impactful than data-poisoning, it is not practical as the number of compromised gradient providers is very low in practice. (see, e.g., below Takeaway VII-D of Shejwalkar et al. 2021). Our equivalence result implies that in order to guarantee the safety of the learning algorithm, one needs to consider the worst case, i.e.,  an attack which is both impactful (coming from gradient attack) and more practitcal and with a relatively higher percentage of adversary (coming from data poisoning).
> >
> > (Takeaway VII-D of Shejwalkar et al. 2021): “For cross-silo FL, model poisoning attacks are not practical and state-of-the-art data poisoning attacks have no impact even with non-robust Average AGR.”
> >
> > We provided more discussions about the importance of this equivalence in Section 2.2 of the revised paper.

---

> > > ### Comment · Reviewer_gjXp · 2021-11-21
> > > **Thanks for the responses. These were very helpful, but there are a few more questions.**
> > >
> > > Thanks to the authors for significantly revising the paper, and for responding to my questions. The revised manuscript and the response address several of my questions. The quality of the paper has improved, but there still remain several questions:
> > >
> > > 1. While describing Federated GD in Sec. 2.2, the authors mention that each user either solves an optimization problem (similar to [Dinh et al. 2020]) or takes a gradient step using their previous local model (similar to [Hanzley-Richtarik 2021]). Furthermore, it is stated that, for theory the authors consider the first approach while in experiments they consider the second approach. How is the global model used in the second approach? ([Hanzley-Richtarik 2021] probabilistically switch between using a global model and taking a local step in each round.) It would be important to add more details.
> > >
> > > 2. My main concern is that there seems to be a tension between the locally PAC* definition and attack objective in personalized FL. More specifically, according to Definition 1, for a locally PAC* learning algorithm, when all honest users have sufficiently large local datasets, the final local models will be close (in l2-norm) to users’ preferred models. Now, suppose FL system is running a locally PAC* algorithm. Then, how can a strategic user launch a successful data or gradient poisoning attack when all honest users have more than \mathcal{I} data points? In particular, even if the strategic user manages to influence the global model, all honest users would still learn local models that are close to their preferred models (with high probability) as per Definition 1. How can one say an attack was successful in this case? It will be really important to add a discussion on this point.
> > >
> > > 3. In Theorem 1, does the result need to be stated for sufficiently small \epsilon? Stating the result for any \epsilon > 0 seems a bit confusing. In particular, choose a very large \epsilon, say \epsilon = 1e6, and suppose there exists a dataset \mathcal{D}_s such that \|\rho*(D) - \theta^{\perp}_s\| \leq 1e6. Then, how would this imply a successful gradient attack?
> > >
> > > 4. 4. In Fig.3, how is the indifference affine subspace V chosen for the MNIST dataset? Can the authors please give some more details?

---

> > > > ### Author Response · Authors · 2021-11-22
> > > > **Thank you!**
> > > >
> > > > > Thanks to the authors for significantly revising the paper, and for responding to my questions. The revised manuscript and the response address several of my questions. The quality of the paper has improved,
> > > >
> > > > We thank the reviewer for their time and valuable feedback. Below, we respond to the reviewer’s specific questions.
> > > >
> > > > > 1. While describing Federated GD in Sec. 2.2, the authors mention that each user either solves an optimization problem (similar to [Dinh et al. 2020]) or takes a gradient step using their previous local model (similar to [Hanzley-Richtarik 2021]). Furthermore, it is stated that, for theory the authors consider the first approach while in experiments they consider the second approach. How is the global model used in the second approach? ([Hanzley-Richtarik 2021] probabilistically switch between using a global model and taking a local step in each round.) It would be important to add more details.
> > > >
> > > > Note that since the focus of our paper is to design and analyze loss functions but not the optimization part, in our experiments, we took the most basic optimization approach that showed acceptable convergence behavior, i.e., at each iteration of gradient descent, each local model takes one step which is the sum of **both** a local step and a step towards the global model. This approach is similar to [Hanzley-Richtarik 2021] in the sense that at each iteration we take one step using an unbiased estimator of the gradient (See equation (7) in [Hanzley-Richtarik 2021]), but we **do not** use the probabilistic algorithm as the main reason for using this approach in [Hanzley-Richtarik 2021] is to reduce the communication complexity and this is not the topic of our paper.
> > > >
> > > > >2. My main concern is that there seems to be a tension between the locally PAC* definition and attack objective in personalized FL. More specifically, according to Definition 1, for a locally PAC* learning algorithm, when all honest users have sufficiently large local datasets, the final local models will be close (in l2-norm) to users’ preferred models. Now, suppose FL system is running a locally PAC* algorithm. Then, how can a strategic user launch a successful data or gradient poisoning attack when all honest users have more than \mathcal{I} data points? In particular, even if the strategic user manages to influence the global model, all honest users would still learn local models that are close to their preferred models (with high probability) as per Definition 1. How can one say an attack was successful in this case? It will be really important to add a discussion on this point.
> > > >
> > > > The subtlety is that as stated in Definition 1, the required number of data points for honest users \mathcal{I} **also depends on the dataset provided by the attackers $D_{-H}$**. Therefore, if we fix the dataset of the attackers and let the honest users provide more data points we will indeed eventually learn the honest users’ preferred models. But this scenario is not what happens in practice. The reasonable assumption for an attack is that the attacker has some information about the dataset provided by the honest users and crafts its own dataset based on this information. Therefore, as long as the attacker’s dataset is not fixed in advance, no number of honest data points can guarantee that we will learn the honest users’ preferred models.
> > > > Note also that extending our attacks from the global model to honest users’ local models is quite straightforward, but for simplicity this is not discussed in the paper.
> > > >
> > > > > 3.In Theorem 1, does the result need to be stated for sufficiently small \epsilon? Stating the result for any \epsilon > 0 seems a bit confusing. In particular, choose a very large \epsilon, say \epsilon = 1e6, and suppose there exists a dataset \mathcal{D}_s such that |\rho*(D) - \theta^{\perp}_s| \leq 1e6. Then, how would this imply a successful gradient attack?
> > > >
> > > > Note that Theorem 1 (and other results in the paper) are stated for a general value of $\varepsilon$ that **could be chosen** arbitrarily small (e.g., $10^{-10}$). The smaller the value of  $\varepsilon$ is chosen, the more meaningful the attack will be and (probably) it would be more difficult to perform the attack.
> > > >
> > > > > 4.In Fig.3, how is the indifference affine subspace V chosen for the MNIST dataset? Can the authors please give some more details?
> > > >
> > > > The construction of V is fully explained in Appendix I of the paper.
> > > > In short, V is defined as the indifference subspace, i.e. the subspace where all labels are equally probable, according to the attack model. Because we used a linear classifier, this subspace naturally corresponds to a set of 9 linear equations, each saying that the label a must be as probable as the label 0, for all digits a from 1 to 9.

---

> > > > > ### Comment · Reviewer_gjXp · 2021-11-22
> > > > > **Thanks for the clarifications**
> > > > >
> > > > > Thanks to the authors for clarifying the queries that asked. It would be important to add a formal description of the update step in the main paper (or at least in an appendix) to avoid any confusion. Further, it would be good to add a discussion on the subtlety mentioned by the authors while answering the second question above.
> > > > >
> > > > > After these discussion with the authors, I understand the contributions much better. Overall, the paper shows an equivalence between data poisoning and gradient poisoning attacks (as stated in Theorem 1), which holds under fairly strict assumptions, in particular, when local loss functions are convex and smooth, when regularization is \ell_2^2 or smooth-\ell_2, and when learning algorithm is locally PAC* as per Definition 1. The results seem essentially restricted to linear and logistic regression. In my view, it would be important to explicitly mention the assumptions of convexity and regularization in the Introduction (and even in the Abstract).  Because of the fairly restricted assumptions, I believe the contributions are somewhat limited. Nevertheless, I think the paper takes interesting initial steps towards understanding a relationship between data and gradient poisoning in personalized federated learning. I have increased my score to reflect this.

---

### Author Response · Authors · 2021-11-15
**To all reviewers**

Thanks to the reviewers’ useful feedbacks, we have uploaded a new significantly revised version of the paper. In particular, following the reviewers’ advice, the order of the sections has changed, as well as the labeling of the theorems. We have also greatly simplified the statements of our theorems, by stating them for more specific settings. We hope that the new version makes our findings significantly more understandable.

---

> ### Author Response · Authors · 2021-11-24
> **New experiments on a neural network!**
>
> Dear reviewers and AC,
>
> As suggested by the reviewers and thanks to their useful remarks, we added new experimental results to the paper, which is testing different attacks on the last layer of a pre-trained neural network (VGG 13-BN) and with the Cifar10 dataset. The new plots and detailed discussion can be found in Appendix J of the new version of the paper.

---

### Decision · Program_Chairs · 2022-01-20

**Decision:**

Reject

**Comment:**

This paper presents an analysis showing the equivalence between gradient and data poisoning attacks in personalized federated learning settings. The paper contains an analysis of an attack that requires only a single corrupt learning agent, providing results in the setting of PAC learnable models.
The reviewers had several criticisms of the paper, some of which were addressed in the rebuttal.  The first is that the presentation of the paper was at times confusing, and the theoretical results were hard to interpret.  This has been addressed by several changes to the paper writing, including major changes to the layout.  The reviewers feel that other criticisms were not entirely addressed.  This includes the criticism that the experiments are in a fairly simplistic setting (GD on MNIST and Fashion MNIST), and that the theoretical results require strong assumptions and focus mostly on classical models that are learnable in convex frameworks.  While the reviewers agree there are interesting questions posed in this paper, the consensus seems to be that the experimental and theoretical results in this paper should be further revised, and that a future version of this paper will be a great candidate for publication.